# Provable Memory Efficient Self-Play Algorithm for Model-free Reinforcement Learning

**Na Li**[1], **Yuchen Jiao**[2], **Hangguan Shan**[1], **Shefeng Yan**[3]

[1]College of Information Science and Electronic Engineering, Zhejiang University, Hangzhou, China
[2]School of Information and Electronics, Beijing Institute of Technology, Beijing, China
[3]Institute of Acoustics, Chinese Academy of Sciences, Beijing, China

## Abstract

The thriving field of multi-agent reinforcement learning (MARL) studies how a group of interacting agents make decisions autonomously in a shared dynamic environment. Existing theoretical studies in this area suffer from at least two of the following obstacles: memory inefficiency, the heavy dependence of sample complexity on the long horizon and the large state space, the high computational complexity, non-Markov policy, non-Nash policy, and high burn-in cost. In this work, we take a step towards settling this problem by designing a model-free self-play algorithm *Memory-Efficient Nash Q-Learning (ME-Nash-QL)* for two-player zero-sum Markov games, which is a specific setting of MARL. ME-Nash-QL is proven to enjoy the following merits. First, it can output an $\varepsilon$-approximate Nash policy with space complexity $O(SABH)$ and sample complexity $\widetilde{O}(H^4SAB/\varepsilon^2)$, where $S$ is the number of states, $\{A, B\}$ is the number of actions for two players, and $H$ is the horizon length. It outperforms existing algorithms in terms of space complexity for tabular cases, and in terms of sample complexity for long horizons, i.e., when $\min\{A, B\} \ll H^2$. Second, ME-Nash-QL achieves the lowest computational complexity $O(T\mathrm{poly}(AB))$ while preserving Markov policies, where $T$ is the number of samples. Third, ME-Nash-QL also achieves the best burn-in cost $O(SAB\,\mathrm{poly}(H))$, whereas previous algorithms have a burn-in cost of at least $O(S^3AB\,\mathrm{poly}(H))$ to attain the same level of sample complexity with ours.

## 1 Introduction

In this paper, we consider the problem of multi-agent reinforcement learning (MARL), which focuses on developing algorithms for multiple agents to learn and make decisions in multi-agent environments. MARL has attracted a plethora of attention across various domains, including autonomous driving (Shalev-Shwartz et al., 2016), game playing (Silver et al., 2017), and social systems (Baker et al., 2020). Considering that online data collection, storage, and computation can be expensive, time-consuming, or high-stakes in the above applications, achieving memory efficiency and low sample and computational complexity is important in online MARL scenarios. Generally, the approaches to solving MARL can be categorized into model-free and model-based methods. The former involves directly learning an optimal/equilibrium policy for multiple agents without explicitly modeling the environment, such as friend-or-foe Q-Learning (Littman, 2001) and Nash Q-Learning (Hu & Wellman, Nov. 2003). They typically involve using techniques such as Q-learning, actor-critic methods, or policy gradients to optimize the policy. The latter relies on a learned or predefined model of the environment. This usually involves estimating state-transition probabilities and rewards based on agent behaviors and then utilizing these estimations to simulate possible results and choose actions, such as AORPO (Zhang et al., 2021) and OFTRL (Zhang et al., 2022). Although recent research (Bai et al., 2020; Jin et al., 2022; Liu et al., July 2021; Xie et al., Jun. 2022) has demonstrated the effectiveness of both model-free and model-based algorithms in MARL, a more comprehensive understanding of efficient memory, optimal sample complexity and minimal computational complexity is still lacking.

As a specific setting of MARL, two-player zero-sum Markov games (TZMG) are a fascinating area of research. Considering a tabular TZMG with $S$ states, actions $\{A, B\}$ for two players, and

Table 1: Sample complexity (the order of the regret bound, modulo log factors), computational/space complexity, the range of samples to attain $\widetilde{O}(H^4S/\varepsilon^2)$-sample complexity, and whether the output policy is the Markov/Nash policy, for algorithms: V-learning (Jin et al., 2022), Nash V/Q-learning (Bai et al., 2020), PReFI/PReBO (Cui et al., 2023), OMNI-VI (Xie et al., Jun. 2022), Nash-UCRL (Chen et al., 2022), Optimistic PO (Qiu et al., 2021), VI-Explore/VI-UCLB (Bai & Jin, 2020), Nash-VI (Liu et al., July 2021), and Q-learning (Feng et al., 2023). The best results are highlighted.

| Algorithm | Sample complexity $T$ | Computational/ space complexity | Range of samples to attain $H^4S/\epsilon^2$ sample complexity | Markov/ Nash policy |
|---|---|---|---|---|
| V-learning | $H^6S(A+B)/\varepsilon^2$ | $T\mathrm{poly}(AB)/$ $S(A+B)T$ | | No/No |
| Nash V-learning | $H^7S(A+B)/\varepsilon^2$ | | Never | |
| Nash Q-learning | $H^6SAB/\varepsilon^2$ | $T\mathrm{poly}(AB)/$ $SABT$ | | |
| PReFI | $H^{10}S^4(A+B)^4/\varepsilon^4$ | $T^2\mathrm{poly}(SAB)/$ $S(A+B)HT$ | Never | Yes/No |
| PReBO | $H^6S^2(A+B)/\varepsilon^2$ | $T^2\mathrm{poly}(SAB)/$ $SABHT$ | | |
| OMNI-VI | $H^5S^3A^3B^3/\varepsilon^2$ | $T\mathrm{poly}(SAB)/$ $S^2A^2B^2H$ | | Yes/Yes |
| Nash-UCRL | $H^4S^4A^2B^2/\varepsilon^2$ | | | |
| Optimistic PO | $H^5S^2AB/\varepsilon^2$ | $T\mathrm{poly}(SAB)/$ $S^2ABH$ | | |
| VI-Explore | $H^8S^2AB/\varepsilon^3$ | | | |
| VI-ULCB | $H^5S^2AB/\varepsilon^2$ | PPAD-complete/ $S^2ABH$ | | |
| Nash-VI | | $T\mathrm{poly}(SAB)/$ $S^2ABH$ | $[S^3ABH^4, \infty)$ | |
| Q-learning (Feng et al., 2023) | $H^4SAB/\varepsilon^2$ | $T+H^2\mathrm{poly}(SAB)/$ $SA^2B^2H^2$ | $[S^6A^4B^4H^{28}, \infty)$ | No/No |
| ME-Nash-QL | | $T\mathrm{poly}(AB)/$ $SABH$ | $[SABH^{10}, \infty)$ | Yes/Yes |

a horizon length of $H$, one important aspect of it is the sample complexity, which refers to the number of samples $T$ required to achieve an $\varepsilon$-approximate Nash equilibrium (NE). Currently, the Q-FTRL algorithm (Li et al., 2022), which employs a generative model, is the leading method for achieving optimal sample complexity in TZMG. However, the accessibility of the generative model is unclear and restrictive. In the absence of a flexible sampling mechanism, the self-play algorithms with minimal sample complexity are the model-free method V-learning (Jin et al., 2022) and the model-based method Optimistic Nash Value Iteration (Nash-VI) (Liu et al., July 2021), along with the FTRL idea (from adversarial bandit, i.e., $H = 1$) and a different style of bonus term, respectively. V-learning is able to find an $\varepsilon$-approximate policy with sample complexity of $\widetilde{O}(H^6S(A+B)/\varepsilon^2)$ (modulo log factors), but the output policy is non-Markovian. Nash-VI achieves the best sample complexity $\widetilde{O}(H^4SAB/\varepsilon^2)$ among model-based algorithms so far, while its burn-in cost $S^3ABH^4$ has a heavy dependence on $S$. Besides, the computational complexity of Nash-VI is $O(T\mathrm{poly}(SAB))$, higher than $T\mathrm{poly}(AB)$ in V-learning. Moreover, the space complexities of the above algorithms are unsatisfactory, with the former's $O(S(A+B)T)$ increasing with the number of samples $T$ and the latter's $O(S^2ABH)$ relying heavily on $S$. These complexities are large compared with the space $O(SABH)$ required by Q-value in tabular cases. A summary of prior results is shown in Table 1. Thus, a natural question motivated by prior algorithms to pose is:

*Can a TZMG algorithm be designed with memory, sample, and computational efficiency, while having low burn-in cost and a Markov and Nash output policy?*

## 1.1 Contributions

We contribute to the advancement of theoretical understanding by providing a sharp analysis of the model-free algorithm on TZMG. Our main contribution lies in the development of a novel model-free algorithm with a Markov and Nash output policy and theoretically achieves the best space and

computational complexities and the superior sample complexity compared to existing methods for long horizons, i.e., $\min\{A, B\} \ll H^2$. Specifically, we summarize our main contributions as follows.

- We design a model-free algorithm *Memory-Efficient Nash Q-Learning (ME-Nash-QL)*, which is generated from Nash Q-learning, along with the early-settlement method designed and the reference-advantage decomposition technique incorporated in TZMG for the first time. ME-Nash-QL firstly achieves the best memory complexity $O(SABH)$, corresponding to the minimum space to store $Q$-values in tabular cases. Furthermore, the computational complexity of our algorithm is $O(T\text{poly}(AB))$, which is lower than that of prior algorithms.

- We prove that ME-Nash-QL can find an $\varepsilon$-approximate NE for Markov games with samples $\widetilde{O}(H^4 SAB/\varepsilon^2)$, which is equivalent to achieving the regret bound $\widetilde{O}(\sqrt{H^2 SABT})$, provided that samples $T$ exceeds $\widetilde{O}(SABH^{10})$. We remark that the sample complexity of ME-Nash-QL has an optimal dependence on $H$ and $S$. According to Table 1, it outperforms existing algorithms as long as $\min\{A, B\} \ll H^2$. Compared with Nash-VI with the same sample complexity $\widetilde{O}(H^4 SAB/\varepsilon^2)$, Nash-VI requires $T$ to be larger than $\widetilde{O}(S^3 ABH^4)$ to attain the above sample complexity, which is generally significantly larger than that of ME-Nash-QL as $S > H^3$. These conditions are common and are satisfied in many scenarios. For example, in Go, there is $150 \leq H \leq 722$, $S = 2^{361}$ and $\min\{A, B\} = 360$ (Silver et al., 2017). Similar examples include Atari games and Poker.

- Unlike the state-of-the-art model-free algorithms such as Nash V/Q-Learning, our algorithm outputs both the single Markov policy and Nash policy (instead of a nested mixture of Markov policies as returned by Nash V-Learning). Overall, Table 1 shows that our algorithm *outperforms all previous algorithms with a Markov and Nash policy in terms of space, sample, computational complexity, and burn-in cost*. We design an extended algorithm of ME-Nash-QL for multi-player general-sum Markov games and achieve an $\epsilon$-optimal policy in $\widetilde{O}(H^4 S \prod_{i \in [M]} A_i/\epsilon^2)$ samples with $M$ players and $A_i$ actions per player.

## 1.2 RELATED WORK

**Markov games**  Markov games (MGs), also known as stochastic games, were introduced in the early 1950s (Shapley, 1953). Since then, numerous studies have been conducted, with a particular focus on Nash equilibrium (Littman, 1994; Lee et al., 2020). However, these studies are based on two strong assumptions. First, transition and rewards are generally assumed to be known and partly based on the asymptotic setting with an infinite amount of data. Under the curse of dimensionality, the non-asymptotic setting is an important component of relevant research, and the transition and reward should be estimated under a limited amount of data. Second, some work makes strong reachability assumptions and fails to consider the impact of exploration strategies. The agent can sample transition and rewards for any state-action pair based on the assumption of accessing simulators (generative models). For example, (Jia et al., 2019; Sidford et al., Aug. 2020; Zhang et al., 2020a) derive non-asymptotic bounds for achieving $\varepsilon$-approximate Nash equilibrium based on the number of visits to the simulator. Especially, (Wei et al., 2017) studies MGs assuming that one agent can always reach all states using a certain policy under the compliance of the other agent with any policy.

**Non-asymptotic guarantees without reachability assumptions**  As the milestone for MGs, (Bai & Jin, 2020) and (Xie et al., Jun. 2022) firstly provide non-asymptotic sample complexity guarantees for model-based algorithms (e.g., VI-Explore and VI-ULCB) and model-free algorithms (e.g., OMNI-VI), respectively. These are investigated further by the Nash-VI and Nash Q/V-Learning and provide a better sample complexity guarantee. Nash V-learning achieves sample complexity that has optimal dependence on $S$, $A$, and $B$, but the dependence on $H$ is worse than ours. Nash-VI has the same complexity as ours, but it requires a space complexity $O(S^2 ABH)$. Besides, Nash-UCRL obtains sample complexity with near-optimal dependence on $H$. Optimistic PO gets $\sqrt{T}$-regret. PReBO has $O(SABHT)$ space complexity for the storage of historical policies. OMNI-VI has $O(S^2 A^2 B^2 T)$ space complexity. However, neither of them achieves optimal dependence on $S$, $A$, or $B$. The detailed comparison is shown in Table 1. During the preparation of our work, we observed that (Feng et al., 2023) also employs Coarse Correlated Equilibrium (CCE) and reference-advantage decomposition technique, with a higher burn-in cost and outputting policies neither Nash nor Markov .

**Multi-player general-sum Markov games** (Liu et al., July 2021) developed a model-based algorithm with sample complexity $\widetilde{O}(H^5 S^2 \prod_{i \in [M]} A_i / \epsilon^2)$, which suffers from the curse of multi-agent. To alleviate this issue, (Song et al., 2021; Mao et al., 2022; Mao & Başar, 2023; Cui et al., 2023; Wang et al., 2023) proposed V-learning algorithms, coupled with the adversarial bandit subroutine, to break the curse of multi-agent. Among them, the best sample complexity is $\widetilde{O}(H^6 S \prod_{i \in [M]} A_i / \epsilon^2)$ achieved by (Song et al., 2021). In addition, (Daskalakis et al., 2023) with sample complexity $\widetilde{O}(H^{11} S^3 \prod_{i \in [M]} A_i / \epsilon^3)$ learned an approximate CCE that is guaranteed to be Markov.

**Single-agent RL** There is a rich literature on reinforcement learning in MDPs (see e.g., Jaksch et al., April 2010; Osband et al., 2016; Azar et al., 2017; Dann et al., 2017; Strehl et al., 2006; Jin et al., 2018a; Li et al., Dec. 2022). MDPs are special cases of Markov games, where only a single agent interacts with a stochastic environment. For the tabular episodic setting with non-stationary dynamics and no simulators, regret and sample analysis are the commonly adopted analytical paradigm for the trade-off between exploration and exploitation. Notably, the lower bound of the regret is $\sqrt{H^2 SAT}$, corresponding to the sample complexity $\widetilde{O}(H^4 SA / \varepsilon^2)$, which has been achieved by the model-based algorithm in (Azar et al., 2017) and the model-free algorithm in (Li et al., Dec. 2022).

## 2 PROBLEM FORMULATION

In this paper, we consider the tabular episodic version of TZMG problem. Specifically, we denote a finite-horizon TZMG as $\mathcal{M} = (\mathcal{S}, \mathcal{A}, \mathcal{B}, H, \{P_h\}_{h=1}^H, \{r_h\}_{h=1}^H)$. Here $\mathcal{S} := \{1, \cdots, S\}$ is the state space of size $S$, $(\mathcal{A} := \{1, \cdots, A\}, \mathcal{B} := \{1, \cdots, B\})$ denote the action spaces of the max-player and the min-player with size $A$ and $B$, respectively, $H$ is the horizon length, and $P_h := \mathcal{S} \times \mathcal{A} \times \mathcal{B} \to \Delta(\mathcal{S})$(resp. $r_h := \mathcal{S} \times \mathcal{A} \times \mathcal{B} \to [0,1]$) represents the probability transition kernel (resp. reward function) at the $h$-th time step, $1 \le h \le H$. Moreover, $P_h(\cdot \mid s, a, b) \in \Delta(S)$ stands for the transition probability vector from state $s$ at time step $h$ when action pair $(a, b)$ is taken, while $r_h(s, a, b)$ indicates the immediate reward received at time step $h$ on a state-action pair $(s, a, b)$ (which is assumed to be deterministic and falls within the range $[0, 1]$). For TZMG, this reward can represent both the gain of the max-player and the loss of the min-player.

A Markov policy of the max-player is represented by $\mu = \{\mu_h\}_{h=1}^H$, where $\mu_h := \mathcal{S} \to \Delta_{\mathcal{A}}$ is the action selection rule at time step $h$. Similarly, a Markov policy of the min-player is defined as $\nu = \{\nu_h\}_{h=1}^H$ with $\nu_h := \mathcal{S} \to \Delta_{\mathcal{B}}$. Each player executes the MDP sequentially in a total number of $K$ episodes, leading to $T = KH$ samples collected in total. Moreover, in each episode $k = 1, \ldots, K$, we start with an arbitrary initial state $s_1^k$, and both players implement their own policy $\mu^k = \{\mu_h^k\}_{h=1}^H$ and $\nu^k = \{\nu_h^k\}_{h=1}^H$ learned based on the information up to the $(k-1)$-th episode.

**Value function.** $V_h^{\mu,\nu}(s) : \mathcal{S} \to \mathbb{R}$ is denoted as the value function and gives the expected cumulative rewards received starting from state $s$ at step $h$ under policy $\mu$ and $\nu$:

$$V_h^{\mu,\nu}(s) := \mathbb{E}_{\mu,\nu}\left[\sum_{h'=h}^H r_{h'}(s_{h'}, a_{h'}, b_{h'}) \,\middle|\, s_h = s\right], \tag{1}$$

where the expectation is taken over the randomness of the MDP trajectory $\{s_t \mid h \le t \le H\}$. We also define $Q_h^{\mu,\nu} : \mathcal{S} \times \mathcal{A} \times \mathcal{B} \to \mathbb{R}$ to be the action-value function (a.k.a the $Q$ function). $Q_h^{\mu,\nu}(s, a, b)$ gives the cumulative rewards under policy $\mu$ and $\nu$, starting from $(s, a, b)$ at step $h$:

$$Q_h^{\mu,\nu}(s, a, b) := \mathbb{E}_{\mu,\nu}\left[\sum_{h'=h}^H r_{h'}(s_{h'}, a_{h'}, b_{h'}) \,\middle|\, s_h = s, a_h = a, b_h = b\right]. \tag{2}$$

We define $V_{H+1}^{\mu,\nu}(s) = Q_{H+1}^{\mu,\nu}(s, a, b) = 0$ for any $\mu$ and $\nu$ and $(s, a, b) \in \mathcal{S} \times \mathcal{A} \times \mathcal{B}$. Moreover, we define $(\mathbb{D}_\pi Q)(s) := \mathbb{E}_{(a,b) \sim \pi(\cdot, \cdot | s)} Q(s, a, b)$ for any $Q$ function. Thus, we have the Bellman equation

$$Q_h^{\mu,\nu}(s, a, b) = r_h(s, a, b) + \mathbb{E}_{s' \sim P_h(\cdot | s, a, b)}\left[V_{h+1}^{\mu,\nu}(s')\right], \quad V_h^{\mu,\nu}(s) = (\mathbb{D}_{\mu_h, \nu_h} Q_h^{\mu,\nu})(s). \tag{3}$$

**Best response and Nash equilibrium.** For any policy of the max-player $\mu$, there exists the best response of the min-player, which is a policy $\nu^\dagger(\mu)$ satisfying $V_h^{\mu, \nu^\dagger(\mu)}(s) = \inf_\nu V_h^{\mu,\nu}(s)$ for any $(s, h) \in \mathcal{S} \times [H]$. For simplicity, we denote $V^{\mu,\dagger} := V_h^{\mu, \nu^\dagger(\mu)}$. By symmetry, we define $\mu^\dagger(\nu)$ and

$V_h^{\dagger,\nu}$ in a similar way. It is further known (cf. Filar & Vrieze, 2012) that there exist policies $\mu^\star$ and $\nu^\star$ that are optimal against the best responses of the opponents, in the sense that

$$V_h^{\mu^\star,\dagger}(s) = \sup_\mu V_h^{\mu,\dagger}(s), \quad V_h^{\dagger,\nu^\star}(s) = \inf_\nu V_h^{\dagger,\nu}(s), \quad \text{for all } (s,h). \tag{4}$$

These optimal strategies $(\mu^\star, \nu^\star)$ are the Nash equilibrium of the Markov game satisfying:

$$\sup_\mu \inf_\nu V_h^{\mu,\nu}(s) = V_h^{\mu^\star,\nu^\star}(s) = \inf_\nu \sup_\mu V_h^{\mu,\nu}(s). \tag{5}$$

A Nash equilibrium gives a solution in which no player has anything to gain by changing only its own policy. We denote the values of Nash equilibrium $V_h^{\mu^\star,\nu^\star}$ and $Q_h^{\mu^\star,\nu^\star}$ as $V_h^\star$ and $Q_h^\star$, respectively.

**Learning objective.** We use the gap between the max-player and the min-player under the optimal strategy (i.e., NE) as the learning objective, which can be expressed as $V_1^{\dagger,\hat\nu}\left(s_1^k\right) - V_1^{\hat\mu,\dagger}\left(s_1^k\right)$. Our goal is to design an algorithm that can find an $\varepsilon$-approximate NE using several episodes under the space complexity $SABH$ and computational time $T\text{poly}(AB)$ with output policies of two players that are independent of past and independent of each other (i.e., Markov and Nash policy), and achieve regret that is sublinear in $T = KH$ and polynomial in $S$, $A$, $B$, $H$ (regret bound).

**Definition 1 ($\varepsilon$-approximate Nash equilibrium)** *If* $\frac{1}{K}\sum_{k=1}^K \left(V_1^{\dagger,\hat\nu}(s_1^k) - V_1^{\hat\mu,\dagger}(s_1^k)\right) \le \varepsilon$, *then the pair of policies $(\hat\mu, \hat\nu)$ is an $\varepsilon$-approximate Nash equilibrium.*

**Definition 2 (Regret)** *Let $(\mu^k, \nu^k)$ denote the policies deployed by the algorithm in the $k$-th episode. After a total of $K$ episodes, the regret is defined as*

$$\text{Regret}(K) = \sum_{k=1}^K \left(V_1^{\dagger,\nu^k} - V_1^{\mu^k,\dagger}\right)(s_1^k). \tag{6}$$

Notably, sample complexity $T$ refers to the required number of samples to achieve $\frac{1}{K}\text{Regret}(K) \le \varepsilon$.

**Notation.** Before presenting our main results, we introduce some convenient notations used throughout this paper. For any vector $x \in \mathbb{R}^{SAB}$ that constitutes certain quantities for all state-action pairs, we use $x(s,a,b)$ to denote the entry associated with the state-action pair $(s,a,b)$. We shall also let

$$P_{h,s,a,b} = P_h(\cdot|s,a,b) \in \mathbb{R}^{1\times S} \tag{7}$$

abbreviate the transition probability vector given the $(s,a,b)$ pair at time step $h$. Additionally, $e_i$ is denoted as the $i$-th standard basis vector with the $i$-th entry equal to 1 and others are all 0.

## 3 ALGORITHM AND THEORETICAL GUARANTEES

In this section, we present the proposed algorithm called ME-Nash-QL and provide its theoretical guarantee of memory and sample efficiency.

### 3.1 ALGORITHM DESCRIPTION

We propose a model-free algorithm ME-Nash-QL described in Algorithm 1, which is the first to integrate the reference-advantage decomposition technique to TZMG along with an innovative early-settlement approach. For the $n$-th visit of a state-action pair at any time step $h$, the proposed algorithm adopts the linearly rescaled learning rate as $\eta_n = \frac{H+1}{H+n}$. In each episode, the algorithm can be decomposed into two parts, i.e., policy evaluation and improvement, which are standard in the majority of model-free algorithms.

- Lines 4-12 of Algorithm 1 (Policy evaluation): Select actions based on the current policy to obtain the next state and the reward information. The sampled data is then used to estimate several types of $Q$-function: $Q_h^{\text{UCB}}$ and $Q_h^{\text{LCB}}$ with an exploration bonus, $\overline{Q}_h^{\text{R}}$ and $\underline{Q}_h^{\text{R}}$ using the reference-advantage decomposition, and $\overline{Q}_h$ and $\underline{Q}_h$, which are obtained by combining the above estimations.
- Lines 13-19 of Algorithm 1 (Policy improvement): Compute a new (joint) policy $\pi_h$ using the estimated value functions, update those value functions, and perform updates of reference values under the early settlement.

$Q$**-function estimation**  In each episode, the algorithm is designed to maintain estimates of the $Q$-function with low sample complexity that provide optimistic and pessimistic views (namely, over-estimate and under-estimate) of the truth $Q^\star$. The algorithm aims to reduce the bias $\overline{Q} - Q^\star$ and $\underline{Q} - Q^\star$ in two ways. Firstly, we migrate the Q-learning algorithm with the upper-confidence bound (UCB) exploration strategy proposed in (Jin et al., 2018a) to Q-learning with UCB/lower-confidence bound (LCB) exploration strategy. This is expressed as the subroutine update-q() with bonus $\iota_n = c_b\sqrt{\frac{1}{n}H^3\log\frac{SABT}{\delta}}$ in Algorithm 2. Secondly, we leverage the idea of variance reduction to shave the $H$ factor of sample complexity compared to Nash Q-learning based on reference-advantage decomposition (Zhang et al., 2020b) in single-agent scenarios. Taking UCB as an example, we adopt the update rule at each time step $h$ as

$$\overline{Q}_h^R(s,a,b) \leftarrow (1-\eta)\overline{Q}_h^R(s,a,b) + \eta\left\{r_h(s,a) + \widehat{P}_{h,s,a,b}\left(\overline{V}_{h+1} - \overline{V}_{h+1}^R\right) + \big[\widehat{P_h\overline{V}_{h+1}^R}\big](s,a,b) + \overline{b}_h^R\right\}, \quad (8)$$

where $\overline{b}_h^R$ is the exploration bonus, $\overline{V}_{h+1}^R$ is the reference value introduced next, and $\widehat{P}_{h,s,a,b}\left(\overline{V}_{h+1} - \overline{V}_{h+1}^R\right)$ is the stochastic estimate of $\overline{V}_{h+1}(s') - \overline{V}_{h+1}^R(s')$ with $\widehat{P}_{h,s,a,b}$ as the estimate of $P_{h,s,a,b}$. Besides, the term $\widehat{P_h\overline{V}_{h+1}^R}$ indicates an estimate of the one-step look-ahead value $P_h\overline{V}_{h+1}^R$, which can be computed via the batch data like line 12 in Algorithm 2. Accordingly, we combine the UCB for $P_{h,s,a}\left(V_{h+1} - \overline{V}_{h+1}^R\right)$ and $P_h\overline{V}_{h+1}^R$ together as the exploration bonus term $\overline{b}_h^R$.

Thus, based on $Q^{\mathrm{UCB}}$, $Q^{\mathrm{LCB}}$, $\overline{Q}^R$, and $\underline{Q}^R$ we can combine them as line 11-12 in Algorithm 1 to further reduce the bias without violating the optimism or pessimism principle of $Q$-function estimate.

---

**Algorithm 1:** Memory-Efficient Nash Q-Learning (ME-Nash-QL)

1 **Parameter:** some universal constant $c_b > 0$ and probability of failure $\delta \in (0, 1)$

2 **Initialize:** $\overline{Q}_h(s,a,b)$, $Q_h^{\mathrm{UCB}}(s,a,b)$, $\overline{Q}_h^R(s,a,b)$, $\leftarrow H$; $\underline{Q}_h(s,a,b)$, $\underline{Q}_h^R(s,a,b)$,

   $Q_h^{\mathrm{LCB}}(s,a,b) \leftarrow 0$; $\overline{V}_h(s)$, $\overline{V}_h^R(s) \leftarrow H$; $\underline{V}_h(s)$, $\underline{V}_h^R(s) \leftarrow 0$; $N_h(s,a,b) \leftarrow 0$; $\overline{\phi}_h^r(s,a,b)$,
   $\underline{\phi}_h^r(s,a,b)$, $\overline{\psi}_h^r(s,a,b)$, $\underline{\psi}_h^r(s,a,b)$, $\overline{\phi}_h^a(s,a,b)$, $\underline{\phi}_h^a(s,a,b)$, $\overline{\psi}_h^a(s,a,b)$, $\underline{\psi}_h^a(s,a,b)$, $\overline{\varphi}_h^R(s,a,b)$,
   $\underline{\varphi}_h^R(s,a,b)$, $\overline{B}_h^R(s,a,b)$, $\underline{B}_h^R(s,a,b) \leftarrow 0$; and $u_r(s) = \mathsf{True}$ for all
   $(s,a,b,h) \in \mathcal{S} \times \mathcal{A} \times \mathcal{B} \times [H]$.

3 **for** *Episode* $k = 1, \dots, K$ **do**

4 $\quad$ Set initial state $s_1 \leftarrow s_1^k$.

5 $\quad$ **for** *Step* $h = 1, \dots, H$ **do**

6 $\quad\quad$ Take action $(a_h, b_h) \sim \pi_h(\cdot, \cdot | s_h)$, and draw $s_{h+1} \sim P_h(\cdot \mid s_h, a_h, b_h)$.

7 $\quad\quad$ $N_h(s_h, a_h, b_h) \leftarrow N_h(s_h, a_h, b_h) + 1$; $\quad n \leftarrow N_h(s_h, a_h, b_h)$; $\quad \eta_n \leftarrow \frac{H+1}{H+n}$.

8 $\quad\quad$ $\big[Q_h^{\mathrm{UCB}}, Q_h^{\mathrm{LCB}}\big](s_h, a_h, b_h) \leftarrow \text{update-q}()$.

9 $\quad\quad$ $\overline{Q}_h^R(s_h, a_h, b_h) \leftarrow \text{update-ur}()$.

10 $\quad\quad$ $\underline{Q}_h^R(s_h, a_h, b_h) \leftarrow \text{update-lr}()$.

11 $\quad\quad$ $\overline{Q}_h(s_h, a_h, b_h) \leftarrow \min\{\overline{Q}_h^R(s_h, a_h, b_h), Q_h^{\mathrm{UCB}}(s_h, a_h, b_h), \overline{Q}_h(s_h, a_h, b_h)\}$.

12 $\quad\quad$ $\underline{Q}_h(s_h, a_h, b_h) \leftarrow \max\{\underline{Q}_h^R(s_h, a_h, b_h), Q_h^{\mathrm{LCB}}(s_h, a_h, b_h), \underline{Q}_h(s_h, a_h, b_h)\}$.

13 $\quad\quad$ **if** $\overline{Q}_h(s_h, a_h, b_h) = \min\{\overline{Q}_h^R(s_h, a_h, b_h), Q_h^{\mathrm{UCB}}(s_h, a_h, b_h)\}$ *and*
   $\quad\quad\quad \underline{Q}_h(s_h, a_h, b_h) = \max\{\underline{Q}_h^R(s_h, a_h, b_h), Q_h^{\mathrm{LCB}}(s_h, a_h, b_h)\}$ **then**

14 $\quad\quad\quad$ $\pi_h(\cdot, \cdot | s_h) \leftarrow \mathrm{CCE}(\overline{Q}_h(s_h, \cdot, \cdot), \underline{Q}_h(s_h, \cdot, \cdot))$.

15 $\quad\quad$ $\overline{V}_h(s_h) \leftarrow \min\{(\mathbb{D}_{\pi_h}\overline{Q}_h)(s_h), \overline{V}_h(s_h)\}$; $\quad \underline{V}_h(s_h) \leftarrow \max\{(\mathbb{D}_{\pi_h}\underline{Q}_h)(s_h), \underline{V}_h(s_h)\}$.

16 $\quad\quad$ **if** $\overline{V}_h(s_h) - \underline{V}_h(s_h) > 1$ **then**

17 $\quad\quad\quad$ $\overline{V}_h^R(s_h) \leftarrow \overline{V}_h(s_h)$; $\quad \underline{V}_h^R(s_h) \leftarrow \underline{V}_h(s_h)$.

18 $\quad\quad$ **else if** $u_r(s_h) = \mathsf{True}$ **then**

19 $\quad\quad\quad$ $\overline{V}_h^R(s_h) \leftarrow \overline{V}_h(s_h)$; $\quad \underline{V}_h^R(s_h) \leftarrow \underline{V}_h(s_h)$; $\quad u_r(s_h) = \mathsf{False}$.

**Output:** The marginal policies of $\{\pi_h\}_{h=1}^H$: $\left(\{\mu_h\}_{h=1}^H, \{\nu_h\}_{h=1}^H\right)$.

---

---

**Algorithm 2:** Auxiliary functions

---

1 **Function** update-q($\left[Q_h^{\mathrm{UCB}}, Q_h^{\mathrm{LCB}}\right](s_h, a_h, b_h), \left[\overline{V}_{h+1}, \underline{V}_{h+1}\right](s_{h+1})$)**:**

2      $Q_h^{\mathrm{UCB}}(s_h, a_h, b_h) \leftarrow (1-\eta_n) Q_h^{\mathrm{UCB}}(s_h, a_h, b_h) + \eta_n \left(r_h(s_h, a_h, b_h) + \overline{V}_{h+1}(s_{h+1}) + \iota_n\right);$

3      $Q_h^{\mathrm{LCB}}(s_h, a_h, b_h) \leftarrow (1-\eta_n) Q_h^{\mathrm{LCB}}(s_h, a_h, b_h) + \eta_n \left(r_h(s_h, a_h, b_h) + \underline{V}_{h+1}(s_{h+1}) - \iota_n\right).$

4 **Function** update-ur($\left[\overline{\phi}_h^{\mathrm{r}}, \overline{\psi}_h^{\mathrm{r}}, \overline{\phi}_h^{\mathrm{a}}, \overline{\psi}_h^{\mathrm{a}}, \overline{B}_h^{\mathrm{R}}, \overline{Q}_h^{\mathrm{R}}\right](s_h, a_h, b_h), \left[\overline{V}_{h+1}^{\mathrm{R}}, \overline{V}_{h+1}\right](s_{h+1})$)**:**

5      $\left[\overline{\phi}_h^{\mathrm{r}}(s_h, a_h, b_h), \overline{b}_h^{\mathrm{R}}\right] \leftarrow$ update-q-bonus$\left(\left[\overline{\phi}_h^{\mathrm{r}}, \overline{\psi}_h^{\mathrm{r}}, \overline{\phi}_h^{\mathrm{a}}, \overline{\psi}_h^{\mathrm{a}}, \overline{B}_h^{\mathrm{R}}\right](s_h, a_h, b_h), \left[\overline{V}_{h+1}^{\mathrm{R}}, \overline{V}_{h+1}\right](s_{h+1})\right);$

6      $\overline{Q}_h^{\mathrm{R}}(s_h, a_h, b_h) \leftarrow (1 - \eta_n) \overline{Q}_h^{\mathrm{R}}(s_h, a_h, b_h) +$
         $\eta_n \left(r_h(s_h, a_h, b_h) + \overline{V}_{h+1}(s_{h+1}) - \overline{V}_{h+1}^{\mathrm{R}}(s_{h+1}) + \overline{\phi}_h^{\mathrm{r}}(s_h, a_h, b_h) + \overline{b}_h^{\mathrm{R}}\right).$

7 **Function** update-lr($\left[\underline{\phi}_h^{\mathrm{r}}, \underline{\psi}_h^{\mathrm{r}}, \underline{\phi}_h^{\mathrm{a}}, \underline{\psi}_h^{\mathrm{a}}, \underline{B}_h^{\mathrm{R}}, \underline{Q}_h^{\mathrm{R}}\right](s_h, a_h, b_h), \left[\underline{V}_{h+1}^{\mathrm{R}}, \underline{V}_{h+1}\right](s_{h+1})$)**:**

8      $\left[\underline{\phi}_h^{\mathrm{r}}(s_h, a_h, b_h), \underline{b}_h^{\mathrm{R}}\right] \leftarrow$ update-q-bonus$\left(\left[\underline{\phi}_h^{\mathrm{r}}, \underline{\psi}_h^{\mathrm{r}}, \underline{\phi}_h^{\mathrm{a}}, \underline{\psi}_h^{\mathrm{a}}, \underline{B}_h^{\mathrm{R}}\right](s_h, a_h, b_h), \left[\underline{V}_{h+1}^{\mathrm{R}}, \underline{V}_{h+1}\right](s_{h+1})\right);$

9      $\underline{Q}_h^{\mathrm{R}}(s_h, a_h, b_h) \leftarrow (1 - \eta_n) \underline{Q}_h^{\mathrm{R}}(s_h, a_h, b_h) +$
         $\eta_n \left(r_h(s_h, a_h, b_h) + \underline{V}_{h+1}(s_{h+1}) - \underline{V}_{h+1}^{\mathrm{R}}(s_{h+1}) + \underline{\phi}_h^{\mathrm{r}}(s_h, a_h, b_h) - \underline{b}_h^{\mathrm{R}}\right).$

10 **Function** update-q-bonus($\left[\phi_h^{\mathrm{r}}, \psi_h^{\mathrm{r}}, \phi_h^{\mathrm{a}}, \psi_h^{\mathrm{a}}, B_h^{\mathrm{R}}\right](s_h, a_h, b_h), \left[V_{h+1}^{\mathrm{R}}, V_{h+1}\right](s_{h+1})$)**:**

11      $\phi_h^{\mathrm{r}}(s_h, a_h, b_h) \leftarrow \left(1 - \frac{1}{n}\right) \phi_h^{\mathrm{r}}(s_h, a_h, b_h) + \frac{1}{n} V_{h+1}^{\mathrm{R}}(s_{h+1});$

12      $\psi_h^{\mathrm{r}}(s_h, a_h, b_h) \leftarrow \left(1 - \frac{1}{n}\right) \psi_h^{\mathrm{r}}(s_h, a_h, b_h) + \frac{1}{n} \left(V_{h+1}^{\mathrm{R}}(s_{h+1})\right)^2;$

13      $\phi_h^{\mathrm{a}}(s_h, a_h, b_h) \leftarrow (1 - \eta_n) \phi_h^{\mathrm{a}}(s_h, a_h, b_h) + \eta_n \left(V_{h+1}(s_{h+1}) - V_{h+1}^{\mathrm{R}}(s_{h+1})\right);$

14      $\psi_h^{\mathrm{a}}(s_h, a_h, b_h) \leftarrow (1 - \eta_n) \psi_h^{\mathrm{a}}(s_h, a_h, b_h) + \eta_n \left(V_{h+1}(s_{h+1}) - V_{h+1}^{\mathrm{R}}(s_{h+1})\right)^2;$

15      $B_h^{\mathrm{temp}}(s_h, a_h, b_h) \leftarrow c_{\mathrm{b}} \sqrt{\frac{\log^2 \frac{SABT}{\delta}}{n}} \sqrt{\psi_h^{\mathrm{r}}(s_h, a_h, b_h) - \left(\phi_h^{\mathrm{r}}(s_h, a_h, b_h)\right)^2} +$
         $c_{\mathrm{b}} \sqrt{\frac{\log^2 \frac{SABT}{\delta}}{n}} \sqrt{H} \sqrt{\psi_h^{\mathrm{a}}(s_h, a_h, b_h) - \left(\phi_h^{\mathrm{a}}(s_h, a_h, b_h)\right)^2};$

16      $\varphi_h^{\mathrm{R}}(s_h, a_h, b_h) \leftarrow B_h^{\mathrm{temp}}(s_h, a_h, b_h) - B_h^{\mathrm{R}}(s_h, a_h, b_h);$

17      $B_h^{\mathrm{R}}(s_h, a_h, b_h) \leftarrow B_h^{\mathrm{temp}}(s_h, a_h, b_h)$

18      $b_h^{\mathrm{R}} \leftarrow B_h^{\mathrm{R}}(s_h, a_h, b_h) + (1 - \eta_n) \frac{\varphi_h^{\mathrm{R}}(s_h, a_h, b_h)}{\eta_n} + c_{\mathrm{b}} \frac{H^2 \log^2 \frac{SABT}{\delta}}{n^{3/4}};$

---

**Reference values update**    Motivated by (Li et al., Dec. 2022), we implement the following appropriate termination rules to allow for the early settlement of the reference values $\overline{V}_h^{\mathrm{R}}$ and $\underline{V}_h^{\mathrm{R}}$:

$$\overline{V}_h(s_h) - \underline{V}_h(s_h) \leq 1, \tag{9}$$

which is displayed in lines 16-19 of Algorithm 1. Notably, due to references and the early settlement, our algorithm obtains a lower sample complexity and burn-in cost than Nash Q-learning.

- Weaken the dependency of the sample complexity on $H$: the uncertainty of the update rule largely stems from the third and fourth terms on the right-hand side of (8). The reference value $\overline{V}_h^{\mathrm{R}}$ (resp. $\underline{V}_h^{\mathrm{R}}$) stays reasonably close to $\overline{V}_h$ (resp. $\underline{V}_h$), which suggests that the standard deviation of the third term is small. For the fourth term, the reference value is fixed and never changes after satisfying condition (lines 16-19 in Algorithm 1) for the first time, and we can use all the samples collected to estimate $P_h \overline{V}_{h+1}^{\mathrm{R}}$. Therefore, both of these two terms have much smaller variances than that of usual UCB/LCB value functions due to the incorporation of reference terms, which enables the algorithm to estimate them with high accuracy with a limited number of samples.

- Reduce the burn-in cost: reference value $\overline{V}_h^{\mathrm{R}}$ (resp. $\underline{V}_h^{\mathrm{R}}$) is used to keep tracking the value of $\overline{V}_h$ (resp. $\underline{V}_h$) before it stops being updated. Based on the early-settlement method, the reference values will stop being updated shortly after the condition (lines 16-19 in Algorithm 1) is met for the first time. As a result, the algorithm has the ability to quickly settle on a desirable "reference" during the initial stage. Moreover, the aggregate difference between $\overline{V}_h^{\mathrm{R},k}$ (resp. $\underline{V}_h^{\mathrm{R},k}$) and the final reference $\overline{V}_h^{\mathrm{R},K}$ (resp. $\underline{V}_h^{\mathrm{R},K}$) over the entire trajectory can be bounded in a reasonably tight fashion.

**Policy computation**  We apply a relaxation of the Nash equilibrium—*Coarse Correlated Equilibrium (CCE)* to output a single Markov policy rather than a nested mixture of Markov policies, which is introduced by (Moulin & Vial, Sep. 1978) and developed by (Xie et al., Jun. 2022). Specifically, for any pair of matrices $\overline{Q}, \underline{Q} \in [0, H]^{A \times B}$, $\text{CCE}(\overline{Q}, \underline{Q})$ returns a distribution $\pi \in \Delta_{\mathcal{A} \times \mathcal{B}}$ such that

$$\mathbb{E}_{(a,b)\sim\pi}\overline{Q}(s,a,b) \geq \mathbb{E}_{(a,b)\sim\pi}\overline{Q}(s,a',b) \quad \forall a', \tag{10a}$$

$$\mathbb{E}_{(a,b)\sim\pi}\underline{Q}(s,a,b) \leq \mathbb{E}_{(a,b)\sim\pi}\underline{Q}(s,a,b') \quad \forall b'. \tag{10b}$$

Intuitively, no one can benefit from unilateral unconditional deviation in a CCE since the players' action strategies have an underlying correlation, unlike in a Nash equilibrium where each player's strategy is independent. Another advantage of a CCE is its ability to obtain results in polynomial time through linear programming. Moreover, since a Nash equilibrium always exists, and a Nash equilibrium is also a CCE, we can conclude that a CCE always exists as well. We would like to recommend readers interested in the detailed computation of CCE to refer to (Xie et al., Jun. 2022).

Specifically, line 14 is used for computing our output policies. These final policies $(\mu, \nu)$ are simply the *marginal policies* of $\pi_h$. That is, for any given $(s, h) \in \mathcal{S} \times [H]$, we have $\mu_h(\cdot|s) := \sum_{b \in \mathcal{B}} \pi_h(\cdot, b|s)$ and $\nu_h(\cdot|s) := \sum_{a \in \mathcal{A}} \pi_h(a, \cdot|s)$. In contrast to previous algorithms that require space complexity dependent on $T$ to generate a generic history-dependent policy, which can be only written as a nested mixture of Markov policies, our algorithm has the ability to produce a single Markov policy with space complexity $O(SABH)$.

## 3.2 MAIN RESULTS

We begin by proving the theoretical guarantee of ME-Nash-QL by the following theorem.

**Theorem 1** *Consider any $\delta \in (0, 1)$, and suppose that $c_b > 0$ is chosen to be a sufficiently large universal constant. Then there exists some absolute constant $C_0 > 0$ such that Algorithm 1 achieves*

$$\frac{1}{K} \sum_{k=1}^{K} \left( V_1^{\dagger, \nu^k}(s_1^k) - V_1^{\mu^k, \dagger}(s_1^k) \right) \leq \varepsilon \tag{11}$$

*if the number of samples $T$ satisfies*

$$T \geq C_0 \left( \frac{H^4 SAB}{\varepsilon^2} \log^4 \frac{SABT}{\delta} + \frac{H^7 SAB}{\varepsilon} \log^3 \frac{SAB}{\delta} \right) \tag{12}$$

*with probability at least $1 - \delta$.*

**Sample complexity and burn-in cost**  The sample complexity (12) can be simplified as

$$\widetilde{O}\left( H^4 SAB/\varepsilon^2 \right) \tag{13}$$

for sufficiently large $T$. Moreover, our algorithm achieves the best burn-in cost $\widetilde{O}(SABH^{10})$, which is the minimum sample size to guarantee that the sample complexity of the algorithm is near optimal. This corresponds to $\text{Regret}(K) \leq \widetilde{O}(H^2 SABT)$ as Section 1.1. And the state-of-the-art self-play algorithm has a burn-in cost of at least $\widetilde{O}(S^3 ABH^4)$ to attain the $\widetilde{O}(H^4 S/\varepsilon^2)$-sample complexity.

Theorem 1 guarantees that the average Nash gap is smaller than $\varepsilon$ with sample complexity in (13). For supplementary, the following theorem provides the theoretical guarantee of an actual output policy.

**Theorem 2** *Consider the policy $\pi^{k^\star} = (\mu^{k^\star}, \nu^{k^\star})$ with $k^\star = \arg\min_k \left( \overline{V}_1^k(s_1^k) - \underline{V}_1^k(s_1^k) \right)$. If we take $\pi^{k^\star}$ as the output marginal policy, for any $\delta \in (0, 1)$, with probability at least $1 - \delta$, there is $\overline{V}_1^{\dagger, \nu^{k^\star}}(s_1^{k^\star}) - \underline{V}_1^{\mu^{k^\star}, \dagger}(s_1^{k^\star}) \leq \overline{V}_1^{k^\star}(s_1^{k^\star}) - \underline{V}_1^{k^\star}(s_1^{k^\star}) \leq \frac{1}{K}\widetilde{O}\left( \sqrt{H^2 SABT} \right).$*

Besides, our algorithm can be extended to multi-player general-sum Markov games with $m$ players and $A_i$ actions per player with details in Appendix F, called Multi-ME-Nash-QL. We can obtain the following theoretical guarantee of Multi-ME-Nash-QL as

**Theorem 3** *Consider any $\delta \in (0, 1)$, and suppose that $c_b > 0$ is chosen to be a sufficiently large universal constant. Then there exists some absolute constant $C_0 > 0$ such that Multi-ME-Nash-QL achieves sample complexity $\widetilde{O}\left( \sqrt{H^2 ST \prod_{i=1}^{m} A_i} \right)$ with probability at least $1 - \delta$.*

**Memory efficiency, computational complexity, and Markov/Nash policy** ME-Nash-QL achieves space complexity $O(SABH)$, which is essentially unimprovable in the tabular case if the $Q$-values are stored. To the best of our knowledge, this is the first time that such complexity has been achieved along with the delivery of a Markov/Nash policy. Furthermore, the computational complexity of our algorithm is $O(T\text{poly}(AB))$, which is due to the CCE computation by linear programming in polynomial time $O(\text{poly}(AB))$. In comparison, although the previous algorithm Nash-VI also achieves same sample complexity, it has a significantly larger space complexity of $O(S^2ABH)$ due to its model-based nature, along with larger computational complexity $O(T\text{poly}(SAB))$.

## 4 ANALYSIS

In this section, we outline the main steps needed to prove our main result in Theorem 1 with detailed proof in Appendix A.3.2. To simplify the presentation, we have ignored the dependency on $k$ in Algorithms 1 and 2. Next, we need to be more explicit with the following notations for completion.

(i) $(s_h^k, a_h^k, b_h^k)$ is denoted as the state-action pair encountered and chosen at step $h$ in the $k$-th episode.

(ii) $\overline{Q}_h^k(s,a,b), \underline{Q}_h^k(s,a,b), \overline{Q}_h^{\text{R},k}(s,a,b), \underline{Q}_h^{\text{R},k}(s,a,b)$ and $\overline{V}_h^k(s), \underline{V}_h^k(s)$ denote, resp., $\overline{Q}_h(s,a,b)$, $\underline{Q}_h(s,a,b), \overline{Q}_h^{\text{R}}(s,a,b), \underline{Q}_h^{\text{R}}(s,a,b)$ and $\overline{V}_h(s), \underline{V}_h(s)$ *at the beginning* of the $k$-th episode.

**Step 1: regret decomposition.** We can obtain
$$\text{Regret}(K) = \sum_{k=1}^K \left(V_1^{\dagger,\nu^k} - V_1^{\mu^k,\dagger}\right)(s_1^k) \leq \sum_{k=1}^K \left[\overline{Q}_h^{\text{R},k}(s_h^k, a_h^k, b_h^k) - \underline{Q}_h^{\text{R},k}(s_h^k, a_h^k, b_h^k) + \zeta_h^k\right],$$
with $\zeta_h^k := \mathbb{E}_{(a,b)\sim\pi_h^k}(\overline{Q}_h^k - \underline{Q}_h^k)(s_h^k, a, b) - (\overline{Q}_h^k - \underline{Q}_h^k)(s_h^k, a_h^k, b_h^k)$ (see Appendix A.3.2 for details).

**Step 2: managing regret by recursion.** The regret can be further manipulated by leveraging the update rule of $\overline{Q}_h^{\text{R},k}$ and $\underline{Q}_h^{\text{R},k}$. This leads to a key decomposition as summarized as Lemma 5 in Appendix A.3.2 and proved in Appendix D.

**Step 3: controlling the terms in Step 2 separately.** Each of the terms in Step 2 can be well controlled. We provide the bounds for these terms as Lemma 6 in Appendix E and summarize them in Appendix A.3.2. To derive the above bounds, the main strategy is to apply the Bernstein-type concentration inequalities carefully, and to upper bound the sum of variance.

**Step 4: putting all this together.** We now establish our main result. Taking the bounds in Step 3 together with Step 2, we see that with probability at least $1 - \delta$ and a constant $C_0 > 0$, one has
$$\text{Regret}(K) \leq C_0 \left(\sqrt{H^2SABT \log^4 SABT/\delta} + H^6SAB \log^3 SABT/\delta\right). \tag{14}$$

Theorem 1 is proved under sample complexity with $\varepsilon$ average regret (i.e., $\frac{1}{K}\text{Regret}(K) \leq \varepsilon$).

## 5 CONCLUSION

In this paper, we propose a novel model-free algorithm ME-Nash-QL for two-play zero-sum Markov games and provide a sharp analysis. ME-Nash-QL boasts several advantages over previous algorithms. First, to the best of our knowledge, it is the first TZMG algorithm that attains minimal space complexity $O(SABH)$. In addition, it can effectively produce an $\varepsilon$-approximate Nash equilibrium of TZMG in $\widetilde{O}(H^4SAB/\varepsilon^2)$ samples of game playing, along with the computational complexity $O(T\text{poly}(AB))$. This near-optimal sample complexity of the algorithm comes into effect as soon as the sample size exceeds $SABH^{10}$, which is the best burn-in cost compared to the previous algorithms with the same sample complexity. Further, it outputs a single Markov and Nash policy, which is a departure from previous algorithms that output nested mixture policies or non-Markov policies. There are some compelling future directions. For example, can we achieve model-free MG algorithms with $O(A + B)$ sample complexity (thus breaking the curse of multi-agent in the extension to multi-player general-sum MGs) without compromising the performance of existing metrics? How can we design independent actions with this sample complexity? We leave these problems for future work.

ACKNOWLEDGMENTS

We thank Gen Li for helpful discussions and paper feedback. The work was supported in part by the National Natural Science Foundation of China (NSFC) under Grants U21B2029 and U21A20456, in part by the China Postdoctoral Science Foundation under Grant Number 2023M730264, and in part by the Zhejiang Provincial Natural Science Foundation of China under Grant LR23F010006.

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

# A ADDITIONAL NOTATION AND KEY LEMMAS

## A.1 ADDITIONAL NOTATION

In the main text, we ignore the $k$ dependency in Algorithms 1 and 2 for simplicity. However, we rewrite Algorithm 1 with dependency on $k$ as Algorithm 3 for the following proof. In addition, we also rewrite some notations except for those introduced in the main text as below.

- $k_h^n(s, a, b)$(resp. $k_h^n(s)$): the index of the episode in which $(s, a, b)$ (resp. $s$) is visited for the $n$-th time at time step $h$; for the sake of conciseness, we shall sometimes use the shorthand $k^n = k_h^n(s, a, b)$ (resp. $k^n = k_h^n(s)$) whenever it is clear from the context.
- $P_h^k \in \{0, 1\}^{1 \times |\mathcal{S}|}$: the empirical transition at time step $h$ in the $k$-th episode, namely,

$$P_h^k(s) = \mathbb{1}(s = s_{h+1}^k). \tag{15}$$

- $N_h^k(s, a, b)$ denotes $N_h(s, a, b)$ *by the end* of the $k$-th episode; for the sake of conciseness, we shall often abbreviate $N^k = N_h^k(s, a, b)$ or $N^k = N_h^k(s_h^k, a_h^k, b_h^k)$ (depending on which result we are proving).
- $Q_h^{\text{UCB},k}(s, a, b), Q_h^{\text{LCB},k}(s, a, b), \overline{V}_h^{\text{R},k}(s)$ and $\underline{V}_h^{\text{R},k}(s)$, respectively, denote $Q_h^{\text{UCB}}(s, a, b)$, $Q_h^{\text{LCB}}(s, a, b), \overline{V}_h^{\text{R}}(s)$ and $\underline{V}_h^{\text{R}}(s)$ *at the beginning* of the $k$-th episode.
- $u_{\text{r}}^k(s)$ denotes $u_{\text{r}}(s)$ *at the beginning* of the $k$-th episode.
- $\left[\overline{\phi}_h^{\text{r},k}, \overline{\psi}_h^{\text{r},k}, \overline{\phi}_h^{\text{a},k}, \overline{\psi}_h^{\text{a},k}, \overline{\delta}_h^{\text{R},k}, \overline{B}_h^{\text{R},k}\right]$ denotes $\left[\overline{\phi}_h^{\text{r}}, \overline{\psi}_h^{\text{r}}, \overline{\phi}_h^{\text{a}}, \overline{\psi}_h^{\text{a}}, \overline{\delta}_h^{\text{R}}, \overline{B}_h^{\text{R}}\right]$ *at the beginning* of the $k$-th episode.
- $\left[\underline{\phi}_h^{\text{r},k}, \underline{\psi}_h^{\text{r},k}, \underline{\phi}_h^{\text{a},k}, \underline{\psi}_h^{\text{a},k}, \underline{\delta}_h^{\text{R},k}, \underline{B}_h^{\text{R},k}\right]$ denotes $\left[\underline{\phi}_h^{\text{r}}, \underline{\psi}_h^{\text{r}}, \underline{\phi}_h^{\text{a}}, \underline{\psi}_h^{\text{a}}, \underline{\delta}_h^{\text{R}}, \underline{B}_h^{\text{R}}\right]$ *at the beginning* of the $k$-th episode.

Furthermore, for any matrix $P = [P_{i,j}]_{1 \leq i \leq m, 1 \leq j \leq n}$, we have $\|P\|_1 := \max_{1 \leq i \leq m} \sum_{j=1}^n |P_{i,j}|$. Similarly, for any vector $V = [V_i]_{1 \leq i \leq n}$, its $\ell_\infty$ norm is defined as $\|V\|_\infty := \max_{1 \leq i \leq n} |V_i|$. We extend scalar functions and expressions to accept vector-valued inputs, with the expectation that they will be applied in an entrywise fashion. For example, for a vector $x = [x_i]_{1 \leq i \leq n}$, we denote $x^2 = [x_i^2]_{1 \leq i \leq n}$. For any two vectors $x = [x_i]_{1 \leq i \leq n}$ and $y = [y_i]_{1 \leq i \leq n}$, the notation $x \leq y$ (resp. $x \geq y$) means $x_i \leq y_i$ (resp. $x_i \geq y_i$) for all $1 \leq i \leq n$.

For any given vector $V \in \mathbb{R}^S$, we define the variance parameter w.r.t. $P_{h,s,a,b}$ (cf. (7) as follows

$$\text{Var}_{h,s,a,b}(V) := \mathbb{E}_{s' \sim P_{h,s,a,b}}\left[\left(V(s') - P_{h,s,a,b}V\right)^2\right] = P_{h,s,a,b}(V^2) - (P_{h,s,a,b}V)^2. \tag{16}$$

The notation $f(x) \lesssim g(x)$ (resp. $f(x) \gtrsim g(x)$) means that there exists a universal constant $C_0 > 0$ such that $f(x) \leq C_0 g(x)$ (resp. $f(x) \geq C_0 g(x)$). Besides, $f(x) \asymp g(x)$ represents that $f(x) \lesssim g(x)$ and $f(x) \gtrsim g(x)$ hold simultaneously.

## A.2 PRELIMINARIES: BASIC PROPERTIES ABOUT LEARNING RATES

For notation convenience, we first introduce two sequences of quantities related to the learning rate for any integer $N \geq 0$ and $n \geq 1$:

$$\eta_n^N := \begin{cases} \eta_n \prod_{i=n+1}^N (1 - \eta_i), & \text{if } N > n, \\ \eta_n, & \text{if } N = n, \quad \text{and} \quad \eta_0^N := \begin{cases} \prod_{i=1}^N (1 - \eta_i) = 0, & \text{if } N > 0, \\ 1, & \text{if } N = 0. \end{cases} \tag{17} \\ 0, & \text{if } N < n \end{cases}$$

We can obtain

$$\sum_{n=1}^N \eta_n^N = \begin{cases} 1, & \text{if } N > 0, \\ 0, & \text{if } N = 0. \end{cases} \tag{18}$$

Based on the above definitions, We introduce the following important properties before analysis.

---

**Algorithm 3:** ME-Nash-QL (a rewrite of Algorithm 1 that specifies dependency on $k$)

---

1 **Parameter:** some universal constant $c_{\mathrm{b}} > 0$ and probability of failure $\delta \in (0, 1)$;

2 **Initialize:** $\overline{Q}_h^1(s, a, b), Q_h^{\mathrm{UCB},1}(s, a, b), \overline{Q}_h^{\mathrm{R},1}(s, a, b), \leftarrow H; \underline{Q}_h^1(s, a, b), \underline{Q}_h^{\mathrm{R},1}(s, a, b),$

   $Q_h^{\mathrm{LCB},1}(s, a, b) \leftarrow 0; \overline{V}_h^1(s), \overline{V}_h^{\mathrm{R},1}(s) \leftarrow H; \underline{V}_h^1(s), \underline{V}_h^{\mathrm{R},1}(s) \leftarrow 0; N_h^0(s, a, b) \leftarrow 0; \overline{\phi}_h^{\mathrm{r}}(s, a, b),$

   $\underline{\phi}_h^{\mathrm{r}}(s, a, b), \overline{\psi}_h^{\mathrm{r}}(s, a, b), \underline{\psi}_h^{\mathrm{r}}(s, a, b), \overline{\phi}_h^{\mathrm{a}}(s, a, b), \underline{\phi}_h^{\mathrm{a}}(s, a, b), \overline{\psi}_h^{\mathrm{a}}(s, a, b), \underline{\psi}_h^{\mathrm{a}}(s, a, b), \overline{\varphi}_h^{\mathrm{R}}(s, a, b),$

   $\underline{\varphi}_h^{\mathrm{R}}(s, a, b), \overline{B}_h^{\mathrm{R}}(s, a, b), \underline{B}_h^{\mathrm{R}}(s, a, b) \leftarrow 0;$ and $u_{\mathrm{r}}^1(s) = \mathsf{True}$ for all

   $(s, a, b, h) \in \mathcal{S} \times \mathcal{A} \times \mathcal{B} \times [H]$.

3 **for** *Episode* $k = 1, \ldots, K$ **do**

4     Set initial state $s_1 \leftarrow s_1^k$.

5     **for** *Step* $h = 1, \ldots, H$ **do**

6        Take action $(a_h^k, b_h^k) \sim \pi_h^k(\cdot, \cdot | s_h^k)$, and draw $s_{h+1}^k \sim P_h(\cdot \mid s_h^k, a_h^k, b_h^k)$.

7        $N_h^k(s_h^k, a_h^k, b_h^k) \leftarrow N_h^{k-1}(s_h^k, a_h^k, b_h^k) + 1; \quad n \leftarrow N_h^k(s_h^k, a_h^k, b_h^k); \quad \eta_n \leftarrow \frac{H+1}{H+n}$.

8        $\left[ Q_h^{\mathrm{UCB},k+1}, Q_h^{\mathrm{LCB},k+1} \right](s_h^k, a_h^k, b_h^k) \leftarrow \text{update-q}\,()$.

9        $\overline{Q}_h^{\mathrm{R},k+1}(s_h^k, a_h^k, b_h^k) \leftarrow \text{update-ur}\,()$.

10       $\underline{Q}_h^{\mathrm{R},k+1}(s_h^k, a_h^k, b_h^k) \leftarrow \text{update-ur}\,()$.

11       $\overline{Q}_h^{k+1}(s_h^k, a_h^k, b_h^k) \leftarrow \min \{\overline{Q}_h^{\mathrm{R},k+1}(s_h^k, a_h^k, b_h^k), Q_h^{\mathrm{UCB},k+1}(s_h^k, a_h^k, b_h^k), \overline{Q}_h^k(s_h^k, a_h^k, b_h^k)\}$.

12       $\underline{Q}_h^{k+1}(s_h^k, a_h^k, b_h^k) \leftarrow \max \{\underline{Q}_h^{\mathrm{R},k+1}(s_h^k, a_h^k, b_h^k), Q_h^{\mathrm{LCB},k+1}(s_h^k, a_h^k, b_h^k), \underline{Q}_h^k(s_h^k, a_h^k, b_h^k)\}$.

13       **if** $\overline{Q}_h^{k+1}(s_h^k, a_h^k, b_h^k) = \min \{\overline{Q}_h^{\mathrm{R},k+1}(s_h^k, a_h^k, b_h^k), Q_h^{\mathrm{UCB},k+1}(s_h^k, a_h^k, b_h^k)\}$ *and*

           $\underline{Q}_h^{k+1}(s_h^k, a_h^k, b_h^k) = \max \{\underline{Q}_h^{\mathrm{R},k+1}(s_h^k, a_h^k, b_h^k), Q_h^{\mathrm{LCB},k+1}(s_h^k, a_h^k, b_h^k)\}$ **then**

14         $\pi_h^{k+1}(\cdot, \cdot | s_h^k) \leftarrow \text{CCE}(\overline{Q}_h^{k+1}(s_h^k, \cdot, \cdot), \underline{Q}_h^{k+1}(s_h^k, \cdot, \cdot))$.

15       $\overline{V}_h^{k+1}(s_h^k) \leftarrow \min\{(\mathbb{D}_{\pi_h^{k+1}}\overline{Q}_h^{k+1})(s_h^k), \overline{V}_h^k(s_h^k)\}$;

           $\underline{V}_h^{k+1}(s_h^k) \leftarrow \max\{(\mathbb{D}_{\pi_h^{k+1}}\underline{Q}_h^{k+1})(s_h^k), \underline{V}_h^k(s_h^k)\}$.

16       **if** $\overline{V}_h^{k+1}(s_h^k) - \underline{V}_h^{k+1}(s_h^k) > 1$ **then**

17         $\overline{V}_h^{\mathrm{R},k+1}(s_h^k) \leftarrow \overline{V}_h^{k+1}(s_h^k); \quad \underline{V}_h^{\mathrm{R},k+1}(s_h^k) \leftarrow \underline{V}_h^{k+1}(s_h^k)$.

18       **else if** $u_{\mathrm{r}}^k(s_h^k) = \mathsf{True}$ **then**

19         $\overline{V}_h^{\mathrm{R},k+1}(s_h^k) \leftarrow \overline{V}_h^{k+1}(s_h^k); \quad \underline{V}_h^{\mathrm{R},k+1}(s_h^k) \leftarrow \underline{V}_h^{k+1}(s_h^k); \quad u_{\mathrm{r}}^{k+1}(s_h^k) = \mathsf{False}$.

20 **Output:** The marginal policies of $\{\pi_h^{K+1}\}_{h=1}^H$: $\left(\{\mu_h\}_{h=1}^H, \{\nu_h\}_{h=1}^H\right)$.

---

**Lemma 1** *For any integer $N > 0$, the following properties hold:*

$$\frac{1}{N^a} \le \sum_{n=1}^N \frac{\eta_n^N}{n^a} \le \frac{2}{N^a}, \qquad \text{for all} \quad \frac{1}{2} \le a \le 1, \tag{19a}$$

$$\max_{1 \le n \le N} \eta_n^N \le \frac{2H}{N}, \qquad \sum_{n=1}^N (\eta_n^N)^2 \le \frac{2H}{N}, \qquad \sum_{N=n}^\infty \eta_n^N \le 1 + \frac{1}{H}. \tag{19b}$$

A.3   KEY LEMMAS

A.3.1   KEY PROPERTIES AND AUXILIARY SEQUENCES

**Properties of the Q-estimate and V-estimate**   Variables $\overline{Q}_h^{\mathrm{R},k}$ and $\overline{Q}_h^k$ (resp. $\overline{V}_h^k$) provide an "optimistic view" of $Q^{\dagger,\nu_h^k}$ and $Q^*$ (resp. $V^{\dagger,\nu_h^k}$ and $V^*$), as stated in Lemma 2. Similarly, $\underline{Q}_h^{\mathrm{R},k}$ and $\underline{Q}_h^k$ (resp. $\underline{V}_h^k$) provide a "pessimism view" of $Q^{\mu_h^k,\dagger}$ and $Q^*$ (resp. $V^{\mu_h^k,\dagger}$ and $V^*$). Lemma 2 is proved in Appendix C.1.

**Lemma 2** *Consider any $\delta \in (0,1)$. Suppose that $c_b > 0$ is some sufficiently large constant. Then with probability at least $1 - \delta$,*

$$\overline{Q}_h^{\mathrm{R},k}(s,a,b) \geq \overline{Q}_h^k(s,a,b) \geq Q^{\dagger,\nu_h^k}(s,a,b) \geq Q^\star(s,a,b), \quad \overline{V}_h^k(s) \geq V^{\dagger,\nu_h^k}(s) \geq V^\star(s), \quad (20)$$

$$\underline{Q}_h^{\mathrm{R},k}(s,a,b) \leq \underline{Q}_h^k(s,a,b) \leq Q^{\mu_h^k,\dagger}(s,a,b) \leq Q^\star(s,a,b), \quad \underline{V}_h^k(s) \leq V^{\mu_h^k,\dagger}(s) \leq V^\star(s) \quad (21)$$

*hold simultaneously for all $(s,a,b,h) \in \mathcal{S} \times \mathcal{A} \times \mathcal{B} \times [H]$.*

**Properties of the Q-estimate and V-estimate**  To begin with, it is straightforward to see that the update rule in Algorithm 3 (cf. lines 11-12) ensures the following monotonicity property: for all $(s,a,b,k,h) \in \mathcal{S} \times \mathcal{A} \times \mathcal{B} \times [K] \times [H]$

$$\overline{Q}_h^{k+1}(s,a,b) \leq \overline{Q}_h^k(s,a,b), \quad \underline{Q}_h^{k+1}(s,a,b) \geq \underline{Q}_h^k(s,a,b). \quad (22)$$

Similarly, based on line 15 of Algorithm 3, the monotonicity of $\overline{V}_h^k$ and $\underline{V}_h^k$ can be obtained as

$$\overline{V}_h^{k+1}(s) \leq \overline{V}_h^k(s), \quad \underline{V}_h^{k+1}(s) \geq \underline{V}_h^k(s). \quad (23)$$

Besides, the update rules in line 11-12 of Algorithm 3 also result in the following property:

$$\overline{Q}_h^{\mathrm{R},k}(s,a,b) \geq \overline{Q}_h^k(s,a,b), \quad \underline{Q}_h^{\mathrm{R},k}(s,a,b) \leq \underline{Q}_h^k(s,a,b). \quad (24)$$

Lemma 2 implies that $\overline{V}_h^k$ (resp. $\underline{V}_h^k$) is a pointwise upper bound (resp. lower bound) on $V_h^\star$. Taking this result together with the non-increasing (resp. non-decreasing) property (23), we see that $\overline{V}_h^k$ and $\underline{V}_h^k$ become increasingly tighter estimates- of $V_h^\star$ as the number of episodes $k$ increases, which means that it becomes increasingly more likely for $\overline{V}_h^k$ and $\underline{V}_h^k$ to stay close to each other as $k$ increases. Furthermore, it indicates that the confidence interval that contains the optimal value $V_h^\star$ becomes shorter and shorter, as asserted by the following lemma.

**Lemma 3** *For any given $\delta \in (0,1)$, with probability at least $1 - \delta$,*

$$\sum_{h=1}^H \sum_{k=1}^K \mathbb{1}\left(\overline{V}_h^k(s_h^k,a_h^k,b_h^k) - \underline{V}_h^k(s_h^k,a_h^k,b_h^k) > \varepsilon\right) \lesssim \frac{H^6 SAB \log \frac{SABT}{\delta}}{2\varepsilon^2} \quad (25)$$

*holds for all $\varepsilon \in (0,H]$.*

Combining with (24), we can straightforwardly see that with probability at least $1 - \delta$:

$$\overline{Q}_h^{\mathrm{R},k}(s,a,b) \geq Q_h^\star(s,a,b) \geq \underline{Q}_h^{\mathrm{R},k}(s,a,b) \quad \text{for all } (k,h,s,a,b) \in [K] \times [H] \times \mathcal{S} \times \mathcal{A} \times \mathcal{B}. \quad (26)$$

**Properties of the reference $\overline{V}_h^{\mathrm{R},k}$ and $\underline{V}_h^{\mathrm{R},k}$**  As stated in Section 3.1, the conclusions of reference values guarantee that (i) our value function estimate and the reference value are always sufficiently close, and (ii) the aggregate difference between $\overline{V}_h^{\mathrm{R},k}$ and the final reference value $\overline{V}_h^{\mathrm{R},K}$ (resp. $\underline{V}_h^{\mathrm{R},k}$ and the final reference value $\underline{V}_h^{\mathrm{R},K}$) is nearly independent of the sample size $T$ (except for some logarithmic scaling). The above conclusions are summarised as follows and justified in Appendix C.3.

**Lemma 4** *Consider any $\delta \in (0,1)$. Suppose that $c_b > 0$ is some sufficiently large constant. Then with probability exceeding $1 - \delta$, one has*

$$\left|\overline{V}_h^k(s) - \overline{V}_h^{\mathrm{R},k}(s)\right| \leq 2, \quad \left|\underline{V}_h^k(s) - \underline{V}_h^{\mathrm{R},k}(s)\right| \leq 2 \quad (27)$$

*for all $(s,k,h) \in \mathcal{S} \times [K] \times [H]$, and*

$$\sum_{h=1}^H \sum_{k=1}^K \left(\overline{V}_h^{\mathrm{R},k}(s_h^k) - \overline{V}_h^{\mathrm{R},K}(s_h^k)\right) \lesssim H^6 SAB \log \frac{SABT}{\delta}, \quad (28)$$

$$\sum_{h=1}^H \sum_{k=1}^K \left(\underline{V}_h^{\mathrm{R},K}(s_h^k) - \underline{V}_h^{\mathrm{R},k}(s_h^k)\right) \lesssim H^6 SAB \log \frac{SABT}{\delta}. \quad (29)$$

### A.3.2 MAIN SUMMARY OF STEPS IN SECTION 4

The key proof of Step 1 is based on the Lemma 2, which allows one to upper bound the regret as follows

$$\mathrm{Regret}(K) = \sum_{k=1}^{K}\left(V_1^{\dagger,\nu^k} - V_1^{\mu^k,\dagger}\right)\left(s_1^k\right) \le \sum_{k=1}^{K}\left(\overline{V}_1^k - \underline{V}_1^k\right)\left(s_1^k\right). \tag{30}$$

To continue, it boils down to controlling $\left(\overline{V}_1^k - \underline{V}_1^k\right)\left(s_1^k\right)$. Towards this end, we intend to examine $\left(\overline{V}_1^k - \underline{V}_1^k\right)\left(s_1^k\right)$ across all time steps $1 \le h \le H$, which admits the following decomposition:

$$\begin{aligned}
\overline{V}_h^k(s_h^k) - \underline{V}_h^k(s_h^k) &\le \mathbb{E}_{(a,b)\sim\pi_h^k}(\overline{Q}_h^k - \underline{Q}_h^k)(s_h^k,a,b) = \overline{Q}_h^k(s_h^k,a_h^k,b_h^k) - \underline{Q}_h^k(s_h^k,a_h^k,b_h^k) + \zeta_h^k \\
&\le \overline{Q}_h^{\mathrm{R},k}(s_h^k,a_h^k,b_h^k) - \underline{Q}_h^{\mathrm{R},k}(s_h^k,a_h^k,b_h^k) + \zeta_h^k,
\end{aligned} \tag{31}$$

where

$$\zeta_h^k := \mathbb{E}_{(a,b)\sim\pi_h^k}(\overline{Q}_h^k - \underline{Q}_h^k)(s_h^k,a,b) - (\overline{Q}_h^k - \underline{Q}_h^k)(s_h^k,a_h^k,b_h^k). \tag{32}$$

Summing (30) and (31) over $1 \le k \le K$, we reach at

$$\sum_{k=1}^{K}\left(V_h^{\dagger,\nu^k} - V_h^{\mu^k,\dagger}\right)\left(s_h^k\right) \le \sum_{k=1}^{K}\left(\overline{Q}_h^{\mathrm{R},k}(s_h^k,a_h^k,b_h^k) - \underline{Q}_h^{\mathrm{R},k}(s_h^k,a_h^k,b_h^k)\right) + \sum_{k=1}^{K}\zeta_h^k. \tag{33}$$

The key decomposition terms and their bounds in steps 2 and 3 are summarized in Lemmas 5 and 6, respectively.

**Lemma 5** *Fix $\delta \in (0,1)$. Suppose that $c_{\mathrm{b}} > 0$ is some sufficiently large constant. Then with probability at least $1 - \delta$, one has*

$$\sum_{k=1}^{K}\overline{V}_1^k(s_1^k) - \underline{V}_1^k(s_1^k) \le \mathcal{D}_1 + \mathcal{D}_2 + \mathcal{D}_3, \tag{34}$$

*where*

$$\mathcal{D}_1 = \sum_{h=1}^{H}\left(1 + \frac{1}{H}\right)^{h-1}\left(2HSAB + 16c_{\mathrm{b}}(SAB)^{3/4}K^{1/4}H^2\log\frac{SABT}{\delta} + \sum_{k=1}^{K}\zeta_h^k\right), \tag{35a}$$

$$\mathcal{D}_2 = \sum_{h=1}^{H}\left(1 + \frac{1}{H}\right)^{h-1}\left(\sum_{k=1}^{K}\left(\overline{B}_h^{\mathrm{R},k}\left(s_h^k,a_h^k,b_h^k\right) + \underline{B}_h^{\mathrm{R},k}\left(s_h^k,a_h^k,b_h^k\right)\right)\right), \tag{35b}$$

$$\begin{aligned}
\mathcal{D}_3 = &\sum_{h=1}^{H}\sum_{k=1}^{K}\lambda_h^k\left(\left(P_h^k - P_{h,s_h^k,a_h^k,b_h^k}\right)\left(V_{h+1}^{\star} - \overline{V}_{h+1}^{\mathrm{R},k}\right) - \left(P_h^k - P_{h,s_h^k,a_h^k,b_h^k}\right)\left(V_{h+1}^{\star} - \underline{V}_{h+1}^{\mathrm{R},k}\right)\right) \\
&+ \sum_{h=1}^{H}\sum_{k=1}^{K}\lambda_h^k\frac{\sum_{i=1}^{N_h^k(s_h^k,a_h^k,b_h^k)}\left(\overline{V}_{h+1}^{\mathrm{R},k^i}\left(s_{h+1}^{k^i}\right) - P_{h,s_h^k,a_h^k,b_h^k}\overline{V}_{h+1}^{\mathrm{R},k}\right)}{N_h^k\left(s_h^k,a_h^k,b_h^k\right)} \\
&- \sum_{h=1}^{H}\sum_{k=1}^{K}\lambda_h^k\left(\frac{\sum_{i=1}^{N_h^k(s_h^k,a_h^k,b_h^k)}\left(\underline{V}_{h+1}^{\mathrm{R},k^i}\left(s_{h+1}^{k^i}\right) - P_{h,s_h^k,a_h^k,b_h^k}\underline{V}_{h+1}^{\mathrm{R},k}\right)}{N_h^k\left(s_h^k,a_h^k,b_h^k\right)}\right)
\end{aligned} \tag{35c}$$

*with*

$$\lambda_h^k = \left(1 + \frac{1}{H}\right)^{h-1}\sum_{N=N_h^k(s_h^k,a_h^k,b_h^k)}^{N_h^{K-1}(s_h^k,a_h^k,b_h^k)}\eta_{N_h^k(s_h^k,a_h^k,b_h^k)}^{N}. \tag{36}$$

**Lemma 6** *With any $\delta \in (0,1)$, the following upper bounds hold with probability at least $1 - \delta$:*

$$\mathcal{D}_1 \lesssim \sqrt{H^2SABT\log\frac{SABT}{\delta}} + H^{4.5}SAB\log^2\frac{SABT}{\delta}, \tag{37}$$

$$\mathcal{D}_2 \lesssim \sqrt{H^2SABT\log\frac{SABT}{\delta}} + H^4SAB\log^2\frac{SABT}{\delta}, \tag{38}$$

$$\mathcal{D}_3 \lesssim \sqrt{H^2SABT\log^4\frac{SABT}{\delta}} + H^6SAB\log^3\frac{SABT}{\delta}. \tag{39}$$

## B   FREEDMAN'S INEQUALITY

### B.1   A USER-FRIENDLY VERSION OF FREEDMAN'S INEQUALITY

Before proceeding to the analysis that follows, we first introduce the well-known Freedman's inequality (Freedman, 1975; Tropp, Jan. 2011), which extends Bernstein's inequality to accommodate martingales and is exactly matched with the Markovian structure of our problem. We first provide a user-friendly version of the Freedman inequality introduced in (Li et al., 2023, Section C).

**Theorem 4 (Freedman's inequality)** *Consider a filtration $\mathcal{F}_0 \subset \mathcal{F}_1 \subset \mathcal{F}_2 \subset \cdots$, and let $\mathbb{E}_k$ stand for the expectation conditioned on $\mathcal{F}_k$. Suppose that $Y_n = \sum_{k=1}^n X_k \in \mathbb{R}$, where $\{X_k\}$ is a real-valued scalar sequence obeying*

$$|X_k| \leq R \qquad and \qquad \mathbb{E}_{k-1}[X_k] = 0 \qquad for\ all\ k \geq 1$$

*for some quantity $R < \infty$. We also define*

$$F_n := \sum_{k=1}^n \mathbb{E}_{k-1}[X_k^2].$$

*In addition, suppose that $F_n \leq \psi^2$ holds deterministically for some given quantity $\psi^2 < \infty$. Then for any positive integer $m \geq 1$, with probability at least $1 - \delta$ one has*

$$|Y_n| \leq \sqrt{8 \max\left\{F_n, \frac{\psi^2}{2^m}\right\} \log \frac{2m}{\delta}} + \frac{4}{3} R \log \frac{2m}{\delta}. \tag{40}$$

### B.2   APPLICATION OF FREEDMAN'S INEQUALITY

Before diving into the subsequent proof, we first introduce two lemmas based on Friedman's inequality. Our first result is concerned with a martingale concentration bound as follows.

**Lemma 7** *Let $\{F_h^k \in \mathbb{R}^S \mid 1 \leq k \leq K, 1 \leq h \leq H+1\}$ and $\{G_h^i(s,a,b,N) \in \mathbb{R} \mid 1 \leq k \leq K, 1 \leq h \leq H+1\}$ be the collections of vectors and scalars, respectively, and suppose that they obey the following properties:*

- *$F_h^k$ is fully determined by the samples collected up to the end of the $(h-1)$-th step of the $i$-th episode;*

- *$\|F_h^k\|_\infty \leq C_{\mathrm{f}}$;*

- *$G_h^k(s,a,b,N)$ is fully determined by the samples collected up to the end of the $(h-1)$-th step of the $i$-th episode, and a given positive integer $N \in [K]$;*

- *$0 \leq G_h^k(s,a,b,N) \leq C_{\mathrm{g}}$;*

- *$0 \leq \sum_{n=1}^{N_h^k(s,a,b)} G_h^{k_h^n(s,a,b)}(s,a,b,N) \leq 2$.*

*In addition, for $1 \leq k \leq K$, consider the following sequence*

$$X_k(s,a,b,h,N) := G_h^k(s,a,b,N)\big(P_h^k - P_{h,s,a,b}\big)F_{h+1}^k \mathbb{1}\big\{(s_h^k, a_h^k, b_h^k) = (s,a,b)\big\}, \tag{41}$$

*with $P_h^k$ defined in (15). Consider any $\delta \in (0,1)$. Then with probability at least $1 - \delta$,*

$$\left|\sum_{i=1}^k X_k(s,a,b,h,N)\right| \lesssim \sqrt{C_{\mathrm{g}} \log^2 \frac{SABT}{\delta}} \sqrt{\sum_{n=1}^{N_h^k(s,a,b)} u_h^{k_h^n(s,a,b)}(s,a,b,N) \mathsf{Var}_{h,s,a,b}\big(W_{h+1}^{k_h^n(s,a,b)}\big)}$$

$$+ \left(C_{\mathrm{g}}C_{\mathrm{f}} + \sqrt{\frac{C_{\mathrm{g}}}{N}} C_{\mathrm{f}}\right) \log^2 \frac{SABT}{\delta} \tag{42}$$

*holds simultaneously for all $(k,h,s,a,b,N) \in [K] \times [H] \times \mathcal{S} \times \mathcal{A} \times \mathcal{B} \times [K]$.*

**Proof B.1** *On a non-ambiguous basis, we will abbreviate $X_k(s, a, b, h, N)$ as $X_i$ throughout the proof of this lemma. The main idea of our proof is to control the $\sum_{i=1}^{k} X_i$ term by Freedman's inequality (see Theorem 4).*

*Consider any given $(k, h, s, a, b, N) \in [K] \times [H] \times \mathcal{S} \times \mathcal{A} \times \mathcal{B} \times [K]$. It can be easily verified that*

$$\mathbb{E}_{k-1}[X_k] = 0,$$

*where $\mathbb{E}_{k-1}$ denotes the expectation conditioned on everything happening up to the end of the $(h-1)$-th step of the $k$-th episode. Furthermore, under the assumptions $\|F_{h+1}^k\|_\infty \leq C_{\mathrm{f}}$, $0 \leq G_h^k(s, a, b, N) \leq C_{\mathrm{g}}$, and the basic facts $\|P_h^k\|_1 = \|P_{h,s,a,b}\|_1 = 1$, we have the following crude bound:*

$$\left|X_i\right| \leq G_h^k(s, a, b, N)\left|\left(P_h^k - P_{h,s,a,b}\right)F_{h+1}^k\right|$$

$$\leq G_h^k(s, a, b, N)\left(\left\|P_h^k\right\|_1 + \left\|P_{h,s,a,b}\right\|_1\right)\left\|F_{h+1}^k\right\|_\infty \leq 2C_{\mathrm{f}}C_{\mathrm{g}}. \tag{43}$$

*Under the definition of the variance parameter in (16), we obtain*

$$\sum_{k=1}^{k} \mathbb{E}_{k-1}\left[\left|X_k\right|^2\right] = \sum_{k=1}^{k}\left(G_h^k(s, a, b, N)\right)^2 \mathbb{1}\left\{(s_h^k, a_h^k, b_h^k) = (s, a, b)\right\}\mathbb{E}_{k-1}\left[\left|\left(P_h^k - P_{h,s,a,b}\right)F_{h+1}^k\right|^2\right]$$

$$= \sum_{n=1}^{N_h^k(s,a,b)}\left(G_h^{k_h^n(s,a,b)}(s, a, b, N)\right)^2 \mathsf{Var}_{h,s,a,b}\left(F_{h+1}^{k_h^n(s,a,b)}\right)$$

$$\leq C_{\mathrm{g}}\left(\sum_{n=1}^{N_h^k(s,a,b)} G_h^{k_h^n(s,a,b)}(s, a, b, N)\right)\left\|F_{h+1}^{k_h^n(s,a,b)}\right\|_\infty^2 \leq 2C_{\mathrm{g}}C_{\mathrm{f}}^2, \tag{44}$$

*where the inequalities hold true due to the assumptions $\|F_h^k\|_\infty \leq C_{\mathrm{f}}$, $0 \leq G_h^k(s, a, b, N) \leq C_{\mathrm{g}}$, and $0 \leq \sum_{n=1}^{N_h^k(s,a,b)} G_h^{k_h^n(s,a,b)}(s, a, b, N) \leq 2$.*

*Armed with (43) and (44), we can invoke Theorem 4 (with $m = \lceil\log_2 N\rceil$) and take the union bound over all $(k, h, s, a, N) \in [K] \times [H] \times \mathcal{S} \times \mathcal{A} \times \mathcal{B} \times [K]$ to show that: with probability at least $1 - \delta$,*

$$\left|\sum_{k=1}^{k} X_k\right| \lesssim C_{\mathrm{g}}C_{\mathrm{f}} \log \frac{SABT^2 \log N_h^k}{\delta}$$

$$+ \sqrt{\max\left\{C_{\mathrm{g}} \sum_{n=1}^{N_h^k(s,a,b)} u_h^{k_h^n(s,a,b)}(s, a, b, N)\mathsf{Var}_{h,s,a}\left(W_{h+1}^{k_h^n(s,a,b)}\right), \frac{C_{\mathrm{g}}C_{\mathrm{f}}^2}{N}\right\} \log \frac{SABT^2 \log N}{\delta}}$$

$$\lesssim \sqrt{C_{\mathrm{g}}\log^2 \frac{SABT}{\delta}}\sqrt{\sum_{n=1}^{N_h^k(s,a,b)} u_h^{k_h^n(s,a,b)}(s, a, b, N)\mathsf{Var}_{h,s,a}\left(W_{h+1}^{k_h^n(s,a,b)}\right)}$$

$$+ \left(C_{\mathrm{g}}C_{\mathrm{f}} + \sqrt{\frac{C_{\mathrm{g}}}{N}}C_{\mathrm{f}}\right)\log^2 \frac{SABT}{\delta}$$

*holds simultaneously for all $(k, h, s, a, b, N) \in [K] \times [H] \times \mathcal{S} \times \mathcal{A} \times \mathcal{B} \times [K]$.*

**Lemma 8** *Let $\left\{N(s, a, b, h) \in [K] \mid (s, a, b, h) \in \mathcal{S} \times \mathcal{A} \times \mathcal{B} \times [H]\right\}$ be a collection of positive integers, and let $\{c_h : 0 \leq c_h \leq e, h \in [H]\}$ be a collection of fixed and bounded universal constants. Moreover, let $\left\{F_h^k \in \mathbb{R}^S \mid 1 \leq k \leq K, 1 \leq h \leq H+1\right\}$ and $\left\{G_h^k(s_h^k, a_h^k, b_h^k) \in \mathbb{R} \mid 1 \leq i \leq K, 1 \leq h \leq H+1\right\}$ represent respectively a collection of random vectors and scalars, which obey the following properties.*

- *$F_h^k$ is fully determined by the samples collected up to the end of the $(h-1)$-th step of the $k$-th episode;*

- *$\|F_h^k\|_\infty \leq C_{\mathrm{f}}$ and $F_h^k \geq 0$;*

- $G_h^k(s_h^k, a_h^k, b_h^k)$ is fully determined by the integer $N(s_h^k, a_h^k, b_h^k, h)$ and all samples collected up to the end of the $(h-1)$-th step of the $k$-th episode;

- $0 \leq G_h^k(s_h^k, a_h^k, b_h^k) \leq C_{\mathrm{g}}$.

*Consider any $\delta \in (0, 1)$, and introduce the following sequences*

$$X_{k,h} := G_h^k(s_h^k, a_h^k, b_h^k)\big(P_h^k - P_{h,s_h^k,a_h^k,b_h^k}\big)F_{h+1}^k, \qquad 1 \leq k \leq K, 1 \leq h \leq H+1, \quad (45)$$

$$Y_{k,h} := c_h\big(P_h^k - P_{h,s_h^k,a_h^k,b_h^k}\big)F_{h+1}^k, \qquad 1 \leq k \leq K, 1 \leq h \leq H+1. \quad (46)$$

*Then with probability at least $1 - \delta$,*

$$\left|\sum_{h=1}^H \sum_{k=1}^K X_{k,h}\right| \lesssim \sqrt{C_{\mathrm{g}}^2 \sum_{h=1}^H \sum_{k=1}^K \mathbb{E}_{k,h-1}\left[\left|(P_h^k - P_{h,s_h^k,a_h^k,b_h^k})F_{h+1}^k\right|^2\right] \log \frac{T^{HSAB}}{\delta}}$$

$$+ C_{\mathrm{g}} C_{\mathrm{f}} \log \frac{T^{HSAB}}{\delta}$$

$$\lesssim \sqrt{C_{\mathrm{g}}^2 C_{\mathrm{f}} \sum_{h=1}^H \sum_{k=1}^K \mathbb{E}_{k,h-1}\left[P_h^k W_{h+1}^k\right] \log \frac{T^{HSAB}}{\delta}} + C_{\mathrm{g}} C_{\mathrm{f}} \log \frac{T^{HSAB}}{\delta}$$

$$\left|\sum_{h=1}^H \sum_{k=1}^K Y_{k,h}\right| \lesssim \sqrt{T C_{\mathrm{f}}^2 \log \frac{1}{\delta}} + C_{\mathrm{f}} \log \frac{1}{\delta}$$

*hold simultaneously for all possible collections $\{N(s, a, b, h) \in \mathcal{S} \times \mathcal{A} \times \mathcal{B} \times [H]\}$.*

**Proof B.2** *This lemma can also be proved by Freedman's inequality (cf. Theorem 4).*

- *We start by controlling the first term $\sum_{h=1}^H \sum_{k=1}^K X_{k,h}$. It is readily seen that for any given $\{N(s, a, b, h) \in \mathcal{S} \times \mathcal{A} \times \mathcal{B} \times [H]\}$, we have*

$$\mathbb{E}_{k,h-1}\left[X_i\right] = \mathbb{E}_{k,h-1}\left[G_h^k(s_h^k, a_h^k, b_h^k)\big(P_h^k - P_{h,s_h^k,a_h^k,b_h^k}\big)F_{h+1}^k\right] = 0,$$

*where $\mathbb{E}_{k,h-1}$ denotes the expectation conditioned on everything happening before step $h$ of the $k$-th episode. In addition, we make note of the following crude bound:*

$$\left|X_{k,h}\right| \leq G_h^k(s_h^k, a_h^k, b_h^k)\left|\big(P_h^k - P_{h,s_h^k,a_h^k,b_h^k}\big)F_{h+1}^k\right|$$

$$\leq G_h^k(s_h^k, a_h^k, b_h^k)\big(\|P_h^k\|_1 + \|P_{h,s_h^k,a_h^k,b_h^k}\|_1\big)\|F_{h+1}^k\|_\infty \leq 2C_{\mathrm{f}} C_{\mathrm{g}}, \quad (47)$$

*which arises from the assumptions $\|F_{h+1}^k\|_\infty \leq C_{\mathrm{f}}$, $0 \leq G_h^k(s, a, b, N) \leq C_{\mathrm{g}}$ together with the basic fact $\|P_h^k\|_1 = \|P_{h,s_h^k,a_h^k,b_h^k}\|_1 = 1$. Additionally, we can calculate that*

$$\sum_{h=1}^H \sum_{k=1}^K \mathbb{E}_{k,h-1}\left[\left|X_{k,h}\right|^2\right] = \sum_{h=1}^H \sum_{k=1}^K \big(G_h^k(s_h^k, a_h^k, b_h^k)\big)^2 \mathbb{E}_{k,h-1}\left[\left|(P_h^k - P_{h,s_h^k,a_h^k,b_h^k})F_{h+1}^k\right|^2\right]$$

$$\leq C_{\mathrm{g}}^2 \sum_{h=1}^H \sum_{k=1}^K \mathbb{E}_{k,h-1}\left[\left|(P_h^k - P_{h,s_h^k,a_h^k,b_h^k})F_{h+1}^k\right|^2\right] \quad (48)$$

$$\leq C_{\mathrm{g}}^2 \sum_{h=1}^H \sum_{k=1}^K \mathbb{E}_{k,h-1}\left[\left|P_h^k F_{h+1}^k\right|^2\right], \quad (49)$$

*where the first inequality comes from the assumption $0 \le G_h^k(s_h^k, a_h^k, b_h^k) \le C_{\mathrm{g}}$. Furthermore, (49) can be expressed as*

$$\sum_{h=1}^{H}\sum_{k=1}^{K} \mathbb{E}_{k,h-1}\left[\left|X_{k,h}\right|^2\right] \overset{(a)}{\le} C_{\mathrm{g}}^2 \sum_{h=1}^{H}\sum_{k=1}^{K} \mathbb{E}_{k,h-1}\left[P_h^k\left(F_{h+1}^k\right)^2\right]$$

$$\overset{(b)}{\le} C_{\mathrm{g}}^2 \sum_{h=1}^{H}\sum_{k=1}^{K}\left\|F_{h+1}^k\right\|_\infty \mathbb{E}_{k,h-1}\left[P_h^k W_{h+1}^k\right]$$

$$\overset{(c)}{\le} C_{\mathrm{g}}^2 C_{\mathrm{f}} \sum_{h=1}^{H}\sum_{k=1}^{K} \mathbb{E}_{k,h-1}\left[P_h^k F_{h+1}^k\right] \tag{50}$$

$$\le C_{\mathrm{g}}^2 C_{\mathrm{f}} \sum_{h=1}^{H}\sum_{i=1}^{K}\left\|F_{h+1}^k\right\|_\infty \overset{(d)}{\le} HKC_{\mathrm{g}}^2 C_{\mathrm{f}}^2 = TC_{\mathrm{g}}^2 C_{\mathrm{f}}^2. \tag{51}$$

*Notably, (a) comes from the fact that $P_h^k$ only has one non-zero entry (cf. (15)), (b) holds due to the assumption that $F_h^k$ is non-negative, whereas (c) and (d) rely on $\|F_h^k\|_\infty \le C_{\mathrm{f}}$.*

*With (47), (50), and (51), we can invoke Theorem 4 (with $m = \lceil \log_2 T \rceil$) and take the union bound over all possible collections $\left\{ N(s,a,b,h) \in [K] \mid (s,a,b,h) \in \mathcal{S} \times \mathcal{A} \times \mathcal{B} \times [H] \right\}$, which have at most $K^{HSAB}$ possibilities, to show that: with probability at least $1 - \delta$,*

$$\left|\sum_{h=1}^{H}\sum_{k=1}^{k} X_{k,h}\right| \lesssim C_{\mathrm{g}}C_{\mathrm{f}} \log \frac{K^{HSAB} \log T}{\delta}$$

$$+ \sqrt{\max\left\{ C_{\mathrm{g}}^2 \sum_{h=1}^{H}\sum_{k=1}^{K} \mathbb{E}_{k,h-1}\left[\left|(P_h^k - P_{h,s_h^k,a_h^k,b_h^k})F_{h+1}^k\right|^2\right], \frac{TC_{\mathrm{g}}^2 C_{\mathrm{f}}^2}{2^m} \right\} \log \frac{K^{HSAB} \log T}{\delta}}$$

$$\lesssim \sqrt{C_{\mathrm{g}}^2 \sum_{h=1}^{H}\sum_{k=1}^{K} \mathbb{E}_{k,h-1}\left[\left|(P_h^k - P_{h,s_h^k,a_h^k,b_h^k})F_{h+1}^k\right|^2\right] \log \frac{T^{HSAB}}{\delta}} + C_{\mathrm{g}}C_{\mathrm{f}} \log \frac{T^{HSAB}}{\delta}$$

$$\lesssim \sqrt{C_{\mathrm{g}}^2 C_{\mathrm{f}} \sum_{h=1}^{H}\sum_{k=1}^{K} \mathbb{E}_{k,h-1}\left[P_h^k F_{h+1}^k\right] \log \frac{T^{HSAB}}{\delta}} + C_{\mathrm{g}}C_{\mathrm{f}} \log \frac{T^{HSAB}}{\delta}$$

*holds simultaneously for all $\left\{ N(s,a,b,h) \in [K] \mid (s,a,b,h) \in \mathcal{S} \times \mathcal{A} \times \mathcal{B} \times [H] \right\}$.*

- *Then we turn to control the second term $\left|\sum_{h=1}^{H}\sum_{k=1}^{K} Y_{k,h}\right|$ of interest. Similar to $\left|\sum_{h=1}^{H}\sum_{k=1}^{K} X_{k,h}\right|$, we have*

$$|Y_{k,h}| \le 2eC_{\mathrm{f}},$$

$$\sum_{h=1}^{H}\sum_{k=1}^{K} \mathbb{E}_{k,h-1}\left[\left|Y_{k,h}\right|^2\right] \le e^2 TC_{\mathrm{f}}^2.$$

*Invoke Theorem 4 (with $m = 1$) to arrive at*

$$\left|\sum_{h=1}^{H}\sum_{k=1}^{K} Y_{k,h}\right| \lesssim \sqrt{TC_{\mathrm{f}}^2 \log \frac{1}{\delta}} + C_{\mathrm{f}} \log \frac{1}{\delta}. \tag{52}$$

## C   PROOF OF KEY LEMMAS IN SECTION A.3.1

### C.1   PROOF OF LEMMA 2

The proof is by backward induction. Suppose the bounds hold for the $Q$-values in the $(h+1)$-th step, we now establish the bounds for the $V$-values in the $(h+1)$-th step and $Q$-values in the $h$-th step.

For any state $s$,

$$\overline{V}_{h+1}^k(s) = \mathbb{D}_{\pi_{h+1}^k}(\overline{Q}_{h+1}^k)(s) \overset{(i)}{\geq} \max_a \mathbb{E}_{b \sim \nu_{h+1}^k} \overline{Q}_{h+1}^k(s,a,b)$$

$$\geq \max_a \mathbb{E}_{b \sim \nu_{h+1}^k} Q_h^{\dagger,\nu_{h+1}^k}(s,a,b) = V_{h+1}^{\dagger,\nu_{h+1}^k}, \tag{53}$$

where (i) comes from the property (10a) of CCE. Similarly, we can show $\underline{V}_{h+1}^k(s) \leq V^{\mu_{h+1}^k,\dagger}(s)$. Therefore, we have: for all $s$,

$$\overline{V}_{h+1}^k(s) \geq V_{h+1}^{\dagger,\nu_{h+1}^k}(s) \geq V_{h+1}^\star(s) \geq V_{h+1}^{\mu_{h+1}^k,\dagger}(s) \geq \underline{V}_{h+1}^k(s).$$

Now consider an arbitrary triple $(s,a,b)$ in the $h$-th step. We have

$$Q_h^{\dagger,\nu_h^k}(s,a,b) - Q_h^\star(s,a,b) = \mathbb{E}_{s' \sim P_h(\cdot|s,a,b)} \left[ V_{h+1}^{\dagger,\nu_{h+1}^k}(s') - V_{h+1}^\star(s') \right] \geq 0. \tag{54}$$

Similarly, we can show $Q^{\mu_h^k,\dagger}(s,a,b) \leq Q^\star(s,a,b)$. To complete the induction argument, we should prove

$$\overline{Q}_h^k(s,a,b) \geq Q_h^{\dagger,\nu_h^k}(s,a,b), \tag{55}$$

which we shall further accomplish by induction. When $k = 1$, (55) holds since that the initialization obeys $\overline{Q}_h^1 = H \geq Q^{\dagger,\nu_h^1}(s,a,b)$ for all $(h,a,s,b) \in [H] \times \mathcal{S} \times \mathcal{A} \times \mathcal{B}$. Next, under the assumption that (55) holds up to the $k$-th episode, we wish to get it for the $(k+1)$-th episode to get Lemma 2. According to line 11 and lines 13-14 of Algorithm 3, it suffices to justify

$$\min\left\{ \overline{Q}_h^{\text{R},k+1}(s,a,b), Q_h^{\text{UCB},k+1}(s,a,b) \right\} \geq Q_h^{\dagger,\nu_h^{k+1}}(s,a,b). \tag{56}$$

In order to get (56), we need to prove that

$$Q_h^{\text{UCB},k+1}(s,a,b) \geq Q_h^{\dagger,\nu_h^{k+1}}(s,a,b), \tag{57}$$

and

$$\overline{Q}_h^{\text{R},k+1}(s,a,b) \geq Q_h^{\dagger,\nu_h^{k+1}}(s,a,b). \tag{58}$$

For the sake of convenience, suppose $Q_h^{\text{UCB},k'}(s_h^k,a_h^k,b_h^k)$ and $\overline{Q}_h^{\text{R},k'}(s_h^k,a_h^k,b_h^k)$ are latest updated in the $k$-th episode with $k \leq k'$, it suffices to verify

$$Q_h^{\text{UCB},k+1}(s_h^k,a_h^k,b_h^k) \geq Q_h^{\dagger,\nu_h^{k'+1}}(s_h^k,a_h^k,b_h^k),$$

and

$$\overline{Q}_h^{\text{R},k+1}(s_h^k,a_h^k,b_h^k) \geq Q_h^{\dagger,\nu_h^{k'+1}}(s_h^k,a_h^k,b_h^k).$$

First, the proof of (57) is performed.

**Step 1: decomposing** $Q_h^{\text{UCB},k+1}(s_h^k,a_h^k,b_h^k) \geq Q_h^{\dagger,\nu_h^{k'+1}}(s_h^k,a_h^k,b_h^k)$. Firstly, according to the definition of $N_h^k$ and $k^n$ in Appendix A.1, we can obtain

$$Q_h^{\text{UCB},k+1}(s_h^k,a_h^k,b_h^k) = Q_h^{\text{UCB},k^{N_h^k}+1}(s_h^k,a_h^k,b_h^k), \tag{59}$$

since $k^{N_h^k} = k^{N_h^k(s_h^k,a_h^k,b_h^k)} = k$. According to the update rule (i.e., line 2 in Algorithm 2 and line 8 in Algorithm 3), we obtain

$$Q_h^{\text{UCB},k+1}(s_h^k,a_h^k,b_h^k) = Q_h^{\text{UCB},k^{N_h^k}+1}(s_h^k,a_h^k,b_h^k)$$

$$= (1-\eta_{N_h^k})Q_h^{\text{UCB},k^{N_h^k}}(s_h^k,a_h^k,b_h^k) + \eta_{N_h^k}\left\{ r_h(s_h^k,a_h^k,b_h^k) + \overline{V}_{h+1}^{k^{N_h^k}}(s_{h+1}^{N_h^k}) + c_{\text{b}}\sqrt{\frac{H^3 \log \frac{SABT}{\delta}}{N_h^k}} \right\}$$

$$= (1-\eta_{N_h^k})Q_h^{\text{UCB},k^{N_h^{k-1}}+1}(s_h^k,a_h^k,b_h^k) + \eta_{N_h^k}\left\{ r_h(s_h^k,a_h^k,b_h^k) + \overline{V}_{h+1}^{k^{N_h^k}}(s_{h+1}^{N_h^k}) + c_{\text{b}}\sqrt{\frac{H^3 \log \frac{SABT}{\delta}}{N_h^k}} \right\},$$

where the last identity again follows from our argument for justifying (59). Applying this relation recursively and combining with the definitions of $\eta_0^N$ and $\eta_n^N$ in (17), we obtain

$$
Q_h^{\mathrm{UCB},k+1}(s_h^k, a_h^k, b_h^k)
$$
$$
= \eta_0^{N_h^k} Q_h^{\mathrm{UCB},1}(s_h^k, a_h^k, b_h^k) + \sum_{n=1}^{N_h^k} \eta_n^{N_h^k} \left\{ r_h(s_h^k, a_h^k, b_h^k) + \overline{V}_{h+1}^{k^n}(s_{h+1}^{k^n}) + c_{\mathrm{b}} \sqrt{\frac{H^3 \log \frac{SABT}{\delta}}{n}} \right\}.
$$
(60)

With $\eta_0^{N_h^k} + \sum_{n=1}^{N_h^k} \eta_n^{N_h^k} = 1$ in (17) and (18), there is

$$
Q^{\dagger, \nu_h^{k'+1}}(s_h^k, a_h^k, b_h^k) = \eta_0^{N_h^k} Q^{\dagger, \nu_h^{k'+1}}(s_h^k, a_h^k, b_h^k) + \sum_{n=1}^{N_h^k} \eta_n^{N_h^k} Q^{\dagger, \nu_h^{k'+1}}(s_h^k, a_h^k, b_h^k).
$$
(61)

Combining (60) and (61), we can further get

$$
Q_h^{\mathrm{UCB},k+1}(s_h^k, a_h^k, b_h^k) - Q^{\dagger, \nu_h^{k'+1}}(s_h^k, a_h^k, b_h^k) = \eta_0^{N_h^k} \left( Q_h^{\mathrm{UCB},1}(s_h^k, a_h^k, b_h^k) - Q^{\dagger, \nu_h^{k'+1}}(s_h^k, a_h^k, b_h^k) \right)
$$
$$
+ \sum_{n=1}^{N_h^k} \left\{ r_h(s_h^k, a_h^k, b_h^k) + \overline{V}_{h+1}^{k^n}(s_{h+1}^{k^n}) + c_{\mathrm{b}} \sqrt{\frac{H^3 \log \frac{SABT}{\delta}}{n}} - Q^{\dagger, \nu_h^{k'+1}}(s_h^k, a_h^k, b_h^k) \right\}.
$$
(62)

To continue, invoking the Bellman optimality equation

$$
Q^{\dagger, \nu_h^{k'+1}}(s_h^k, a_h^k, b_h^k) = r_h(s_h^k, a_h^k, b_h^k) + P_{h, s_h^k, a_h^k, b_h^k} V_{h+1}^{\dagger, \nu_h^{k'+1}},
$$
(63)

we have

$$
r_h(s_h^k, a_h^k, b_h^k) + \overline{V}_{h+1}^{k^n}(s_{h+1}^{k^n}) - Q^{\dagger, \nu_h^{k'+1}}(s_h^k, a_h^k, b_h^k) = \overline{V}_{h+1}^{k^n}(s_{h+1}^{k^n}) - P_{h, s_h^k, a_h^k, b_h^k} V_{h+1}^{\dagger, \nu_h^{k'+1}}
$$
$$
= \left( P_h^{k^n} - P_{h, s_h^k, a_h^k, b_h^k} \right) V_{h+1}^{\dagger, \nu_h^{k'}} + P_h^{k^n} \left( \overline{V}_{h+1}^{k^n} - V_{h+1}^{\dagger, \nu_h^{k'+1}} \right),
$$
(64)

where $P_h^k$ is defined in (15). Bringing (64 into (62) yields

$$
Q_h^{\mathrm{UCB},k+1}(s_h^k, a_h^k, b_h^k) - Q^{\dagger, \nu_h^{k'+1}}(s_h^k, a_h^k, b_h^k) = \eta_0^{N_h^k} \left( Q_h^{\mathrm{UCB},1}(s_h^k, a_h^k, b_h^k) - Q^{\dagger, \nu_h^{k'+1}}(s_h^k, a_h^k, b_h^k) \right)
$$
$$
+ \sum_{n=1}^{N_h^k} \left\{ \left( P_h^{k^n} - P_{h, s_h^k, a_h^k, b_h^k} \right) V_{h+1}^{\dagger, \nu_h^{k'+1}} + P_h^{k^n} \left( \overline{V}_{h+1}^{k^n} - V_{h+1}^{\dagger, \nu_h^{k'+1}} \right) + c_{\mathrm{b}} \sqrt{\frac{H^3 \log \frac{SABT}{\delta}}{n}} \right\}.
$$
(65)

**Step 2: two key quantities for lower bounding** $Q_h^{\mathrm{UCB},k+1}(s_h^k, a_h^k, b_h^k) \geq Q^{\dagger, \nu_h^{k'+1}}(s_h^k, a_h^k, b_h^k)$.
In order to develop a lower bound on $Q_h^{\mathrm{UCB},k+1}(s_h^k, a_h^k, b_h^k) \geq Q^{\dagger, \nu_h^{k'+1}}(s_h^k, a_h^k, b_h^k)$ based on the decomposition (65), we make note of several simple facts as follows.

(a) The initialization satisfies $Q_h^{\mathrm{UCB},1}(s_h^k, a_h^k, b_h^k) - Q^{\dagger, \nu_h^{k'+1}}(s_h^k, a_h^k, b_h^k) \geq 0$.

(b) According to (53), monotonicity of $\overline{V}_h^k$ in (23) and $k \leq k'$, we can obtaion

$$
\overline{V}_{h+1}^{k^n}(s_{h+1}) \geq \overline{V}_{h+1}^{k'+1} \geq V_{h+1}^{\dagger, \nu_h^{k'+1}}.
$$
(66)

(c) By the Azuma-Hoeffding inequality and a union bound, we have that with probability at least $1 - \delta$, one has

$$
\left| \left( P_h^{k^n} - P_{h, s_h^k, a_h^k, b_h^k} \right) V_{h+1}^{\dagger, \nu_h^{k'}} \right| \leq c_{\mathrm{b}} \sqrt{\frac{H^3 \log \frac{SABT}{\delta}}{n}},
$$
(67)

Thus, $Q_h^{\mathrm{UCB},k+1}(s_h^k, a_h^k, b_h^k) \geq Q^{\dagger, \nu_h^{k'+1}}(s, a, b) \geq 0$ and we have concluded the proof of (57). Next is the proof of (58).

**Step 1: decomposing $\overline{Q}_h^{\mathrm{R},k+1}(s_h^k,a_h^k,b_h^k) - Q^{\dagger,\nu_h^{k'+1}}(s_h^k,a_h^k,b_h^k)$.** Similar to (59), we can get

$$\overline{Q}_h^{\mathrm{R},k^{N_h^k}+1}(s_h^k,a_h^k,b_h^k) = \overline{Q}_h^{\mathrm{R},k^{N_h^k}+1}(s_h^k,a_h^k,b_h^k). \tag{68}$$

According to the update rule (i.e., line 6 in Algorithm 2 and line 9 in Algorithm 3), we obtain

$$\overline{Q}_h^{\mathrm{R},k+1}(s_h^k,a_h^k,b_h^k) = \overline{Q}_h^{\mathrm{R},k^{N_h^k}+1}(s_h^k,a_h^k,b_h^k) = (1-\eta_{N_h^k})\overline{Q}_h^{\mathrm{R},k^{N_h^k}}(s_h^k,a_h^k,b_h^k)$$

$$+\eta_{N_h^k}\left\{r_h(s_h^k,a_h^k,b_h^k) + \overline{V}_{h+1}^{k^{N_h^k}}(s_{h+1}^{k^{N_h^k}}) - \overline{V}_{h+1}^{\mathrm{R},k^{N_h^k}}(s_{h+1}^{k^{N_h^k}}) + \overline{\phi}_h^{\mathrm{r},k^{N_h^k}+1}(s_h^k,a_h^k,b_h^k) + \overline{b}_h^{\mathrm{R},k^{N_h^k}+1}\right\}$$

$$= (1-\eta_{N_h^k})\overline{Q}_h^{\mathrm{R},k^{N_h^k-1}+1}(s_h^k,a_h^k,b_h^k)$$

$$+\eta_{N_h^k}\left\{r_h(s_h^k,a_h^k,b_h^k) + \overline{V}_{h+1}^{k^{N_h^k}}(s_{h+1}^{k^{N_h^k}}) - \overline{V}_{h+1}^{\mathrm{R},k^{N_h^k}}(s_{h+1}^{k^{N_h^k}}) + \overline{\phi}_h^{\mathrm{r},k^{N_h^k}+1}(s_h^k,a_h^k,b_h^k) + \overline{b}_h^{\mathrm{R},k^{N_h^k}+1}\right\},$$

where the last identity again follows from our argument for justifying (68). Applying this relation recursively and invoking the definitions of $\eta_0^N$ and $\eta_n^N$ in (17), we are left with

$$\overline{Q}_h^{\mathrm{R},k+1}(s_h^k,a_h^k,b_h^k) = \eta_0^{N_h^k}\overline{Q}_h^{\mathrm{R},1}(s_h^k,a_h^k,b_h^k)$$

$$+\sum_{n=1}^{N_h^k}\eta_n^{N_h^k}\left\{r_h(s_h^k,a_h^k,b_h^k) + \overline{V}_{h+1}^{k^n}(s_{h+1}^{k^n}) - \overline{V}_{h+1}^{\mathrm{R},k^n}(s_{h+1}^{k^n}) + \overline{\phi}_h^{\mathrm{r},k^n+1}(s_h^k,a_h^k,b_h^k) + \overline{b}_h^{\mathrm{R},k^n+1}\right\}. \tag{69}$$

Additionally, (61) combined with (69) leads to

$$\overline{Q}_h^{\mathrm{R},k+1}(s_h^k,a_h^k,b_h^k) - Q^{\dagger,\nu_h^{k'+1}}(s_h^k,a_h^k,b_h^k) = \eta_0^{N_h^k}\left(\overline{Q}_h^{\mathrm{R},1}(s_h^k,a_h^k,b_h^k) - Q^{\dagger,\nu_h^{k'+1}}(s_h^k,a_h^k,b_h^k)\right)$$

$$+\sum_{n=1}^{N_h^k}\eta_n^{N_h^k}\left\{r_h(s_h^k,a_h^k,b_h^k) + \overline{V}_{h+1}^{k^n}(s_{h+1}^{k^n}) - \overline{V}_{h+1}^{\mathrm{R},k^n}(s_{h+1}^{k^n})\right\}$$

$$+\sum_{n=1}^{N_h^k}\eta_n^{N_h^k}\left\{\overline{\phi}_h^{\mathrm{r},k^n+1}(s_h^k,a_h^k,b_h^k) + \overline{b}_h^{\mathrm{R},k^n+1} - Q^{\dagger,\nu_h^{k'+1}}(s_h^k,a_h^k,b_h^k)\right\}. \tag{70}$$

To continue, invoking the Bellman optimality equation (63) and using the construction of $\overline{\phi}_h^{\mathrm{r}}$ in line 12 of Algorithm 2 (which is the running mean of $\overline{V}_{h+1}^{\mathrm{R}}$), we reach

$$r_h(s_h^k,a_h^k,b_h^k) + \overline{V}_{h+1}^{k^n}(s_{h+1}^{k^n}) - \overline{V}_{h+1}^{\mathrm{R},k^n}(s_{h+1}^{k^n}) + \overline{\phi}_h^{\mathrm{r},k^n+1}(s_h^k,a_h^k,b_h^k) - Q^{\dagger,\nu_h^{k'+1}}(s_h^k,a_h^k,b_h^k)$$

$$= \overline{V}_{h+1}^{k^n}(s_{h+1}^{k^n}) - \overline{V}_{h+1}^{\mathrm{R},k^n}(s_{h+1}^{k^n}) + \frac{\sum_{i=1}^n \overline{V}_{h+1}^{\mathrm{R},k^i}(s_{h+1}^{k^i})}{n} - P_{h,s_h^k,a_h^k,b_h^k}V_{h+1}^{\dagger,\nu_h^{k'+1}}. \tag{71}$$

Next, combined with the definition of

$$\varpi_h^{k^n} := \left(P_h^{k^n} - P_{h,s_h^k,a_h^k,b_h^k}\right)\left(\overline{V}_{h+1}^{k^n} - \overline{V}_{h+1}^{\mathrm{R},k^n}\right) + \frac{1}{n}\sum_{i=1}^n \left(P_h^{k^i} - P_{h,s_h^k,a_h^k,b_h^k}\right)\overline{V}_{h+1}^{\mathrm{R},k^i}, \tag{72}$$

we can obtain

$$(71) = P_{h,s_h^k,a_h^k,b_h^k}\left\{\overline{V}_{h+1}^{k^n} - \overline{V}_{h+1}^{\mathrm{R},k^n}\right\} + \frac{\sum_{i=1}^n P_{h,s_h^k,a_h^k,b_h^k}\left(\overline{V}_{h+1}^{\mathrm{R},k^i}\right)}{n} - P_{h,s_h^k,a_h^k,b_h^k}V_{h+1}^{\dagger,\nu_h^{k'+1}} + \varpi_h^{k^n}$$

$$= P_{h,s_h^k,a_h^k,b_h^k}\left\{\overline{V}_{h+1}^{k^n} - V_{h+1}^{\dagger,\nu_h^{k'+1}} + \frac{\sum_{i=1}^n \left(\overline{V}_{h+1}^{\mathrm{R},k^i} - \overline{V}_{h+1}^{\mathrm{R},k^n}\right)}{n}\right\} + \varpi_h^{k^n}, \tag{73}$$

with the notation $P_h^k$ defined in (15). Putting (73) into (70) together gives

$$\overline{Q}_h^{\mathrm{R},k+1}(s_h^k,a_h^k,b_h^k) - Q^{\dagger,\nu_h^{k'+1}}(s_h^k,a_h^k,b_h^k) = \eta_0^{N_h^k}\left(\overline{Q}_h^{\mathrm{R},1}(s_h^k,a_h^k,b_h^k) - Q^{\dagger,\nu_h^{k'+1}}(s_h^k,a_h^k,b_h^k)\right)$$

$$+\sum_{n=1}^{N_h^k}\eta_n^{N_h^k}\left\{P_{h,s_h^k,a_h^k,b_h^k}\left(\overline{V}_{h+1}^{k^n} - V_{h+1}^{\dagger,\nu_h^{k'+1}} + \frac{\sum_{i=1}^n \left(\overline{V}_{h+1}^{\mathrm{R},k^i} - \overline{V}_{h+1}^{\mathrm{R},k^n}\right)}{n}\right) + \overline{b}_h^{\mathrm{R},k^n+1} + \varpi_h^{k^n}\right\}. \tag{74}$$

**Step 2: two key quantities for lower bounding** $\overline{Q}_h^{\mathrm{R},k+1}(s_h^k, a_h^k, b_h^k) - Q^{\dagger, \nu_h^{k'+1}}(s_h^k, a_h^k, b_h^k)$. In order to develop a lower bound on $\overline{Q}_h^{\mathrm{R},k+1}(s_h^k, a_h^k, b_h^k) - Q^{\dagger, \nu_h^{k'+1}}(s_h^k, a_h^k, b_h^k)$ based on the decomposition (74), we make note of several simple facts as follows.

(a) The initialization satisfies $\overline{Q}_h^{\mathrm{R},1}(s_h^k, a_h^k, b_h^k) - Q^{\dagger, \nu_h^{k'+1}}(s_h^k, a_h^k, b_h^k) \geq 0$.

(b) For any $1 \leq k^n \leq k'$, one has

$$\overline{V}_{h+1}^{k^n} \geq \overline{V}_{h+1}^{k'+1} \geq V_{h+1}^{\dagger, \nu_h^{k'+1}}, \tag{75}$$

owing to the induction hypotheses of $Q$-values in the $(h+1)$-th step and (53) that holds up to $k$.

(c) For all $0 \leq i \leq n$ and any $s \in \mathcal{S}$, one has

$$\overline{V}_{h+1}^{\mathrm{R},k^i} - \overline{V}_{h+1}^{\mathrm{R},k^n} \geq 0, \tag{76}$$

which holds since the reference value $\overline{V}_h^{\mathrm{R}}(s)$ is monotonically non-increasing in view of the monotonicity of $\overline{V}_h(s)$ in (23) and the update rule in line 17 of Algorithm 3.

Based on the three statements above, we can simplify (74) to

$$\overline{Q}_h^{\mathrm{R},k+1}(s_h^k, a_h^k, b_h^k) - Q^{\dagger, \nu_h^{k'+1}}(s_h^k, a_h^k, b_h^k) \geq \sum_{n=1}^{N_h^k} \eta_n^{N_h^k} \left( \overline{b}_h^{\mathrm{R},k^n+1} + \varpi_h^{k^n} \right). \tag{77}$$

In the sequel, we aim to establish $\overline{Q}_h^{\mathrm{R},k+1}(s_h^k, a_h^k, b_h^k) \geq Q^{\dagger, \nu_h^{k'+1}}(s_h^k, a_h^k, b_h^k)$ based on this inequality (77). As it turns out, if one could show that

$$\left| \sum_{n=1}^{N_h^k} \eta_n^{N_h^k} \varpi_h^{k^n} \right| \leq \sum_{n=1}^{N_h^k} \eta_n^{N_h^k} \overline{b}_h^{\mathrm{R},k^n+1}, \tag{78}$$

we claim that (77) satisfies

$$\overline{Q}_h^{\mathrm{R},k+1}(s_h^k, a_h^k, b_h^k) - Q^{\dagger, \nu_h^{k'+1}}(s_h^k, a_h^k, b_h^k) \geq \sum_{n=1}^{N_h^k} \eta_n^{N_h^k} \overline{b}_h^{\mathrm{R},k^n+1} - \left| \sum_{n=1}^{N_h^k} \eta_n^{N_h^k} \varpi_h^{k^n} \right| \geq 0. \tag{79}$$

where the first inequality comes from the triangle inequality. As a result, the remaining steps come down to justifying the assumption (78). And thus, we need to control the following two quantities (in view of (72)

$$W_1 := \sum_{n=1}^{N_h^k} \eta_n^{N_h^k} \left( P_h^{k^n} - P_{h, s_h^k, a_h^k, b_h^k} \right) \left( \overline{V}_{h+1}^{k^n} - \overline{V}_{h+1}^{\mathrm{R},k^n} \right), \tag{80a}$$

$$W_2 := \sum_{n=1}^{N_h^k} \frac{1}{n} \eta_n^{N_h^k} \sum_{i=1}^{n} \left( P_h^{k^i} - P_{h, s_h^k, a_h^k, b_h^k} \right) \overline{V}_{h+1}^{\mathrm{R},k^i} \tag{80b}$$

separately, which constitute the next two steps. As will be seen momentarily, these two terms can be controlled in a similar fashion using Freedman's inequality.

**Step 3: controlling $W_1$.** In the following, we intend to invoke Lemma 7 to control term $W_1$ defined in (80a). To begin with, consider any $(N, h) \in [K] \times [H]$, and introduce

$$F_{h+1}^k := \overline{V}_{h+1}^k - \overline{V}_{h+1}^{\mathrm{R},k} \quad \text{and} \quad G_h^k(s, a, b, N) := \eta_{N_h^k(s,a,b)}^N \geq 0. \tag{81}$$

Accordingly, we can derive and define the following two equations:

$$\|F_{h+1}^k\|_\infty \leq \|\overline{V}_{h+1}^k\|_\infty + \|\overline{V}_{h+1}^{\mathrm{R},k}\|_\infty \leq 2H =: C_{\mathrm{f}}, \tag{82}$$

$$\max_{N,h,s,a,b \in [K] \times [H] \times \mathcal{S} \times \mathcal{A} \times \mathcal{B}} \eta_{N_h^k(s,a,b)}^N \leq \frac{2H}{N} =: C_{\mathrm{g}}, \tag{83}$$

where the last inequality comes from

$$\eta_{N_h^k(s,a,b)}^N \le \frac{2H}{N}, \qquad \text{if } 1 \le N_h^k(s,a,b) \le N;$$
$$\eta_{N_h^k(s,a,b)}^N = 0, \qquad \text{if } N_h^k(s,a,b) > N,$$

along with Lemma 1 and the definition in (17). Moreover, observed from (18), we have

$$0 \le \sum_{n=1}^N G_h^{k_h^n(s,a,b)}(s,a,b,N) = \sum_{n=1}^N \eta_n^N \le 1 \tag{84}$$

holds for all $(N,s,a,b) \in [K] \times \mathcal{S} \times \mathcal{A} \times \mathcal{B}$. Therefore, applying Lemma 7 with the quantities (81) and $(N,s,a,b) = (N_h^k, s_h^k, a_h^k, b_h^k)$ implies that, with probability at least $1-\delta$

$$|W_1| = \left| \sum_{n=1}^{N_h^k} \eta_n^{N_h^k} \big( P_h^{k^n} - P_{h,s_h^k,a_h^k,b_h^k} \big) \big( \overline{V}_{h+1}^{k^n} - \overline{V}_{h+1}^{\mathrm{R},k^n} \big) \right|$$

$$\lesssim \sqrt{C_{\mathrm{g}} \log^2 \frac{SABT}{\delta}} \sqrt{\sum_{n=1}^{N_h^k} G_h^{k^n}(s_h^k, a_h^k, b_h^k, N_h^k) \mathsf{Var}_{h,s_h^k,a_h^k,b_h^k}\big( W_{h+1}^{k^n} \big)}$$

$$+ \left( C_{\mathrm{g}} C_{\mathrm{f}} + \sqrt{\frac{C_{\mathrm{g}}}{N}} C_{\mathrm{f}} \right) \log^2 \frac{SABT}{\delta}$$

$$\asymp \sqrt{\frac{H}{N_h^k} \log^2 \frac{SABT}{\delta}} \sqrt{\sum_{n=1}^{N_h^k} \eta_n^{N_h^k} \mathsf{Var}_{h,s_h^k,a_h^k,b_h^k}\big( \overline{V}_{h+1}^{k^n} - \overline{V}_{h+1}^{\mathrm{R},k^n} \big)} + \frac{H^2 \log^2 \frac{SABT}{\delta}}{N_h^k}, \tag{85}$$

where the second equation comes from $X_i(s_h^k, a_h^k, b_h^k, h, N_h^k) = \eta_n^{N_h^k} \big( P_h^{k^n} - P_{h,s_h^k,a_h^k,b_h^k} \big) \big( \overline{V}_{h+1}^{k^n} - \overline{V}_{h+1}^{\mathrm{R},k^n} \big)$. Furthermore, (85) can be simplified to

$$(85) \lesssim \sqrt{\frac{H}{N_h^k} \log^2 \frac{SABT}{\delta}} \sqrt{\overline{\psi}_h^{\mathrm{a},k^{N_h^k}+1}(s_h^k, a_h^k, b_h^k) - \big( \overline{\phi}_h^{\mathrm{a},k^{N_h^k}+1}(s_h^k, a_h^k, b_h^k) \big)^2} + \frac{H^2 \log^2 \frac{SABT}{\delta}}{(N_h^k)^{3/4}}, \tag{86}$$

where the proof (86) is postponed to Appendix C.1.1 to streamline the presentation.

**Step 4: controlling $W_2$.** In the following, our aim is to the quantity $W_2$ defined in (80b). Rearranging terms in (80b), we obtain

$$W_2 = \sum_{n=1}^{N_h^k} \eta_n^{N_h^k} \frac{\sum_{i=1}^n \big( P_h^{k^i} - P_{h,s_h^k,a_h^k} \big) \overline{V}_{h+1}^{\mathrm{R},k^i}}{n} = \sum_{i=1}^{N_h^k} \left( \sum_{n=i}^{N_h^k} \frac{\eta_n^{N_h^k}}{n} \right) \big( P_h^{k^i} - P_{h,s_h^k,a_h^k} \big) \overline{V}_{h+1}^{\mathrm{R},k^i},$$

which can again be controlled by Lemma 7. And thus, we abuse the notation by taking

$$F_{h+1}^k := \overline{V}_{h+1}^{\mathrm{R},k^i} \qquad \text{and} \qquad G_h^k(s,a,b,N) := \sum_{n=N_h^i(s,a,b)}^N \frac{\eta_n^N}{n} \ge 0. \tag{87}$$

These quantities satisfy

$$\big\| F_{h+1}^k \big\|_\infty \le \big\| \overline{V}_{h+1}^{\mathrm{R},i} \big\|_\infty \le H =: C_{\mathrm{f}}, \tag{88}$$

$$\max_{N,h,s,a,b \in [K] \times [H] \times \mathcal{S} \times \mathcal{A} \times \mathcal{B}} \sum_{n=N_h^i(s,a,b)}^N \frac{\eta_n^N}{n} \le \sum_{n=1}^N \frac{\eta_n^N}{n} \le \frac{2}{N} =: C_{\mathrm{g}}. \tag{89}$$

where (89) holds according to Lemma 1. Then for all $(N,s,a,b) \in [K] \times \mathcal{S} \times \mathcal{A} \times \mathcal{B}$, it is readily seen from (89) that

$$0 \le \sum_{n=1}^N G_h^{k_h^n(s,a,b)}(s,a,b,N) \le \sum_{n=1}^N \frac{2}{N} \le 2. \tag{90}$$

With the above relations in mind, taking $(N, s, a, b) = (N_h^k, s_h^k, a_h^k, b_h^k)$ and applying Lemma 7 w.r.t. quantities (87) reveals that, with probability exceeding $1 - \delta$

$$|W_2| = \left| \sum_{i=1}^{N_h^k} \sum_{n=i}^{N_h^k} \frac{\eta_n^{N_h^k}}{n} \big( P_h^{k^i} - P_{h, s_h^k, a_h^k, b_h^k} \big) \overline{V}_{h+1}^{R, i} \right| = \left| \sum_{i=1}^{k} X_i(s_h^k, a_h^k, b_h^k, h, N_h^k) \right| \qquad (91)$$

$$\lesssim \sqrt{C_{\mathrm{g}} \log^2 \frac{SABT}{\delta}} \sqrt{\sum_{n=1}^{N_h^k} G_h^{k^n}(s_h^k, a_h^k, b_h^k, N_h^k) \mathsf{Var}_{h, s_h^k, a_h^k, b_h^k}\big( F_{h+1}^{k^n} \big)}$$

$$+ \left( C_{\mathrm{g}} C_{\mathrm{f}} + \sqrt{\frac{C_{\mathrm{g}}}{N}} C_{\mathrm{f}} \right) \log^2 \frac{SABT}{\delta}$$

$$\lesssim \sqrt{\frac{1}{N_h^k} \log^2 \frac{SABT}{\delta}} \sqrt{\frac{1}{N_h^k} \sum_{n=1}^{N_h^k} \mathsf{Var}_{h, s_h^k, a_h^k, b_h^k}\big( \overline{V}_{h+1}^{R, i} \big)} + \frac{H}{N_h^k} \log^2 \frac{SABT}{\delta}, \qquad (92)$$

where the second comes from $X_i(s_h^k, a_h^k, b_h^k, h, N_h^k) = \sum_{n=i}^{N_h^k} \frac{\eta_n^{N_h^k}}{n} \big( P_h^{k^i} - P_{h, s_h^k, a_h^k, b_h^k} \big) \overline{V}_{h+1}^{R, i}$ and $k = N_h^k$. Moreover, (92) can be simplified to

$$(92) \lesssim \sqrt{\frac{1}{N_h^k} \log^2 \frac{SABT}{\delta}} \sqrt{\overline{\psi}_h^{\mathrm{r}, k^{N_h^k+1}}(s_h^k, a_h^k, b_h^k) - \big( \overline{\phi}_h^{\mathrm{r}, k^{N_h^k+1}}(s_h^k, a_h^k, b_h^k) \big)^2} + \frac{H}{(N_h^k)^{3/4}} \log^2 \frac{SABT}{\delta}, \qquad (93)$$

where the proof (93) is postponed to Appendix C.1.2 to streamline the presentation.

**Step 5: combining the above bounds.** Combining with the results in (86) and (93), we obtain an upper bound on $\big| \sum_{n=1}^{N_h^k} \eta_n^{N_h^k} \varpi_h^{k^n} \big|$ as follows:

$$\left| \sum_{n=1}^{N_h^k} \eta_n^{N_h^k} \varpi_h^{k^n} \right| \leq |W_1| + |W_2|$$

$$\lesssim \sqrt{\frac{H}{N_h^k} \log^2 \frac{SABT}{\delta}} \sqrt{\overline{\psi}_h^{\mathrm{a}, k^{N_h^k+1}}(s_h^k, a_h^k, b_h^k) - \big( \overline{\phi}_h^{\mathrm{a}, k^{N_h^k+1}}(s_h^k, a_h^k, b_h^k) \big)^2}$$

$$+ \sqrt{\frac{1}{N_h^k} \log^2 \frac{SABT}{\delta}} \sqrt{\overline{\psi}_h^{\mathrm{r}, k^{N_h^k+1}}(s_h^k, a_h^k, b_h^k) - \big( \overline{\phi}_h^{\mathrm{r}, k^{N_h^k+1}}(s_h^k, a_h^k, b_h^k) \big)^2} + \frac{H^2 \log^2 \frac{SABT}{\delta}}{(N_h^k)^{3/4}}$$

$$\leq \overline{B}_h^{\mathrm{R}, k^{N_h^k+1}}(s_h^k, a_h^k, b_h^k) + c_{\mathrm{b}} \frac{H^2 \log^2 \frac{SABT}{\delta}}{(N_h^k)^{3/4}} \qquad (94)$$

for some sufficiently large constant $c_{\mathrm{b}} > 0$, where the last inequality follows from the definition of $\overline{B}_h^{\mathrm{R}, k^{N_h^k+1}}(s_h^k, a_h^k, b_h^k)$ in line 15 of Algorithm 2.

In order to establish the desired bound (78), we still need to control the sum $\sum_{n=1}^{N_h^k} \eta_n^{N_h^k} \overline{b}_h^{\mathrm{R}, k^n+1}$. Towards this end, the definition of $\overline{b}_h^{\mathrm{R}, k^n+1}$ (resp. $\overline{\delta}_h^{\mathrm{R}}$) in line 8 (resp. line 16) of Algorithm 2 yields

$$\overline{b}_h^{\mathrm{R}, k^n+1} = \left( 1 - \frac{1}{\eta_n} \right) \overline{B}_h^{\mathrm{R}, k^n}(s_h^k, a_h^k, b_h^k) + \frac{1}{\eta_n} \overline{B}_h^{\mathrm{R}, k^n+1}(s_h^k, a_h^k, b_h^k) + \frac{c_{\mathrm{b}}}{n^{3/4}} H^2 \log^2 \frac{SABT}{\delta}. \qquad (95)$$

This taken collectively with the definition (17) of $\eta_n^N$ allows us to expand

$$\sum_{n=1}^{N_h^k} \eta_n^{N_h^k} \overline{b}_h^{\mathrm{R}, k^n+1} - c_{\mathrm{b}} \sum_{n=1}^{N_h^k} \frac{\eta_n^{N_h^k}}{n^{3/4}} H^2 \log^2 \frac{SABT}{\delta}$$

$$= \sum_{n=1}^{N_h^k} \eta_n \prod_{i=n+1}^{N_h^k} (1 - \eta_i) \left( \left( 1 - \frac{1}{\eta_n} \right) \overline{B}_h^{\mathrm{R}, k^n}(s_h^k, a_h^k, b_h^k) + \frac{1}{\eta_n} \overline{B}_h^{\mathrm{R}, k^n+1}(s_h^k, a_h^k, b_h^k) \right). \qquad (96)$$

And thus, we can reach

$$
(96) = \sum_{n=1}^{N_h^k} \prod_{i=n+1}^{N_h^k} (1-\eta_i) \left( -(1-\eta_n)\overline{B}_h^{\mathrm{R},k^n}(s_h^k, a_h^k, b_h^k) + \overline{B}_h^{\mathrm{R},k^n+1}(s_h^k, a_h^k, b_h^k) \right)
$$

$$
= \sum_{n=1}^{N_h^k} \left( \prod_{i=n+1}^{N_h^k} (1-\eta_i)\overline{B}_h^{\mathrm{R},k^n+1}(s_h^k, a_h^k, b_h^k) - \prod_{i=n}^{N_h^k}(1-\eta_i)\overline{B}_h^{\mathrm{R},k^n}(s_h^k, a_h^k, b_h^k) \right). \tag{97}
$$

Under the fact that $\overline{B}_h^{\mathrm{R},k^1}(s_h^k, a_h^k, b_h^k) = 0$, we can reach

$$
(97) = \sum_{n=1}^{N_h^k} \prod_{i=n+1}^{N_h^k} (1-\eta_i)\overline{B}_h^{\mathrm{R},k^n+1}(s_h^k, a_h^k, b_h^k) - \sum_{n=2}^{N_h^k} \prod_{i=n}^{N_h^k}(1-\eta_i)\overline{B}_h^{\mathrm{R},k^n}(s_h^k, a_h^k, b_h^k)
$$

$$
= \sum_{n=1}^{N_h^k} \prod_{i=n+1}^{N_h^k} (1-\eta_i)\overline{B}_h^{\mathrm{R},k^n+1}(s_h^k, a_h^k, b_h^k) - \sum_{n=1}^{N_h^k-1} \prod_{i=n+1}^{N_h^k}(1-\eta_i)\overline{B}_h^{\mathrm{R},k^n+1}(s_h^k, a_h^k, b_h^k)
$$

$$
= \overline{B}_h^{\mathrm{R},k^{N_h^k}+1}(s_h^k, a_h^k, b_h^k), \tag{98}
$$

where the second equality follows from the fact that

$$
\sum_{n=2}^{N_h^k}\prod_{i=n}^{N_h^k}(1-\eta_i)\overline{B}_h^{\mathrm{R},k^n}(s_h^k, a_h^k, b_h^k) = \sum_{n=1}^{N_h^k-1} \prod_{i=n+1}^{N_h^k} (1-\eta_i)\overline{B}_h^{\mathrm{R},k^{n+1}}(s_h^k, a_h^k, b_h^k)
$$

$$
= \sum_{n=1}^{N_h^k-1} \prod_{i=n+1}^{N_h^k} (1-\eta_i)\overline{B}_h^{\mathrm{R},k^n+1}(s_h^k, a_h^k, b_h^k).
$$

To be specific, the first relation can be seen by replacing $n$ with $n + 1$, and the last relation holds true since the state-action pair $(s_h^k, a_h^k, b_h^k)$ has not been visited at step $h$ between the $(k^n + 1)$-th episode and the $(k^{n+1} - 1)$-th episode. Combining the above identity (98) with the following property (see Lemma 1)

$$
\frac{1}{(N_h^k)^{3/4}} \leq \sum_{n=1}^{N_h^k} \frac{\eta_n^{N_h^k}}{n^{3/4}} \leq \frac{2}{(N_h^k)^{3/4}},
$$

we can immediately demonstrate that

$$
\overline{B}_h^{\mathrm{R},k^{N_h^k}+1}(s_h^k, a_h^k, b_h^k) + c_{\mathrm{b}}\frac{H^2 \log^2 \frac{SAT}{\delta}}{(N_h^k)^{3/4}} \leq \sum_{n=1}^{N_h^k} \eta_n^{N_h^k} b_h^{\mathrm{R},k^n+1}
$$

$$
\leq \overline{B}_h^{\mathrm{R},k^{N_h^k}+1}(s_h^k, a_h^k, b_h^k) + 2c_{\mathrm{b}}\frac{H^2 \log^2 \frac{SABT}{\delta}}{(N_h^k)^{3/4}}. \tag{99}
$$

Taking (94) and (99) collectively demonstrates that

$$
\left| \sum_{n=1}^{N_h^k} \eta_n^{N_h^k} \varpi_h^{k^n} \right| \leq \overline{B}_h^{\mathrm{R},k^{N_h^k}+1}(s_h^k, a_h^k, b_h^k) + c_{\mathrm{b}}\frac{H^2 \log^2 \frac{SABT}{\delta}}{(N_h^k)^{3/4}} \leq \sum_{n=1}^{N_h^k} \eta_n^{N_h^k}\overline{b}_h^{\mathrm{R},k^n+1} \tag{100}
$$

as claimed in (78). We have thus concluded the proof of Lemma 2 based on the argument in Step 2.

### C.1.1 PROOF OF INEQUALITY (86)

In order to establish inequality (86), it suffices to look at the following term

$$
W_3 := \sum_{n=1}^{N_h^k} \eta_n^{N_h^k} \mathsf{Var}_{h,s_h^k,a_h^k,b_h^k}\left( \overline{V}_{h+1}^{k^n} - \overline{V}_{h+1}^{\mathrm{R},k^n} \right) - \overline{\psi}_h^{\mathrm{a},k^{N_h^k}+1}(s_h^k, a_h^k, b_h^k) + \left( \overline{\phi}_h^{\mathrm{a},k^{N_h^k}+1}(s_h^k, a_h^k, b_h^k) \right)^2,
$$

$$
\tag{101}
$$

which forms the main content of this subsection.

First of all, according to the update rules of $\overline{\phi}_h^{\mathrm{a},k^{n+1}}$ and $\overline{\psi}_h^{\mathrm{a},k^{n+1}}$ in lines 13-14 of Algorithm 2, we can get

$$
\begin{aligned}
\overline{\phi}_h^{\mathrm{a},k^{n+1}}(s_h^k,a_h^k,b_h^k) &= \overline{\phi}_h^{\mathrm{a},k^n+1}(s_h^k,a_h^k,b_h^k) \\
&= (1-\eta_n)\overline{\phi}_h^{\mathrm{a},k^n}(s_h^k,a_h^k,b_h^k) + \eta_n\big(\overline{V}_{h+1}^{k^n}(s_{h+1}^{k^n}) - \overline{V}_{h+1}^{\mathrm{R},k^n}(s_{h+1}^{k^n})\big), \\
\overline{\psi}_h^{\mathrm{a},k^{n+1}}(s_h^k,a_h^k,b_h^k) &= \overline{\psi}_h^{\mathrm{a},k^n+1}(s_h^k,a_h^k,b_h^k) \\
&= (1-\eta_n)\overline{\psi}_h^{\mathrm{a},k^n}(s_h^k,a_h^k,b_h^k) + \eta_n\big(\overline{V}_{h+1}^{k^n}(s_{h+1}^{k^n}) - \overline{V}_{h+1}^{\mathrm{R},k^n}(s_{h+1}^{k^n})\big)^2.
\end{aligned}
$$

Applying this relation recursively and invoking the definitions of $\eta_n^N$ (resp. $P_h^k$) in (17) (resp. (15)), there is

$$
\overline{\phi}_h^{\mathrm{a},k^{N_h^k}+1}(s_h^k,a_h^k,b_h^k) \overset{\text{(i)}}{=} \sum_{n=1}^{N_h^k} \eta_n^{N_h^k}\big(\overline{V}_{h+1}^{k^n}(s_{h+1}^{k^n}) - \overline{V}_{h+1}^{\mathrm{R},k^n}(s_{h+1}^{k^n})\big)
$$
$$
= \sum_{n=1}^{N_h^k} \eta_n^{N_h^k} P_h^{k^n}\big(\overline{V}_{h+1}^{k^n} - \overline{V}_{h+1}^{\mathrm{R},k^n}\big), \tag{102a}
$$
$$
\overline{\psi}_h^{\mathrm{a},k^{N_h^k}+1}(s_h^k,a_h^k,b_h^k) \overset{\text{(ii)}}{=} \sum_{n=1}^{N_h^k} \eta_n^{N_h^k}\big(\overline{V}_{h+1}^{k^n}(s_{h+1}^{k^n}) - \overline{V}_{h+1}^{\mathrm{R},k^n}(s_{h+1}^{k^n})\big)^2
$$
$$
= \sum_{n=1}^{N_h^k} \eta_n^{N_h^k} P_h^{k^n}\big(\overline{V}_{h+1}^{k^n} - \overline{V}_{h+1}^{\mathrm{R},k^n}\big)^2. \tag{102b}
$$

Recognizing that $\sum_{n=1}^{N_h^k}\eta_n^{N_h^k} = 1$ (see (18), we can immediately apply Jensen's inequality to the expressions (i) and (ii) to yield

$$
\overline{\psi}_h^{\mathrm{a},k^{N_h^k}+1}(s_h^k,a_h^k,b_h^k) \geq \Big(\overline{\phi}_h^{\mathrm{a},k^{N_h^k}+1}(s_h^k,a_h^k,b_h^k)\Big)^2. \tag{103}
$$

Further, in view of definition (16), we have

$$
\mathsf{Var}_{h,s_h^k,a_h^k,b_h^k}\big(\overline{V}_{h+1}^{k^n} - \overline{V}_{h+1}^{\mathrm{R},k^n}\big) = P_{h,s_h^k,a_h^k,b_h^k}\big(\overline{V}_{h+1}^{k^n} - \overline{V}_{h+1}^{\mathrm{R},k^n}\big)^2 - \Big(P_{h,s_h^k,a_h^k,b_h^k}\big(\overline{V}_{h+1}^{k^n} - \overline{V}_{h+1}^{\mathrm{R},k^n}\big)\Big)^2,
$$

which allows one to decompose and bound $W_3$ as follows

$$
\begin{aligned}
W_3 = &\sum_{n=1}^{N_h^k} \eta_n^{N_h^k} P_{h,s_h^k,a_h^k,b_h^k}\big(\overline{V}_{h+1}^{k^n} - \overline{V}_{h+1}^{\mathrm{R},k^n}\big)^2 - \sum_{n=1}^{N_h^k} \eta_n^{N_h^k} P_h^{k^n}\big(\overline{V}_{h+1}^{k^n} - \overline{V}_{h+1}^{\mathrm{R},k^n}\big)^2 \\
&+ \bigg(\sum_{n=1}^{N_h^k} \eta_n^{N_h^k} P_h^{k^n}\big(\overline{V}_{h+1}^{k^n} - \overline{V}_{h+1}^{\mathrm{R},k^n}\big)\bigg)^2 - \sum_{n=1}^{N_h^k} \eta_n^{N_h^k}\Big(P_{h,s_h^k,a_h^k,b_h^k}\big(\overline{V}_{h+1}^{k^n} - \overline{V}_{h+1}^{\mathrm{R},k^n}\big)\Big)^2.
\end{aligned} \tag{104}
$$

And thus, (104) reaches to

$$
W_3 \leq W_{3,1} + W_{3,2} \tag{105}
$$

where

$$
W_{3,1} = \bigg|\sum_{n=1}^{N_h^k} \eta_n^{N_h^k}\big(P_h^{k^n} - P_{h,s_h^k,a_h^k,b_h^k}\big)\big(\overline{V}_{h+1}^{k^n} - \overline{V}_{h+1}^{\mathrm{R},k^n}\big)^2\bigg|
$$

and

$$
W_{3,2} = \bigg(\sum_{n=1}^{N_h^k} \eta_n^{N_h^k} P_h^{k^n}\big(\overline{V}_{h+1}^{k^n} - \overline{V}_{h+1}^{\mathrm{R},k^n}\big)\bigg)^2 - \sum_{n=1}^{N_h^k} \eta_n^{N_h^k}\Big(P_{h,s_h^k,a_h^k,b_h^k}\big(\overline{V}_{h+1}^{k^n} - \overline{V}_{h+1}^{\mathrm{R},k^n}\big)\Big)^2.
$$

It then boils down to control the above two terms in (105) separately.

**Step 1: bounding $W_{3,1}$.** To upper bound the term $W_{3,1}$ in (104), we resort to Lemma 7 by setting

$$F_{h+1}^k := \left(\overline{V}_{h+1}^i - \overline{V}_{h+1}^{\mathrm{R},i}\right)^2 \qquad \text{and} \qquad G_h^i(s, a, N) := \eta_{N_h^i(s,a,b)}^N. \tag{106}$$

According to (83), it is easily seen that

$$\left\|F_{h+1}^k\right\|_\infty \leq \left(\left\|\overline{V}_{h+1}^{\mathrm{R},i}\right\|_\infty + \left\|\overline{V}_{h+1}^i\right\|_\infty\right)^2 \leq 4H^2 =: C_{\mathrm{f}}, \tag{107}$$

$$\max_{N,h,s,a,b\in[K]\times[H]\times\mathcal{S}\times\mathcal{A}\times\mathcal{B}} \eta_{N_h^i(s,a,b)}^N \leq \frac{2H}{N} =: C_{\mathrm{g}}. \tag{108}$$

Armed with the properties (107) and (108) and recalling (90), we can invoke Lemma 7 w.r.t. (106), $X_i(s_h^k, a_h^k, b_h^k, h, N_h^k) = \eta_n^{N_h^k}\left(P_h^{k^n} - P_{h,s_h^k,a_h^k,b_h^k}\right)\left(\overline{V}_{h+1}^{k^n} - \overline{V}_{h+1}^{\mathrm{R},k^n}\right)^2$, and set $(N,s,a) = (N_h^k, s_h^k, a_h^k, b_h^k)$ to yield, with probability at least $1 - \delta$

$$W_{3,1} = \left|\sum_{n=1}^{N_h^k} \eta_n^{N_h^k}\left(P_h^{k^n} - P_{h,s_h^k,a_h^k,b_h^k}\right)\left(\overline{V}_{h+1}^{k^n} - \overline{V}_{h+1}^{\mathrm{R},k^n}\right)^2\right|$$

$$\lesssim \sqrt{C_{\mathrm{g}}\log^2\frac{SABT}{\delta}}\sqrt{\sum_{n=1}^{N_h^k} G_h^{k^n}(s_h^k, a_h^k, b_h^k, N_h^k)\mathsf{Var}_{h,s_h^k,a_h^k,b_h^k}\left(W_{h+1}^{k^n}\right)} \tag{109}$$

$$+ \left(C_{\mathrm{g}}C_{\mathrm{f}} + \sqrt{\frac{C_{\mathrm{g}}}{N}}C_{\mathrm{f}}\right)\log^2\frac{SABT}{\delta}$$

$$\lesssim \sqrt{\frac{H}{N_h^k}\log^2\frac{SABT}{\delta}}\sqrt{\sum_{n=1}^{N_h^k} \eta_n^{N_h^k}\mathsf{Var}_{h,s_h^k,a_h^k,b_h^k}\left(\left(\overline{V}_{h+1}^{k^n} - \overline{V}_{h+1}^{\mathrm{R},k^n}\right)^2\right)} + \frac{H^3\log^2\frac{SABT}{\delta}}{N_h^k}. \tag{110}$$

With $\sum_{n=1}^{N_h^k}\eta_n^{N_h^k} \leq 1$ (see (18)) and the following trivial result:

$$\mathsf{Var}_{h,s_h^k,a_h^k,b_h^k}\left(\left(\overline{V}_{h+1}^{k^n} - \overline{V}_{h+1}^{\mathrm{R},k^n}\right)^2\right) \leq \left\|\left(\overline{V}_{h+1}^{k^n} - \overline{V}_{h+1}^{\mathrm{R},k^n}\right)^4\right\|_\infty \leq 16H^4. \tag{111}$$

We can further obtain

$$W_{3,1} \lesssim \sqrt{\frac{H^5}{N_h^k}\log^2\frac{SABT}{\delta}} + \frac{H^3}{N_h^k}\log^2\frac{SABT}{\delta}. \tag{112}$$

**Step 2: bounding $W_{3,2}$.** Jensen's inequality tells us that

$$\left(\sum_{n=1}^{N_h^k} \eta_n^{N_h^k} P_{h,s_h^k,a_h^k,b_h^k}\left(\overline{V}_{h+1}^{k^n} - \overline{V}_{h+1}^{\mathrm{R},k^n}\right)\right)^2$$

$$= \left(\sum_{n=1}^{N_h^k} \left(\eta_n^{N_h^k}\right)^{1/2}\cdot\left(\eta_n^{N_h^k}\right)^{1/2} P_{h,s_h^k,a_h^k,b_h^k}\left(\overline{V}_{h+1}^{k^n} - \overline{V}_{h+1}^{\mathrm{R},k^n}\right)\right)^2$$

$$\leq \left\{\sum_{n=1}^{N_h^k}\eta_n^{N_h^k}\right\}\left\{\sum_{n=1}^{N_h^k}\eta_n^{N_h^k}\left(P_{h,s_h^k,a_h^k,b_h^k}\left(\overline{V}_{h+1}^{k^n} - \overline{V}_{h+1}^{\mathrm{R},k^n}\right)\right)^2\right\}$$

$$\leq \sum_{n=1}^{N_h^k}\eta_n^{N_h^k}\left(P_{h,s_h^k,a_h^k,b_h^k}\left(\overline{V}_{h+1}^{k^n} - \overline{V}_{h+1}^{\mathrm{R},k^n}\right)\right)^2, \tag{113}$$

where the last line arises from (18). Substitution into $W_{3,2}$ (cf. (104)) gives

$$W_{3,2} \leq \left(\sum_{n=1}^{N_h^k}\eta_n^{N_h^k} P_h^{k^n}\left(\overline{V}_{h+1}^{k^n} - \overline{V}_{h+1}^{\mathrm{R},k^n}\right)\right)^2 - \left(\sum_{n=1}^{N_h^k}\eta_n^{N_h^k} P_{h,s_h^k,a_h^k,b_h^k}\left(\overline{V}_{h+1}^{k^n} - \overline{V}_{h+1}^{\mathrm{R},k^n}\right)\right)^2$$

$$= \left\{\sum_{n=1}^{N_h^k}\eta_n^{N_h^k}\left(P_h^{k^n} - P_{h,s_h^k,a_h^k,b_h^k}\right)\left(\overline{V}_{h+1}^{k^n} - \overline{V}_{h+1}^{\mathrm{R},k^n}\right)\right\}\left\{\sum_{n=1}^{N_h^k}\eta_n^{N_h^k}\left(P_h^{k^n} + P_{h,s_h^k,a_h^k,b_h^k}\right)\left(\overline{V}_{h+1}^{k^n} - \overline{V}_{h+1}^{\mathrm{R},k^n}\right)\right\}. \tag{114}$$

In what follows, we would like to use this relation to show that

$$W_{3,2} \leq C_{32} \left\{ \sqrt{\frac{H^5}{N_h^k} \log^2 \frac{SABT}{\delta}} + \frac{H^3}{N_h^k} \log^2 \frac{SABT}{\delta} \right\} \tag{115}$$

for some universal constant $C_{32} > 0$.

If $W_{3,2} \leq 0$, then (115) holds true trivially. Consequently, it is sufficient to study the case where $W_{3,2} > 0$. To this end, we first note that the term in the first pair of curly brackets of (114 is exactly $W_1$ (see (80a)), which can be bounded by recalling (85):

$$|W_1| \lesssim \sqrt{\frac{H}{N_h^k} \log^2 \frac{SABT}{\delta}} \sqrt{\sum_{n=1}^{N_h^k} \eta_n^{N_h^k} \mathsf{Var}_{h,s_h^k,a_h^k,b_h^k} \left( \overline{V}_{h+1}^{k^n} - \overline{V}_{h+1}^{\mathrm{R},k^n} \right)} + \frac{H^2 \log^2 \frac{SABT}{\delta}}{N_h^k} \tag{116}$$

with probability at least $1 - \delta$. According to the property that

$$\mathsf{Var}_{h,s_h^k,a_h^k,b_h^k} \left( \overline{V}_{h+1}^{k^n} - \overline{V}_{h+1}^{\mathrm{R},k^n} \right) \leq \left\| \left( \overline{V}_{h+1}^{k^n} - \overline{V}_{h+1}^{\mathrm{R},k^n} \right)^2 \right\|_\infty \leq 4H^2, \tag{117}$$

And thus, (116) can be simplified as

$$|W_1| \lesssim \sqrt{\frac{H^3}{N_h^k} \log^2 \frac{SABT}{\delta}} \sqrt{\sum_{n=1}^{N_h^k} \eta_n^{N_h^k}} + \frac{H^2 \log^2 \frac{SABT}{\delta}}{N_h^k} \lesssim \sqrt{\frac{H^3}{N_h^k} \log^2 \frac{SABT}{\delta}} + \frac{H^2}{N_h^k} \log^2 \frac{SABT}{\delta}, \tag{118}$$

whereas the last inequality (118) holds as a result of the fact $\sum_{n=1}^{N_h^k} \eta_n^{N_h^k} \leq 1$ (see (18)).

Moreover, the term in the second pair of curly brackets of (114) can be bounded straightforwardly as follows

$$\left| \sum_{n=1}^{N_h^k} \eta_n^{N_h^k} \left( P_h^{k^n} + P_{h,s_h^k,a_h^k,b_h^k} \right) \left( \overline{V}_{h+1}^{k^n} - \overline{V}_{h+1}^{\mathrm{R},k^n} \right) \right|$$

$$\leq \sum_{n=1}^{N_h^k} \eta_n^{N_h^k} \left( \left\| P_h^{k^n} \right\|_1 + \left\| P_{h,s_h^k,a_h^k,b_h^k} \right\|_1 \right) \left\| \overline{V}_{h+1}^{k^n} - \overline{V}_{h+1}^{\mathrm{R},k^n} \right\|_\infty \leq 2H, \tag{119}$$

where we have made use of property (18), as well as the elementary facts $\left\| \overline{V}_{h+1}^{k^n} - \overline{V}_{h+1}^{\mathrm{R},k^n} \right\|_\infty \leq H$ and $\left\| P_h^{k^n} \right\|_1 = \left\| P_{h,s_h^k,a_h^k,b_h^k} \right\|_1 = 1$. Substituting the above two results (118) and (119) back into (114), we arrive at the bound (115) as long as $W_{3,2} > 0$. Putting all cases together, we have established the claim (115).

**Step 3: putting all this together.** To finish up, plugging the bounds (112) and (115) into (104), we can conclude that

$$W_3 \leq W_{3,1} + W_{3,2} \leq C_3 \left\{ \sqrt{\frac{H^5}{N_h^k} \log^2 \frac{SABT}{\delta}} + \frac{H^3}{N_h^k} \log^2 \frac{SABT}{\delta} \right\}$$

for some constant $C_3 > 0$. This together with definition (101) of $W_3$ results in

$$\sum_{n=1}^{N_h^k} \eta_n^{N_h^k} \mathsf{Var}_{h,s_h^k,a_h^k,b_h^k} \left( \overline{V}_{h+1}^{k^n} - \overline{V}_{h+1}^{\mathrm{R},k^n} \right)$$

$$\leq \left\{ \overline{\psi}_h^{\mathrm{a},k^{N_h^k}+1}(s_h^k, a_h^k, b_h^k) - \left( \overline{\phi}_h^{\mathrm{a},k^{N_h^k}+1}(s_h^k, a_h^k, b_h^k) \right)^2 \right\}$$

$$+ C_3 \left( \sqrt{\frac{H^5}{N_h^k} \log^2 \frac{SABT}{\delta}} + \frac{H^3}{N_h^k} \log^2 \frac{SABT}{\delta} \right),$$

which combined with the elementary inequality $\sqrt{u+v} \leq \sqrt{u} + \sqrt{v}$ for any $u, v \geq 0$ and (103) yields

$$\left\{ \sum_{n=1}^{N_h^k} \eta_n^{N_h^k} \mathsf{Var}_{h,s_h^k,a_h^k,b_h^k}\left(\overline{V}_{h+1}^{k^n} - \overline{V}_{h+1}^{\mathrm{R},k^n}\right) \right\}^{1/2}$$

$$\lesssim \left\{ \overline{\psi}_h^{\mathrm{a},k^{N_h^k}+1}(s_h^k, a_h^k, b_h^k) - \left(\overline{\phi}_h^{\mathrm{a},k^{N_h^k}+1}(s_h^k, a_h^k, b_h^k)\right)^2 \right\}^{1/2}$$

$$+ \frac{H^{5/4}}{\left(N_h^k\right)^{1/4}} \log^{1/2} \frac{SABT}{\delta} + \frac{H^{3/2}}{\left(N_h^k\right)^{1/2}} \log \frac{SABT}{\delta}.$$

Substitution into (85) establishes the desired result (86).

### C.1.2 PROOF OF INEQUALITY (93)

In order to prove inequality (93), it suffices to look at the following term

$$W_4 := \frac{1}{N_h^k} \sum_{n=1}^{N_h^k} \mathsf{Var}_{h,s_h^k,a_h^k,b_h^k}(\overline{V}_{h+1}^{\mathrm{R},k^n}) - \left(\overline{\psi}_h^{\mathrm{r},k^{N_h^k}+1}(s_h^k, a_h^k, b_h^k) - \left(\overline{\phi}_h^{\mathrm{r},k^{N_h^k}+1}(s_h^k, a_h^k, b_h^k)\right)^2\right). \quad (120)$$

In view of the update rules of $\overline{\phi}_h^{\mathrm{r},k^{n+1}}$ and $\overline{\psi}_h^{\mathrm{r},k^{n+1}}$ in lines 11-12 of Algorithm 2, we have

$$\overline{\phi}_h^{\mathrm{r},k^{n+1}}(s_h^k, a_h^k, b_h^k) = \overline{\phi}_h^{\mathrm{r},k^n+1}(s_h^k, a_h^k, b_h^k) = \left(1 - \frac{1}{n}\right)\overline{\phi}_h^{\mathrm{r},k^n}(s_h^k, a_h^k, b_h^k) + \frac{1}{n}\overline{V}_{h+1}^{\mathrm{R},k^n}(s_{h+1}^{k^n}),$$

$$\overline{\psi}_h^{\mathrm{r},k^{n+1}}(s_h^k, a_h^k, b_h^k) = \overline{\psi}_h^{\mathrm{r},k^n+1}(s_h^k, a_h^k, b_h^k) = \left(1 - \frac{1}{n}\right)\overline{\psi}_h^{\mathrm{r},k^n}(s_h^k, a_h^k, b_h^k) + \frac{1}{n}\left(\overline{V}_{h+1}^{\mathrm{R},k^n}(s_{h+1}^{k^n})\right)^2.$$

Through simple recursion, these identities together with definition (15) of $P_h^k$ lead to

$$\overline{\phi}_h^{\mathrm{r},k^{N_h^k}+1}(s_h^k, a_h^k, b_h^k) \overset{\text{(i)}}{=} \frac{1}{N_h^k} \sum_{n=1}^{N_h^k} \overline{V}_{h+1}^{\mathrm{R},k^n}(s_{h+1}^{k^n}) = \frac{1}{N_h^k} \sum_{n=1}^{N_h^k} P_h^{k^n} \overline{V}_{h+1}^{\mathrm{R},k^n}, \quad (121a)$$

$$\overline{\psi}_h^{\mathrm{r},k^{N_h^k}+1}(s_h^k, a_h^k, b_h^k) \overset{\text{(ii)}}{=} \frac{1}{N_h^k} \sum_{n=1}^{N_h^k} \left(\overline{V}_{h+1}^{\mathrm{R},k^n}(s_{h+1}^{k^n})\right)^2 = \frac{1}{N_h^k} \sum_{n=1}^{N_h^k} P_h^{k^n} \left(\overline{V}_{h+1}^{\mathrm{R},k^n}\right)^2. \quad (121b)$$

Expressions (i) and (ii) combined with Jensen's inequality give

$$\overline{\psi}_h^{\mathrm{r},k^{N_h^k}+1}(s_h^k, a_h^k, b_h^k) \geq \left(\overline{\phi}_h^{\mathrm{r},k^{N_h^k}+1}(s_h^k, a_h^k, b_h^k)\right)^2. \quad (122)$$

Taking these together with the definition

$$\mathsf{Var}_{h,s_h^k,a_h^k,b_h^k}(\overline{V}_{h+1}^{\mathrm{R},k^n}) = P_{h,s_h^k,a_h^k,b_h^k}(\overline{V}_{h+1}^{\mathrm{R},k^n})^2 - \left(P_{h,s_h^k,a_h^k,b_h^k}\overline{V}_{h+1}^{\mathrm{R},k^n}\right)^2,$$

we obtain

$$W_4 = \frac{1}{N_h^k} \sum_{n=1}^{N_h^k} \left( P_{h,s_h^k,a_h^k,b_h^k}(\overline{V}_{h+1}^{\mathrm{R},k^n})^2 - \left(P_{h,s_h^k,a_h^k,b_h^k}\overline{V}_{h+1}^{\mathrm{R},k^n}\right)^2 \right)$$

$$- \frac{1}{N_h^k} \sum_{n=1}^{N_h^k} P_h^{k^n}(\overline{V}_{h+1}^{\mathrm{R},k^n})^2 + \left(\frac{1}{N_h^k} \sum_{n=1}^{N_h^k} P_h^{k^n}\overline{V}_{h+1}^{\mathrm{R},k^n}\right)^2. \quad (123)$$

And thus, (124) reaches to

$$W_4 = W_{4,1} + W_{4,2}. \quad (124)$$

where

$$W_{4,1} = \frac{1}{N_h^k} \sum_{n=1}^{N_h^k} \left(P_{h,s_h^k,a_h^k,b_h^k} - P_h^{k^n}\right)\left(\overline{V}_{h+1}^{\mathrm{R},k^n}\right)^2$$

and

$$W_{4,2} = \left(\frac{1}{N_h^k} \sum_{n=1}^{N_h^k} P_h^{k^n}\overline{V}_{h+1}^{\mathrm{R},k^n}\right)^2 - \frac{1}{N_h^k} \sum_{n=1}^{N_h^k} \left(P_{h,s_h^k,a_h^k,b_h^k}\overline{V}_{h+1}^{\mathrm{R},k^n}\right)^2$$

In what follows, we shall bound terms $W_{4,1}$ and $W_{4,2}$ in (124) separately.

**Step 1: bounding $W_{4,1}$.** According to Lemma 7, the first term $W_{4,1}$ in (124) can be bounded in an almost identical fashion as $W_{3,1}$ in (112). Specifically, let us set

$$F_{h+1}^k := (\overline{V}_{h+1}^{\mathrm{R},k})^2 \qquad \text{and} \qquad G_h^k(s,a,b,N) := \frac{1}{N},$$

which clearly obey

$$\|F_{h+1}^k\|_\infty \le H^2 =: C_{\mathrm{f}} \qquad \text{and} \qquad |G_h^k(s,a,b,N)| = \frac{1}{N} =: C_{\mathrm{g}}.$$

Thus, for all $(N, s, a, b) \in [K] \times \mathcal{S} \times \mathcal{A} \times \mathcal{B}$, there is

$$\sum_{n=1}^N G_h^{k^n(s,a,b)}(s,a,b,N) = \sum_{n=1}^N \frac{1}{N} = 1$$

Hence, we can take $(N, s, a, b) = (N_h^k, s_h^k, a_h^k, b_h^k)$ and apply Lemma 7 to yield, with probability at least $1 - \delta$

$$
\begin{aligned}
|W_{4,1}| &= \left| \frac{1}{N_h^k} \sum_{n=1}^{N_h^k} \left( P_h^{k^n} - P_{h,s_h^k,a_h^k,b_h^k} \right) \left( \overline{V}_{h+1}^{\mathrm{R},k^n} \right)^2 \right| = \left| \sum_{k=1}^k X_k(s_h^k, a_h^k, b_h^k, h, N_h^k) \right| \\
&\lesssim \sqrt{C_{\mathrm{g}} \log^2 \frac{SABT}{\delta}} \sqrt{\sum_{n=1}^{N_h^k} G_h^{k^n}(s_h^k, a_h^k, b_h^k, N_h^k) \mathsf{Var}_{h,s_h^k,a_h^k,b_h^k} \left( W_{h+1}^{k^n} \right)} \\
&\quad + \left( C_{\mathrm{g}} C_{\mathrm{f}} + \sqrt{\frac{C_{\mathrm{g}}}{N}} C_{\mathrm{f}} \right) \log^2 \frac{SABT}{\delta} \\
&\lesssim \sqrt{\frac{H^4 \log^2 \frac{SABT}{\delta}}{N_h^k} + \frac{H^2 \log^2 \frac{SABT}{\delta}}{N_h^k}},
\end{aligned}
\tag{125}
$$

where the first inequality comes from

$$X_k(s_h^k, a_h^k, b_h^k, h, N_h^k) = \frac{1}{N_h^k} \left( P_h^{k^n} - P_{h,s_h^k,a_h^k,b_h^k} \right) \left( \overline{V}_{h+1}^{\mathrm{R},k^n} \right)^2$$

with $k = \frac{1}{N_h^k}$, and the last inequality results from the fact that

$$\mathsf{Var}_{h,s_h^k,a_h^k,b_h^k} \left( W_{h+1}^{k^n} \right) \le \left\| W_{h+1}^{k^n} \right\|_\infty^2 \le C_{\mathrm{f}}^2 = H^4.$$

**Step 2: bounding $W_{4,2}$.** We now turn to the other term $W_{4,2}$ defined in (124). Towards this, we first make the observation that

$$\left( \frac{1}{N_h^k} \sum_{n=1}^{N_h^k} P_{h,s_h^k,a_h^k,b_h^k} \overline{V}_{h+1}^{\mathrm{R},k^n} \right)^2 \le \frac{1}{N_h^k} \sum_{n=1}^{N_h^k} \left( P_{h,s_h^k,a_h^k,b_h^k} \overline{V}_{h+1}^{\mathrm{R},k^n} \right)^2, \tag{126}$$

which follows from Jensen's inequality. Based on this relation, we can upper bound $W_{4,2}$ as

$$
\begin{aligned}
W_{4,2} &\le \left( \frac{1}{N_h^k} \sum_{n=1}^{N_h^k} P_h^{k^n} \overline{V}_{h+1}^{\mathrm{R},k^n} \right)^2 - \left( \frac{1}{N_h^k} \sum_{n=1}^{N_h^k} P_{h,s_h^k,a_h^k,b_h^k} \overline{V}_{h+1}^{\mathrm{R},k^n} \right)^2 \\
&= \left\{ \frac{1}{N_h^k} \sum_{n=1}^{N_h^k} \left( P_h^{k^n} - P_{h,s_h^k,a_h^k,b_h^k} \right) \overline{V}_{h+1}^{\mathrm{R},k^n} \right\} \left\{ \frac{1}{N_h^k} \sum_{n=1}^{N_h^k} \left( P_h^{k^n} + P_{h,s_h^k,a_h^k,b_h^k} \right) \overline{V}_{h+1}^{\mathrm{R},k^n} \right\}. \tag{127}
\end{aligned}
$$

In the following, we would like to apply this relation to prove

$$W_{4,2} \le C_{42} \left( \sqrt{\frac{H^4}{N_h^k} \log^2 \frac{SABT}{\delta}} + \frac{H^2}{N_h^k} \log^2 \frac{SABT}{\delta} \right) \tag{128}$$

for some constant $C_{42} > 0$.

When $W_{4,2} \leq 0$, the claim (128) holds trivially. As a result, we shall focus on the case where $W_{4,2} > 0$. Let us begin with the term in the first pair of curly brackets of (127). Toward this, let us abuse the notation and set

$$F_{h+1}^k := \overline{V}_{h+1}^{\mathrm{R},k} \qquad \text{and} \qquad G_h^k(s,a,b,N) := \frac{1}{N},$$

which satisfy

$$\|F_{h+1}^k\|_\infty \leq H =: C_{\mathrm{f}} \qquad \text{and} \qquad |G_h^k(s,a,b,N)| = \frac{1}{N} =: C_{\mathrm{g}}.$$

Akin to our argument for bounding $W_{4,1}$, invoking Lemma 7 and setting $(N,s,a,b) = (N_h^k, s_h^k, a_h^k, b_h^k)$ imply that, with probability at least $1 - \delta$, there is

$$\left| \frac{1}{N_h^k} \sum_{n=1}^{N_h^k} (P_h^{k^n} - P_{h,s_h^k,a_h^k,b_h^k}) \overline{V}_{h+1}^{\mathrm{R},k^n} \right| \lesssim \sqrt{\frac{H^2 \log^2 \frac{SABT}{\delta}}{N_h^k}} + \frac{H \log^2 \frac{SABT}{\delta}}{N_h^k}.$$

In addition, under the fact that $\left\| \overline{V}_{h+1}^{\mathrm{R},k^n} \right\|_\infty \leq H$ and $\left\| P_h^{k^n} \right\|_1 = \left\| P_{h,s_h^k,a_h^k,b_h^k} \right\|_1 = 1$, the term in the second pair of curly brackets of (127) can be straightly bounded by

$$\left| \frac{1}{N_h^k} \sum_{n=1}^{N_h^k} \left( P_h^{k^n} + P_{h,s_h^k,a_h^k,b_h^k} \right) \overline{V}_{h+1}^{\mathrm{R},k^n} \right| \leq \frac{1}{N_h^k} \sum_{n=1}^{N_h^k} \left( \left\| P_h^{k^n} \right\|_1 + \left\| P_{h,s_h^k,a_h^k,b_h^k} \right\|_1 \right) \left\| \overline{V}_{h+1}^{\mathrm{R},k^n} \right\|_\infty \leq 2H.$$

We have thus finished the proof of the claim (128).

**Step 3: putting all pieces together.** Combining the results (125) and (128) with (124) yields

$$W_4 \leq |W_{4,1}| + W_{4,2} \leq C_4 \left\{ \sqrt{\frac{H^4}{N_h^k} \log^2 \frac{SABT}{\delta}} + \frac{H^2}{N_h^k} \log^2 \frac{SABT}{\delta} \right\}$$

for some constant $C_4 > 0$. Taking the definition (120) of $W_4$ together, this bound gives

$$\frac{1}{N_h^k} \sum_{n=1}^{N_h^k} \mathsf{Var}_{h,s_h^k,a_h^k,b_h^k}(\overline{V}_{h+1}^{\mathrm{R},k^n}) \leq \left\{ \overline{\psi}_h^{\mathrm{r},k^{N_h^k+1}}(s_h^k,a_h^k,b_h^k) - \left( \overline{\phi}_h^{\mathrm{r},k^{N_h^k+1}}(s_h^k,a_h^k,b_h^k) \right)^2 \right\}$$

$$+ C_4 \left\{ \sqrt{\frac{H^4}{N_h^k} \log^2 \frac{SABT}{\delta}} + \frac{H^2}{N_h^k} \log^2 \frac{SABT}{\delta} \right\}.$$

Invoke the elementary inequality $\sqrt{u+v} \leq \sqrt{u} + \sqrt{v}$ for any $u,v \geq 0$ and use the property (122) to obtain

$$\left( \frac{1}{N_h^k} \sum_{n=1}^{N_h^k} \mathsf{Var}_{h,s_h^k,a_h^k,b_h^k}(V_{h+1}^{\mathrm{R},k^n}) \right)^{1/2} \lesssim \left\{ \overline{\psi}_h^{\mathrm{r},k^{N_h^k+1}}(s_h^k,a_h^k,b_h^k) - \left( \overline{\phi}_h^{\mathrm{r},k^{N_h^k+1}}(s_h^k,a_h^k,b_h^k) \right)^2 \right\}^{1/2}$$

$$+ \frac{H}{(N_h^k)^{1/4}} \log^{1/2} \frac{SABT}{\delta} + \frac{H}{(N_h^k)^{1/2}} \log \frac{SABT}{\delta}.$$

Substitution into (92) directly establishes the desired result (93).

### C.2 PROOF OF LEMMA 3

Before the proof of (25), we present the following Lemma 9 for an auxiliary illustration. It is worth noting that, similar to (Yang et al., 2021, Lemma 4.2) (see also (Jin et al., 2018b, Lemma C.7)), Lemma 9 is essentially an algebraic result leveraging certain relations, w.r.t. the $Q$-value estimates.

**Lemma 9** *Assume there exists a constant $c > 0$ such that for all $(s, a, b, k, h) \in \mathcal{S} \times \mathcal{A} \times \mathcal{B} \times [K] \times [H]$, it holds that*

$$0 \le \overline{V}_h^k(s) - \underline{V}_h^k(s) \le \overline{Q}_h^k(s, a, b) - \underline{Q}_h^k(s, a, b) + \zeta_h^k$$

$$\le \eta_0^{N_h^k(s,a,b)} H + \sum_{n=1}^{N_h^k(s,a,b)} \eta_n^{N_h^k(s,a,b)} \left( \overline{V}_{h+1}^{k^n}(s_{h+1}^{k^n}) - \underline{V}_{h+1}^{k^n}(s_{h+1}^{k^n}) \right) + 4c_{\mathrm{b}} \sqrt{\frac{H^3 \log \frac{SABT}{\delta}}{N_h^k(s, a, b)}} + \zeta_h^k. \tag{129}$$

*Consider any $\varepsilon \in (0, H]$. Then for all $\beta = 1, \ldots, \lceil \log_2 \frac{H}{\varepsilon} \rceil$, one has*

$$\left| \sum_{h=1}^{H} \sum_{k=1}^{K} \mathbb{1} \left( \overline{V}_h^k(s_h^k) - \underline{V}_h^k(s_h^k) \in \left[ 2^{\beta-1} \varepsilon, 2^\beta \varepsilon \right) \right) \right| \lesssim \frac{H^6 SAB \log \frac{SABT}{\delta}}{4^\beta \varepsilon^2}. \tag{130}$$

The proof of lemma 9 will be carried in Appendix C.2.1. We first show how to justify (25) if inequality (130) holds. As can be seen, the fact (130) immediately leads to

$$\sum_{h=1}^{H} \sum_{k=1}^{K} \mathbb{1} \left( \overline{V}_h^k(s_h^k, a_h^k, b_h^k) - \underline{V}_h^k(s_h^k, a_h^k, b_h^k) > \varepsilon \right)$$

$$\lesssim \sum_{\beta=1}^{\lceil \log_2 \frac{H}{\varepsilon} \rceil} \frac{H^6 SAB \log \frac{SABT}{\delta}}{4^\beta \varepsilon^2} \le \frac{H^6 SAB \log \frac{SABT}{\delta}}{2\varepsilon^2} \tag{131}$$

as desired.

### C.2.1 PROOF OF LEMMA 9

We first return to justify the claim (129). Lemma 2 and Lemma 3 directly verify the left-hand side of (129) since

$$\overline{Q}_h^k(s, a, b) \ge Q_h^\star(s, a, b) \ge \underline{Q}_h^k(s, a, b) \qquad \text{for all } (s, a, b, k, h) \in \mathcal{S} \times \mathcal{A} \times \mathcal{B} \times [K] \times [H]. \tag{132}$$

The remainder of the proof is thus devoted to justifying the upper bound on $\overline{Q}_h^{k+1}(s, a, b) - \underline{Q}_h^{k+1}(s, a, b)$ in (129). Then we reach

$$\overline{Q}_h^{k^{N_h^k}+1}(s, a, b) - \underline{Q}_h^{k^{N_h^k}+1}(s, a, b) \le Q_h^{\mathrm{UCB}, k^{N_h^k}}(s, a, b) - Q_h^{\mathrm{LCB}, k^{N_h^k}}(s, a, b)$$

$$= (1 - \eta_{N_h^k}) \left( Q_h^{\mathrm{UCB}, k^{N_h^k}}(s, a, b) - Q_h^{\mathrm{LCB}, k^{N_h^k}}(s, a, b) \right)$$

$$+ \eta_{N_h^k} \left( \overline{V}_{h+1}^{k^{N_h^k}}(s_{h+1}^{k^{N_h^k}+1}) - \underline{V}_{h+1}^{k^{N_h^k}}(s_{h+1}^{k^{N_h^k}+1}) + \left( P_{h,s,a,b} - P_h^{k^n} \right) \left( \overline{V}_{h+1}^{k^{N_h^k}} - \underline{V}_{h+1}^{k^{N_h^k}} \right) \right),$$

where we abbreviate

$$N_h^k = N_h^k(s, a, b)$$

throughout this subsection as long as it is clear from the context.

Applying this relation recursively leads to the desired result

$$\overline{Q}_h^{k^{N_h^k}}(s, a, b) - \underline{Q}_h^{k^{N_h^k}}(s, a, b) = \eta_0^{N_h^k} \left( Q_h^{\mathrm{UCB}, 1}(s, a, b) - Q_h^{\mathrm{LCB}, 1}(s, a, b) \right)$$

$$+ \sum_{n=1}^{N_h^k} \eta_n^{N_h^k} \left( \overline{V}_{h+1}^{k^n}(s_{h+1}^{k^n+1}) - \underline{V}_{h+1}^{k^n}(s_{h+1}^{k^n}) + \left( P_{h,s,a,b} - P_h^{k^n} \right) \left( \overline{V}_{h+1}^{k^n} - \underline{V}_{h+1}^{k^n} \right) \right)$$

$$\le \eta_0^{N_h^k} H + \sum_{n=1}^{N_h^k} \eta_n^{N_h^k} \left( \overline{V}_{h+1}^{k^n}(s_{h+1}^{k^n}) - \underline{V}_{h+1}^{k^n}(s_{h+1}^{k^n}) \right) + 4c_{\mathrm{b}} \sqrt{\frac{H^3 \log \frac{SABT}{\delta}}{N_h^k}}.$$

Here, the last line is valid due to the property $\underline{Q}_h^1(s,a,b) = 0$ and $\overline{Q}_h^1(s,a,b) = H$,

$$\left| \left( P_{h,s,a,b} - P_h^{k^n} \right) \overline{V}_{h+1}^{k^{N_h^k}} \right| \le c_{\mathrm{b}} \sqrt{\frac{H^3 \log \frac{SABT}{\delta}}{N_h^k}}$$

and

$$\left| \left( P_{h,s,a,b} - P_h^{k^n} \right) \underline{V}_{h+1}^{k^{N_h^k}} \right| \le c_{\mathrm{b}} \sqrt{\frac{H^3 \log \frac{SABT}{\delta}}{N_h^k}}$$

based on the Azuma-Hoeffding inequality and a union bound, and the following fact

$$\sum_{n=1}^{N_h^k} \eta_n^{N_h^k} c_{\mathrm{b}} \sqrt{\frac{H^3 \log \frac{SABT}{\delta}}{N_h^k}} \le 2c_{\mathrm{b}} \sqrt{\frac{H^3 \log \frac{SABT}{\delta}}{N_h^k}},$$

which is an immediate consequence of the elementary property $\sum_{n=1}^{N} \frac{\eta_n^N}{\sqrt{n}} \le \frac{2}{\sqrt{N}}$ (see Lemma 1). Combined with

$$\overline{V}_h^k(s_h^k) - \underline{V}_h^k(s_h^k) \le \overline{Q}_h^{\mathrm{R},k}(s_h^k, a_h^k, b_h^k) - \underline{Q}_h^{\mathrm{R},k}(s_h^k, a_h^k, b_h^k) + \zeta_h^k,$$

this establishes the condition (129).

Afterwards we prove (130). Accounting for the difference between our algorithm and the one in Yang et al. (2021), we paraphrase (Yang et al., 2021, Definition 4.2) into the following form that is convenient for our purpose.

**Definition 3** $((C,w)$**-Sequence**$)$ *A sequence $\{w_k\}_{1 \le k \le K}$ is called a $(C,w)$-Sequence if $0 \le w_k \le w$ for all $k$ and $\sum_k w_k \le C$.*

Combining with (129), we have

$$\sum_{k=1}^{K} w_k \left( \overline{V}_h^k(s,a,b) - \underline{V}_h^k(s,a,b) \right) \le \sum_{k=1}^{K} w_k \left( \overline{Q}_h^k(s,a,b) - \underline{Q}_h^k(s,a,b) \right) + \sum_{k=1}^{K} \zeta_h^k$$

$$\le \sum_{k=1}^{K} w_k \left( \eta_0^{N_h^k(s,a,b)} H + \sum_{n=1}^{N_h^k(s,a,b)} \eta_n^{N_h^k(s,a,b)} \left( \overline{V}_{h+1}^{k^n}(s_{h+1}^{k^n}) - \underline{V}_{h+1}^{k^n}(s_{h+1}^{k^n}) \right) \right)$$

$$+ \sum_{k=1}^{K} w_k \zeta_h^k + \sum_{k=1}^{K} 4 w_k c_{\mathrm{b}} \sqrt{\frac{H^3 \log \frac{SABT}{\delta}}{N_h^k(s,a,b)}}$$

$$\le \sum_{k=1}^{K} w_k \eta_0^{N_h^k(s,a,b)} H + \sum_{k=1}^{K} w_k \sum_{n=1}^{N_h^k(s,a,b)} \eta_n^{N_h^k(s,a,b)} \left( \overline{V}_{h+1}^{k^n}(s_{h+1}^{k^n}) - \underline{V}_{h+1}^{k^n}(s_{h+1}^{k^n}) \right)$$

$$+ \sum_{k=1}^{K} w_k \zeta_h^k + 4 \sum_{k=1}^{K} w_k c_{\mathrm{b}} \sqrt{\frac{H^3 \log \frac{SABT}{\delta}}{N_h^k(s,a,b)}}. \tag{133}$$

For the first term of (133), $N_h^k(s,a,b)$ is at most once for every state-action pair, and we always have $w_k \le w$. Thus

$$\sum_{k=1}^{K} w_k \eta_0^{N_h^k(s,a,b)} H = \sum_{k=1}^{K} w_k H \mathbb{1}\{N_h^k(s,a,b) = 0\} \le wSABH. \tag{134}$$

For the second term in (133), we exchange the order of summation and obtain

$$\sum_{k=1}^{K} w_k \sum_{n=1}^{N_h^k(s,a,b)} \eta_n^{N_h^k(s,a,b)} \left( \overline{V}_{h+1}^{k^n}(s_{h+1}^{k^n}) - \underline{V}_{h+1}^{k^n}(s_{h+1}^{k^n}) \right)$$

$$= \sum_{l=1}^{K} \left( \overline{V}_{h+1}^l(s_{h+1}^l) - \underline{V}_{h+1}^l(s_{h+1}^l) \right) \sum_{j=N_h^l+1}^{N_h^K(s_h^l, a_h^l, b_h^l)} w_{k^j} \eta_{N_h^l+1}^j.$$

Then for $l \in [K]$, we let $\widetilde{w}_l = \sum_{j=N_h^l+1}^{N_h^K(s_h^l, a_h^l, b_h^l)} w_{kj} \eta_{N_h^l+1}^j$ and further simplify the above equation to be

$$
\sum_{k=1}^{K} w_k \sum_{n=1}^{N_h^k(s,a,b)} \eta_n^{N_h^k(s,a,b)} \left( \overline{V}_{h+1}^{k^n}(s_{h+1}^{k^n}) - \underline{V}_{h+1}^{k^n}(s_{h+1}^{k^n}), \right)
$$

$$
= \sum_{l=1}^{K} \widetilde{w}_l \left( \overline{V}_{h+1}^l(s,a,b) - \underline{V}_{h+1}^l(s,a,b) \right). \tag{135}
$$

Next, we use Lemma 1 to verify that $\{\widetilde{w}\}_{l \in [K]}$ is a $(C, (1 + \frac{1}{H})w)$-sequence:

$$
\widetilde{w}_l \leq w \sum_{j=N_h^l+1}^{N_h^K(s_h^l, a_h^l, b_h^l)} \eta_{N_h^l+1}^j \leq w \sum_{j \geq N_h^l+1} \eta_{N_h^l+1}^j \leq \left( 1 + \frac{1}{H} \right) w,
$$

$$
\sum_{l=1}^{K} \widetilde{w}_l = \sum_{l=1}^{K} \sum_{j=N_h^l+1}^{N_h^K(s_h^l, a_h^l, b_h^l)} w_{kj} \eta_{N_h^l+1}^j = \sum_{k=1}^{K} w_k \sum_{t=1}^{N_h^k} \eta_t^{N_h^k} = \sum_{k=1}^{K} w_k \leq C. \tag{136}
$$

For the third term of (133), we know that $\zeta_h^k$ is a martingale difference sequence (w.r.t both $h$ and $k$), and $\left| \zeta_h^k \right| \leq 4H$. Hence, by the Hoeffding inequality, we have with probability at least $1 - \delta/2$,

$$
\sum_{k=1}^{K} w_k \zeta_h^k \leq \sqrt{\sum_{k=1}^{K} w_k H^2 \log \frac{SABT}{\delta}} \leq \sqrt{CH^2 \log \frac{SABT}{\delta}}. \tag{137}
$$

The last term of (133) can be bounded by the following inequalities with respective reasons listed below:

$$
4 \sum_{k=1}^{K} w_k c_{\mathrm{b}} \sqrt{\frac{H^3 \log \frac{SABT}{\delta}}{N_h^k(s,a,b)}} = 4 \sum_{(s,a,b)} \sum_{\substack{k=1 \\ (s_h^k, a_h^k, b_h^k)=(s,a,b)}}^{K} w_k c_{\mathrm{b}} \sqrt{\frac{H^3 \log \frac{SABT}{\delta}}{N_h^k(s,a,b)}} \tag{138}
$$

$$
= 4 c_{\mathrm{b}} \sqrt{H^3 \log \frac{SABT}{\delta}} \sum_{(s,a,b)} \sum_{n=1}^{N_h^K(s,a,b)} \frac{w_{k^n}}{\sqrt{n}} \tag{139}
$$

$$
\overset{(i)}{\leq} 4 c_{\mathrm{b}} \sqrt{H^3 \log \frac{SABT}{\delta}} \sum_{(s,a,b)} \sum_{n=1}^{\lceil C_{s,a,b}/w \rceil} \frac{w}{\sqrt{n}} \tag{140}
$$

$$
\overset{(ii)}{\leq} 10 c_{\mathrm{b}} \sqrt{H^3 \log \frac{SABT}{\delta}} \sum_{(s,a,b)} \sqrt{C_{s,a,b} w} \tag{141}
$$

$$
\overset{(iii)}{\leq} 10 c_{\mathrm{b}} \sqrt{H^3 SABC w \log \frac{SABT}{\delta}}, \tag{142}
$$

where (i) follows from a rearrangement inequality with $C_{s,a,b}$ defined as $C_{s,a,b} := \sum_{i=1}^{N_h^K(s,a,b)} w_{k^n}$ and we always keep in mind that $0 < w_{k^n} \leq w$, (ii) follows from the integral conversion of $\sum_i 1/\sqrt{i}$, and (iii) is true because of Cauchy-Schwartz inequality with $\sum_{(s,a,b)} C_{s,a,b} = \sum_{k=1}^{K} w_k \leq C$.

Plugging the upper bounds of three separate terms in (134), (135), (137) and (142) back into (133) gives us

$$
\sum_{k=1}^{K} w_k \left( \overline{V}_h^k(s,a,b) - \underline{V}_h^k(s,a,b) \right) \leq wSABH + 10 c_{\mathrm{b}} \sqrt{H^3 SABC w \log \frac{SABT}{\delta}}
$$

$$
+ \sqrt{CH^2 \log \frac{SABT}{\delta}} + \sum_{l=1}^{K} \widetilde{w}_l \left( \overline{V}_{h+1}^l(s,a,b) - \underline{V}_{h+1}^l(s,a,b) \right), \tag{143}
$$

where the third term is a weighted sum of learning errors of the same format, but taken at level $h + 1$. In addition, it has weights $\{\widetilde{w}\}_{l \in [K]}$ being a $(C, (1 + \frac{1}{H})w)$-sequence. Under recursing, the above analysis will also yield

$$
\begin{aligned}
&\sum_{k=1}^{K} w_k \left( \overline{V}_h^k(s, a, b) - \underline{V}_h^k(s, a, b) \right) \\
&\leq \sum_{h'}^{H-h} \left( SABH(1 + \frac{1}{H})^{h'} w + \sqrt{CH^2 \log \frac{SABT}{\delta}} \right. \\
&\qquad \left. + 10c_{\mathrm{b}} \sqrt{H^3 SABC(1 + \frac{1}{H})^{h'} w \log \frac{SABT}{\delta}} \right) \\
&\leq H \left( SABHew + \sqrt{CH^2 \log \frac{SABT}{\delta}} + 10c_{\mathrm{b}} \sqrt{H^3 SABCew \log \frac{SABT}{\delta}} \right). \quad (144)
\end{aligned}
$$

For every $n \in [N]$, $h \in [H]$, let

$$
w_k^{(n,h)} := \mathbb{1}\left[ \left( \overline{V}_h^k - \underline{V}_h^k \right) \left( s_h^k, a_h^k, b_h^k \right) \in \left[ 2^{n-1}\varepsilon, 2^n \varepsilon \right] \right], \quad (145)
$$

$$
C^{(n,h)} := \sum_{k=1}^{K} w_k^{(n,h)} = \left| \left\{ k : \left( \overline{V}_h^k - \underline{V}_h^k \right) \left( s_h^k, a_h^k, b_h^k \right) \in \left[ 2^{n-1}\varepsilon, 2^n \varepsilon \right] \right\} \right|. \quad (146)
$$

By definition, $\forall h \in [H]$ and $n \in [N]$, $\{w_k^{(n,h)}\}_{k \in [K]}$ is a $(C^{(n,h)}, 1)$-sequence. Now we consider bounding $\sum_{k=1}^{K} w_k^{(n,h)} \left( \overline{V}_h^k - \underline{V}_h^k \right) \left( s_h^k, a_h^k, b_h^k \right)$ from both sides. On the one hand, by (144,

$$
\begin{aligned}
\sum_{k=1}^{K} w_k^{(n,h)} \left( \overline{V}_h^k - \underline{V}_h^k \right) \left( s_h^k, a_h^k, b_h^k \right) &\leq eH^2 SAB + \sqrt{H^4 C^{(n,h)} \log \frac{SABT}{\delta}} \\
&\qquad + 10c_{\mathrm{b}} \sqrt{eH^5 SABC^{(n,h)} \log \frac{SABT}{\delta}}.
\end{aligned}
$$

On the other hand, according to the definition of $w_k^{(n,h)}$,

$$
\sum_{k=1}^{K} w_k^{(n,h)} \left( \overline{V}_h^k - \underline{V}_h^k \right) \left( s_h^k, a_h^k, b_h^k \right) \geq (2^{n-1}\varepsilon) C^{(n,h)}.
$$

Combining these two sides, we obtaion the following inequality of $C^{(n,h)}$:

$$
(2^{n-1}\varepsilon) C^{(n,h)} \leq eH^2 SAB + \sqrt{H^4 C^{(n,h)} \log \frac{SABT}{\delta}} + 10c_{\mathrm{b}} \sqrt{eH^5 SABC^{(n,h)} \log \frac{SABT}{\delta}},
$$

and thus,

$$
C^{(n,h)} \lesssim \frac{H^5 SAB \log \frac{SABT}{\delta}}{4^n \varepsilon^2}.
$$

Finally, we observe that

$$
\begin{aligned}
&\left| \sum_{h=1}^{H} \sum_{k=1}^{K} \mathbb{1} \left( \overline{V}_h^k(s_h^k, a_h^k, b_h^k) - \underline{V}_h^k(s_h^k, a_h^k, b_h^k) \in \left[ 2^{\beta-1}\varepsilon, 2^\beta \varepsilon \right] \right) \right| \\
&= C^{(\beta)} = \sum_{h=1}^{H} C^{(\beta,h)} \lesssim \frac{H^6 SAB \log \frac{SABT}{\delta}}{4^\beta \varepsilon^2}. \quad (147)
\end{aligned}
$$

Thus we obtain (130) and conclude the proof of the inequality in Lemma 9.

### C.3 PROOF OF LEMMA 4

#### C.3.1 PROOF OF THE INEQUALITY (27)

Consider any state $s$ that has been visited at least once during the $K$ episodes. Throughout this proof, we shall adopt the shorthand notation

$$k^i = k_h^i(s),$$

which denotes the index of the episode in which state $s$ is visited for the $i$-th time at step $h$. Given that $\overline{V}_h(s)$ and $\overline{V}_h^{\mathrm{R}}(s)$ are only updated during the episodes with indices coming from $\{i \mid 1 \le k^i \le K\}$, it suffices to show that for any $s$ and the corresponding $1 \le k^i \le K$, the claim (27) holds in the sense that

$$\left| \overline{V}_h^{k^i+1}(s) - \overline{V}_h^{\mathrm{R},k^i+1}(s) \right| \le 2. \tag{148}$$

Towards this end, we look at three scenarios separately.

**Case 1.** Suppose that $k^i$ obeys

$$\overline{V}_h^{k^i+1}(s) - \underline{V}_h^{k^i+1}(s) > 1 \tag{149}$$

or

$$\overline{V}_h^{k^i+1}(s) - \underline{V}_h^{k^i+1}(s) \le 1 \qquad \text{and} \qquad u_{\mathrm{r}}^{k^i}(s) = \mathsf{True}. \tag{150}$$

The above conditions correspond to the ones in line 17 and line 19 of Algorithm 3, which means that $\overline{V}_h^{\mathrm{R}}$ is updated during the $k^i$-th episode. Thus, it results in

$$\overline{V}_h^{k^i+1}(s) = \overline{V}_h^{\mathrm{R},k^i+1}(s).$$

This satisfies (148) obviously.

**Case 2.** Suppose that $k^{i_0}$ is the first time such that (149) and (150) are violated, namely,

$$i_0 := \min \left\{ j \mid \overline{V}_h^{k^j+1}(s) - \underline{V}_h^{k^j+1}(s) \le 1 \text{ and } u_{\mathrm{r}}^{k^j}(s) = \mathsf{False} \right\}. \tag{151}$$

We make several observations. Firstly, combined with the update rules (lines 16-19 of Algorithm 3), the definition (151) reveals that $\overline{V}_h^{\mathrm{R}}$ has been updated in the $k^{i_0-1}$-th episode, thus indicating that

$$\overline{V}_h^{\mathrm{R},k^{i_0}}(s) = \overline{V}_h^{\mathrm{R},k^{i_0-1}+1}(s) = \overline{V}_h^{k^{i_0-1}+1}(s) = \overline{V}_h^{k^{i_0}}(s). \tag{152}$$

Additionally, $\overline{V}_h^{\mathrm{R}}(s)$ is not updated during the $k^{i_0}$-th episode according to the definition (151), namely,

$$\overline{V}_h^{\mathrm{R},k^{i_0}+1}(s) = \overline{V}_h^{\mathrm{R},k^{i_0}}(s). \tag{153}$$

Therefore, the definition of $k^{i_0}$ indicates that either (149) or (150) is satisfied in the previous episode $k^i = k^{i_0-1}$ in which $s$ was visited. If (149) is satisfied, then lines 18-19 in Algorithm 3 tell us that

$$\mathsf{True} = u_{\mathrm{r}}^{k^{i_0-1}+1}(s) = u_{\mathrm{r}}^{k^{i_0}}(s), \tag{154}$$

which, however, contradicts the assumption $u_{\mathrm{r}}^{k^{i_0}}(s) = \mathsf{False}$ in (151). Therefore, in the $k^{i_0-1}$-th episode, (150) is satisfied, thus leading to

$$\overline{V}_h^{k^{i_0}}(s) - \underline{V}_h^{k^{i_0}}(s) = \overline{V}_h^{k^{i_0-1}+1}(s) - \underline{V}_h^{k^{i_0-1}+1}(s) \le 1. \tag{155}$$

From (152), (153), and (155), we see that

$$\overline{V}_h^{\mathrm{R},k^{i_0}+1}(s) - \overline{V}_h^{k^{i_0}+1}(s) = \overline{V}_h^{\mathrm{R},k^{i_0}}(s) - \overline{V}_h^{k^{i_0}+1}(s) = \overline{V}_h^{k^{i_0}}(s) - \overline{V}_h^{k^{i_0}+1}(s) \tag{156}$$

$$\overset{\mathrm{(i)}}{\le} \overline{V}_h^{k^{i_0}}(s) - \underline{V}_h^{k^{i_0}}(s) \overset{\mathrm{(ii)}}{\le} 1, \tag{157}$$

where (i) holds since $\overline{V}_h^{k^{i_0}+1}(s) \geq V_h^\star(s) \geq \overline{V}_h^{k^{i_0}}(s)$, and (ii) follows from (155). In addition, we make note of the fact that

$$\overline{V}_h^{\mathrm{R},k^{i_0}+1}(s) - \overline{V}_h^{k^{i_0}+1}(s) = \overline{V}_h^{k^{i_0}}(s) - \overline{V}_h^{k^{i_0}+1}(s) \geq 0, \tag{158}$$

which follows from (156) and the monotonicity of $\overline{V}_h^k(s)$ in $k$. With the above results in place, we arrive at the advertised bound (148) when $i = i_0$.

**Case 3.**  Consider any $i > i_0$. It is easy to verify that

$$\overline{V}_h^{k^i+1}(s) - \underline{V}_h^{k^i+1}(s) \leq 1 \qquad \text{and} \qquad u_{\mathrm{r}}^{k^i}(s) = \mathsf{False}. \tag{159}$$

It then follows that

$$\overline{V}_h^{\mathrm{R},k^i+1}(s) \overset{(i)}{\leq} \overline{V}_h^{\mathrm{R},k^{i_0}+1}(s) \overset{(ii)}{\leq} \overline{V}_h^{k^{i_0}+1}(s) + 1 \overset{(iii)}{\leq} \underline{V}_h^{k^{i_0}+1}(s) + 2. \tag{160}$$

Here, (i) holds due to the monotonicity of $\overline{V}_h^{\mathrm{R}}$ and $\overline{V}_h^k$ (see line 15 of Algorithm 3), (ii) is a consequence of (157), (iii) comes from the definition (151) of $i_0$. Further, according to see Lemma 2 and Lemma 3, (160) can be expressed as

$$\overline{V}_h^{\mathrm{R},k^i+1}(s) \leq V_h^\star(s) + 2 \leq \overline{V}_h^{k^i+1}(s) + 2, \tag{161}$$

where the first inequality arises since $\underline{V}_h$ is a lower bound on $V_h^\star$ (see Lemma 3), and the last inequality is valid since $\overline{V}_h^{k^i+1}(s) \geq V_h^\star(s)$ (see Lemma 2). In addition, in view of the monotonicity of $\overline{V}_h^k$ (see line 15 of Algorithm 3) and the update rule in line 17 of Algorithm 3, we know that

$$\overline{V}_h^{\mathrm{R},k^i+1}(s) \geq \overline{V}_h^{k^i+1}(s).$$

The preceding two bounds taken collectively demonstrate that

$$0 \leq \overline{V}_h^{\mathrm{R},k^i+1}(s) - \overline{V}_h^{k^i+1}(s) \leq 2,$$

thus justifying (148) for this case.

Therefore, we have established (148) for all cases, and hence (27) also holds for all cases.

### C.3.2  PROOF OF THE INEQUALITY (28)

Suppose that

$$\overline{V}_h^{\mathrm{R},k}(s_h^k) - \overline{V}_h^{\mathrm{R},K}(s_h^k) \neq 0 \tag{162}$$

holds for some $k < K$. Then there are two possible scenarios to look at:

**Case 1.**  If the condition in line 16 and line 18 of Algorithm 3 are violated at step $h$ of the $k$-th episode, this means that we have

$$\overline{V}_h^{k+1}(s_h^k) - \underline{V}_h^{k+1}(s_h^k) \leq 1 \quad \text{and} \quad u_{\mathrm{r}}^k(s_h^k) = \mathsf{False} \tag{163}$$

in this case. Then for any $k' > k$, one necessarily has

$$\begin{cases} \overline{V}_h^{k'}(s_h^k) - \underline{V}_h^{k'}(s_h^k) \leq \overline{V}_h^{k+1}(s_h^k) - \underline{V}_h^{k+1}(s_h^k) \leq 1, \\ u_{\mathrm{r}}^{k'}(s_h^k) = u_{\mathrm{r}}^k(s_h^k) = \mathsf{False}, \end{cases} \tag{164}$$

where the first property makes use of the monotonicity of $\overline{V}_h^k$ and $\underline{V}_h^k$ (see (23)). In turn, Condition (164) implies that $\overline{V}_h^{\mathrm{R}}$ will no longer be updated after the $k$-th episode (see line 16 of Algorithm 3), thus indicating that

$$\overline{V}_h^{\mathrm{R},k}(s_h^k) = \overline{V}_h^{\mathrm{R},k+1}(s_h^k) = \cdots = \overline{V}_h^{\mathrm{R},K}(s_h^k). \tag{165}$$

This, however, contradicts the assumption (162.

**Case 2.** If the condition in either line 16 or line 18 of Algorithm 3 is satisfied at step $h$ of the $k$-th episode, then the update rule in line 16 of Algorithm 3 implies that

$$\overline{V}_h^{k+1}(s_h^k) - \underline{V}_h^{k+1}(s_h^k) > 1, \tag{166}$$

or

$$\overline{V}_h^{k+1}(s_h^k) - \underline{V}_h^{k+1}(s_h^k) \le 1 \qquad \text{and} \quad u_{\mathrm{r}}^k(s_h^k) = \mathsf{True}. \tag{167}$$

To summarize, the above argument demonstrates that (162) can only occur if either (166) or (167) holds. With the above observation in place, we can proceed with the following decomposition:

$$\sum_{h=1}^{H} \sum_{k=1}^{K} \left( \overline{V}_h^{\mathrm{R},k}(s_h^k) - \overline{V}_h^{\mathrm{R},K}(s_h^k) \right)$$

$$= \sum_{h=1}^{H} \sum_{k=1}^{K} \left( \overline{V}_h^{\mathrm{R},k}(s_h^k) - \overline{V}_h^{\mathrm{R},K}(s_h^k) \right) \mathbb{1}\left( \overline{V}_h^{\mathrm{R},k}(s_h^k) - \overline{V}_h^{\mathrm{R},K}(s_h^k) \ne 0 \right) = \omega_1 + \omega_2, \tag{168}$$

where

$$\omega_1 = \sum_{h=1}^{H} \sum_{k=1}^{K} \left( \overline{V}_h^{\mathrm{R},k}(s_h^k) - \overline{V}_h^{\mathrm{R},K}(s_h^k) \right) \mathbb{1}\left( \overline{V}_h^{k+1}(s_h^k) - \underline{V}_h^{k+1}(s_h^k) \le 1 \text{ and } u_{\mathrm{r}}^k(s_h^k) = \mathsf{True} \right)$$

and

$$\omega_2 = \sum_{h=1}^{H} \sum_{k=1}^{K} \left( \overline{V}_h^{\mathrm{R},k}(s_h^k) - \underline{V}_h^{\mathrm{R},K}(s_h^k) \right) \mathbb{1}\left( \overline{V}_h^{k}(s_h^k) - \underline{V}_h^{k}(s_h^k) > 1 \right).$$

Regarding $\omega_1$ in (168), it is readily seen that for all $s \in \mathcal{S}$,

$$\sum_{k=1}^{K} \mathbb{1}\left( \overline{V}_h^{k+1}(s) - \underline{V}_h^{k+1}(s) \le 1 \text{ and } u_{\mathrm{r}}^k(s) = \mathsf{True} \right) \le 1, \tag{169}$$

which arises since, for each $s \in \mathcal{S}$, the above condition is satisfied in at most one episode, owing to the monotonicity property of $\overline{V}_h, \underline{V}_h$ and the update rule for $u_{\mathrm{r}}$ in line 18 of Algorithm 3. As a result, one has

$$\omega_1 \le H \sum_{h=1}^{H} \sum_{k=1}^{K} \mathbb{1}\left( \overline{V}_h^{k+1}(s_h^k) - \underline{V}_h^{k+1}(s_h^k) \le 1 \text{ and } u_{\mathrm{r}}^k(s_h^k) = \mathsf{True} \right)$$

$$= H \sum_{h=1}^{H} \sum_{s \in \mathcal{S}} \sum_{k=1}^{K} \mathbb{1}\left( \overline{V}_h^{k+1}(s) - \underline{V}_h^{k+1}(s) \le 1 \text{ and } u_{\mathrm{r}}^k(s) = \mathsf{True} \right) \le H \sum_{h=1}^{H} \sum_{s \in \mathcal{S}} 1 = H^2 S,$$

where the first inequality holds since $\|\overline{V}_h^{\mathrm{R},k} - \underline{V}_h^{\mathrm{R},K}\|_\infty \le H$. Substitution into (168) yields

$$\sum_{h=1}^{H} \sum_{k=1}^{K} \left( V_h^{\mathrm{R},k}(s_h^k) - V_h^{\mathrm{R},K}(s_h^k) \right) \le H^2 S + \omega_2. \tag{170}$$

To complete the proof, it boils down to bounding the term $\omega_2$ defined in (168). To begin with, note that

$$\overline{V}_h^{\mathrm{R},K}(s_h^k) \ge V_h^\star(s_h^k) \ge \underline{V}_h^k(s_h^k),$$

where we make use of the optimism of $\overline{V}_h^{\mathrm{R},K}(s_h^k)$ stated in Lemma 2 (cf. (20)) and the pessimism of $\underline{V}_h$ in Lemma 2 (see (21)). As a result, we can obtain

$$\omega_2 \le \sum_{h=1}^{H} \sum_{k=1}^{K} \left( \overline{V}_h^k(s_h^k) - \underline{V}_h^k(s_h^k) \right) \mathbb{1}\left( \overline{V}_h^k(s_h^k) - \underline{V}_h^k(s_h^k) > 1 \right). \tag{171}$$

Further, let us make note of the following elementary identity

$$\overline{V}_h^k(s_h^k, a_h^k, b_h^k) - \underline{V}_h^k(s_h^k, a_h^k, b_h^k) = \int_0^\infty \mathbb{1}\Big(\overline{V}_h^k(s_h^k, a_h^k, b_h^k) - \underline{V}_h^k(s_h^k, a_h^k, b_h^k) > t\Big)\mathrm{d}t.$$

This allows us to obtain

$$\omega_2 \le \sum_{h=1}^H \sum_{k=1}^K \left\{ \left\{ \int_0^\infty \mathbb{1}\big(\overline{V}_h^k(s_h^k, a_h^k, b_h^k) - \underline{V}_h^k(s_h^k, a_h^k, b_h^k) > t\big)\mathrm{d}t \right\} \right.$$

$$\left. \times \mathbb{1}\Big(\overline{V}_h^k(s_h^k, a_h^k, b_h^k) - \underline{V}_h^k(s_h^k, a_h^k, b_h^k) > 1\Big) \right\}$$

$$= \int_1^H \sum_{h=1}^H \sum_{k=1}^K \mathbb{1}\Big(\overline{V}_h^k(s_h^k, a_h^k, b_h^k) - \underline{V}_h^k(s_h^k, a_h^k, b_h^k) > t\Big)\mathrm{d}t.$$

With the property (25) in Lemma 3, we can further obtain

$$\omega_2 \lesssim \int_1^H \frac{H^6 SAB \log \frac{SABT}{\delta}}{t^2}\mathrm{d}t \lesssim H^6 SAB \log \frac{SABT}{\delta}. \tag{172}$$

Combining the above bounds (171) and (172) with (170) establishes

$$\sum_{h=1}^H \sum_{k=1}^K \Big(\overline{V}_h^{\mathrm{R},k}(s_h^k) - \overline{V}_h^{\mathrm{R},K}(s_h^k)\Big) \lesssim H^2 S + H^6 SAB \log \frac{SABT}{\delta} \lesssim H^6 SAB \log \frac{SABT}{\delta}$$

as claimed.

## D  PROOF OF LEMMA 5

A starting point for proving this lemma is the upper bound already derived in (31), and we intend to further bound the first term on the right-hand side of (31). Recalling the expression of $\overline{Q}_h^{\mathrm{R},k+1}(s_h^k, a_h^k, b_h^k)$ in (70) and (71), we can derive

$$\overline{Q}_h^{\mathrm{R},k}(s_h^k, a_h^k, b_h^k) - Q_h^\star(s_h^k, a_h^k, b_h^k) = \overline{Q}_h^{\mathrm{R},k^{N_h^{k-1}(s_h^k,a_h^k,b_h^k)}+1}(s_h^k, a_h^k, b_h^k) - Q_h^\star(s_h^k, a_h^k, b_h^k) \tag{173}$$

$$= \eta_0^{N_h^{k-1}(s_h^k,a_h^k,b_h^k)}\Big(\overline{Q}_h^{\mathrm{R},1}(s_h^k, a_h^k, b_h^k) - Q_h^\star(s_h^k, a_h^k, b_h^k)\Big) + \sum_{n=1}^{N_h^{k-1}(s_h^k,a_h^k,b_h^k)} \eta_n^{N_h^{k-1}(s_h^k,a_h^k,b_h^k)}\overline{b}_h^{\mathrm{R},k^n+1}$$

$$+ \sum_{n=1}^{N_h^{k-1}(s_h^k,a_h^k,b_h^k)} \eta_n^{N_h^{k-1}(s_h^k,a_h^k,b_h^k)}\Big(\overline{V}_{h+1}^{k^n}(s_{h+1}^{k^n}) - \overline{V}_{h+1}^{\mathrm{R},k^n}(s_{h+1}^{k^n})\Big)$$

$$+ \sum_{n=1}^{N_h^{k-1}(s_h^k,a_h^k,b_h^k)} \eta_n^{N_h^{k-1}(s_h^k,a_h^k,b_h^k)}\Big(\frac{1}{n}\sum_{i=1}^n \overline{V}_{h+1}^{\mathrm{R},k^i}(s_{h+1}^{k^i}) - P_{h,s_h^k,a_h^k,b_h^k}V_{h+1}^\star\Big).$$

Specifically, according to (99) with $\overline{B}_h^{\mathrm{R},k^{N_h^{k-1}}+1} = \overline{B}_h^{\mathrm{R},k}$ and the initialization $\overline{Q}_h^{\mathrm{R},1}(s_h^k, a_h^k, b_h^k) = H$, there is

$$\overline{Q}_h^{\mathrm{R},k}(s_h^k, a_h^k, b_h^k) - Q_h^\star(s_h^k, a_h^k, b_h^k) \tag{174}$$

$$\le \eta_0^{N_h^{k-1}(s_h^k,a_h^k,b_h^k)}H + \overline{B}_h^{\mathrm{R},k}(s_h^k, a_h^k, b_h^k) + \frac{2c_{\mathrm{b}}H^2}{\big(N_h^{k-1}(s_h^k,a_h^k,b_h^k)\big)^{3/4}}\log\frac{SABT}{\delta}$$

$$+ \sum_{n=1}^{N_h^{k-1}(s_h^k,a_h^k,b_h^k)} \eta_n^{N_h^{k-1}(s_h^k,a_h^k,b_h^k)}\Big(\overline{V}_{h+1}^{k^n}(s_{h+1}^{k^n}) - \overline{V}_{h+1}^{\mathrm{R},k^n}(s_{h+1}^{k^n})\Big)$$

$$+ \sum_{n=1}^{N_h^{k-1}(s_h^k,a_h^k,b_h^k)} \eta_n^{N_h^{k-1}(s_h^k,a_h^k,b_h^k)}\Big(\frac{1}{n}\sum_{i=1}^n \overline{V}_{h+1}^{\mathrm{R},k^i}(s_{h+1}^{k^i}) - P_{h,s_h^k,a_h^k,b_h^k}V_{h+1}^\star\Big).$$

We can also demonstrate

$$\underline{Q}_h^{\mathrm{R},k}(s_h^k,a_h^k,b_h^k) - Q_h^\star(s_h^k,a_h^k,b_h^k) = \underline{Q}_h^{\mathrm{R},k^{N_h^{k-1}(s_h^k,a_h^k,b_h^k)}+1}(s_h^k,a_h^k,b_h^k) - Q_h^\star(s_h^k,a_h^k,b_h^k) \quad (175)$$

$$= \eta_0^{N_h^{k-1}(s_h^k,a_h^k,b_h^k)}\left(\underline{Q}_h^{\mathrm{R},1}(s_h^k,a_h^k,b_h^k) - Q_h^\star(s_h^k,a_h^k,b_h^k)\right) - \sum_{n=1}^{N_h^{k-1}(s_h^k,a_h^k,b_h^k)} \eta_n^{N_h^{k-1}(s_h^k,a_h^k,b_h^k)}\underline{b}_h^{\mathrm{R},k^n+1}$$

$$+ \sum_{n=1}^{N_h^{k-1}(s_h^k,a_h^k,b_h^k)} \eta_n^{N_h^{k-1}(s_h^k,a_h^k,b_h^k)}\left(\underline{V}_{h+1}^{k^n}(s_{h+1}^{k^n}) - \underline{V}_{h+1}^{\mathrm{R},k^n}(s_{h+1}^{k^n})\right)$$

$$+ \sum_{n=1}^{N_h^{k-1}(s_h^k,a_h^k,b_h^k)} \eta_n^{N_h^{k-1}(s_h^k,a_h^k,b_h^k)}\left(\frac{1}{n}\sum_{i=1}^n \underline{V}_{h+1}^{\mathrm{R},k^i}(s_{h+1}^{k^i}) - P_{h,s_h^k,a_h^k,b_h^k}V_{h+1}^\star\right).$$

Similarly with $\underline{B}_h^{\mathrm{R},k^{N_h^{k-1}}+1} = \underline{B}_h^{\mathrm{R},k}$ and the initialization $\underline{Q}_h^{\mathrm{R},1}(s_h^k,a_h^k,b_h^k) = H$, there is

$$\underline{Q}_h^{\mathrm{R},k}(s_h^k,a_h^k,b_h^k) - Q_h^\star(s_h^k,a_h^k,b_h^k) \quad (176)$$

$$\geq -\eta_0^{N_h^{k-1}(s_h^k,a_h^k,b_h^k)}H - \underline{B}_h^{\mathrm{R},k}(s_h^k,a_h^k,b_h^k) - \frac{2c_{\mathrm{b}}H^2}{\left(N_h^{k-1}(s_h^k,a_h^k,b_h^k)\right)^{3/4}}\log\frac{SABT}{\delta}$$

$$+ \sum_{n=1}^{N_h^{k-1}(s_h^k,a_h^k,b_h^k)} \eta_n^{N_h^{k-1}(s_h^k,a_h^k,b_h^k)}\left(\underline{V}_{h+1}^{k^n}(s_{h+1}^{k^n}) - \underline{V}_{h+1}^{\mathrm{R},k^n}(s_{h+1}^{k^n})\right)$$

$$+ \sum_{n=1}^{N_h^{k-1}(s_h^k,a_h^k,b_h^k)} \eta_n^{N_h^{k-1}(s_h^k,a_h^k,b_h^k)}\left(\frac{1}{n}\sum_{i=1}^n \underline{V}_{h+1}^{\mathrm{R},k^i}(s_{h+1}^{k^i}) - P_{h,s_h^k,a_h^k,b_h^k}V_{h+1}^\star\right).$$

Summing over all $1 \leq k \leq K$, taking (174) and (176) collectively demonstrates that

$$\sum_{k=1}^K (\overline{Q}_h^{\mathrm{R},k}(s_h^k,a_h^k,b_h^k) - \underline{Q}_h^{\mathrm{R},k}(s_h^k,a_h^k,b_h^k))$$

$$\leq \sum_{k=1}^K \left(2H\eta_0^{N_h^{k-1}(s_h^k,a_h^k,b_h^k)} + \overline{B}_h^{\mathrm{R},k}(s_h^k,a_h^k,b_h^k) + \underline{B}_h^{\mathrm{R},k}(s_h^k,a_h^k,b_h^k)\right)$$

$$+ \sum_{k=1}^K \left(\frac{4c_{\mathrm{b}}H^2}{\left(N_h^{k-1}(s_h^k,a_h^k,b_h^k)\right)^{3/4}}\log\frac{SABT}{\delta}\right)$$

$$+ \sum_{k=1}^K \sum_{n=1}^{N_h^{k-1}(s_h^k,a_h^k,b_h^k)} \eta_n^{N_h^{k-1}(s_h^k,a_h^k,b_h^k)}\left(\overline{V}_{h+1}^{k^n}(s_{h+1}^{k^n}) - \underline{V}_{h+1}^{k^n}(s_{h+1}^{k^n})\right)$$

$$+ \sum_{k=1}^K \sum_{n=1}^{N_h^{k-1}(s_h^k,a_h^k,b_h^k)} \eta_n^{N_h^{k-1}(s_h^k,a_h^k,b_h^k)}\left(\underline{V}_{h+1}^{\mathrm{R},k^n}(s_{h+1}^{k^n}) - \overline{V}_{h+1}^{\mathrm{R},k^n}(s_{h+1}^{k^n})\right)$$

$$+ \sum_{k=1}^K \sum_{n=1}^{N_h^{k-1}(s_h^k,a_h^k,b_h^k)} \eta_n^{N_h^{k-1}(s_h^k,a_h^k,b_h^k)}\left(\frac{1}{n}\sum_{i=1}^n \overline{V}_{h+1}^{\mathrm{R},k^i}(s_{h+1}^{k^i}) - \frac{1}{n}\sum_{i=1}^n \underline{V}_{h+1}^{\mathrm{R},k^i}(s_{h+1}^{k^i})\right). \quad (177)$$

Next, we control each term in (177) separately.

- Regarding the first term of (177), we make two observations. To begin with,

$$\sum_{k=1}^K \eta_0^{N_h^{k-1}(s_h^k,a_h^k,b_h^k)} \leq \sum_{(s,a,b)\in\mathcal{S}\times\mathcal{A}\times\mathcal{B}} \sum_{n=0}^{N_h^{K-1}(s,a,b)} \eta_0^n \leq SAB, \quad (178)$$

where the last inequality follows since $\eta_0^n = 0$ for all $n > 0$ (see (17)). Next, it is also observed that

$$
\sum_{k=1}^{K} \frac{1}{\left(N_h^{k-1}(s_h^k, a_h^k, b_h^k)\right)^{3/4}} = \sum_{(s,a,b) \in \mathcal{S} \times \mathcal{A} \times \mathcal{B}} \sum_{n=1}^{N_h^{K-1}(s,a,b)} \frac{1}{n^{3/4}}
$$
$$
\leq \sum_{(s,a,b) \in \mathcal{S} \times \mathcal{A} \times \mathcal{B}} 4\left(N_h^{K-1}(s,a,b)\right)^{1/4} \leq 4(SAB)^{3/4} K^{1/4},
$$

(179)

where the last inequality comes from Holder's inequality

$$
\sum_{(s,a,b) \in \mathcal{S} \times \mathcal{A} \times \mathcal{B}} \left(N_h^{K-1}(s,a,b)\right)^{1/4}
$$
$$
\leq \left[ \sum_{(s,a,b) \in \mathcal{S} \times \mathcal{A} \times \mathcal{B}} 1 \right]^{3/4} \left[ \sum_{(s,a,b) \in \mathcal{S} \times \mathcal{A} \times \mathcal{B}} N_h^{K-1}(s,a,b) \right]^{1/4} \leq (SAB)^{3/4} K^{1/4}.
$$

Combining above bounds yields

$$
\sum_{k=1}^{K} \left( 2H \eta_0^{N_h^{k-1}(s_h^k, a_h^k, b_h^k)} + \overline{B}_h^{\mathrm{R},k}\left(s_h^k, a_h^k, b_h^k\right) + \underline{B}_h^{\mathrm{R},k}\left(s_h^k, a_h^k, b_h^k\right) \right)
$$
$$
+ \sum_{k=1}^{K} \frac{4c_b H^2}{\left(N_h^{k-1}(s_h^k, a_h^k, b_h^k)\right)^{3/4}} \log \frac{SABT}{\delta}
$$
$$
\leq 2HSAB + \overline{B}_h^{\mathrm{R},k}\left(s_h^k, a_h^k, b_h^k\right) + \underline{B}_h^{\mathrm{R},k}\left(s_h^k, a_h^k, b_h^k\right) + 16c_b (SAB)^{3/4} K^{1/4} H^2 \log \frac{SABT}{\delta}.
$$

(180)

- We now turn to the second term of (177). A little algebra gives

$$
\sum_{k=1}^{K} \sum_{n=1}^{N_h^{k-1}(s_h^k, a_h^k, b_h^k)} \eta_n^{N_h^{k-1}(s_h^k, a_h^k, b_h^k)} \left( \overline{V}_{h+1}^{k^n}\left(s_{h+1}^{k^n}\right) - \underline{V}_{h+1}^{k^n}\left(s_{h+1}^{k^n}\right) \right)
$$
$$
= \sum_{l=1}^{K} \sum_{N=N_h^l(s_h^l, a_h^l, b_h^l)}^{N_h^{K-1}(s_h^l, a_h^l, b_h^l)} \eta_{N_h^l(s_h^l, a_h^l, b_h^l)}^{N} \left( \overline{V}_{h+1}^l(s_{h+1}^l) - \underline{V}_{h+1}^l(s_{h+1}^l) \right),
$$

(181)

where the second line replaces $k^n$ (resp. $n$) with $l$ (resp. $N_h^l(s_h^l, a_h^l)$). Combined with the property $\sum_{N=n}^{\infty} \eta_n^N \leq 1 + 1/H$ (see Lemma 1), we get

$$
(181) \leq \left(1 + \frac{1}{H}\right) \sum_{l=1}^{K} \left( \overline{V}_{h+1}^l(s_{h+1}^l) - \underline{V}_{h+1}^l(s_{h+1}^l) \right)
$$
$$
= \left(1 + \frac{1}{H}\right) \sum_{k=1}^{K} \left( \overline{V}_{h+1}^k(s_{h+1}^k) - \underline{V}_{h+1}^k(s_{h+1}^k) \right),
$$

(182)

where the last relation replaces $l$ with $k$.

- When it comes to the last term of (177), based on $V_{h+1}^\star \left( s_{h+1}^{k^n} \right) = P_{h,s_h^k,a_h^k,b_h^k} \overline{V}_{h+1}^{\mathrm{R},k}$, we can derive

$$
\begin{aligned}
&\sum_{k=1}^{K} \sum_{n=1}^{N_h^{k-1}(s_h^k,a_h^k,b_h^k)} \eta_n^{N_h^{k-1}(s_h^k,a_h^k,b_h^k)} \left( \underline{V}_{h+1}^{\mathrm{R},k} \left( s_{h+1}^{k^n} \right) - \overline{V}_{h+1}^{\mathrm{R},k^n} \left( s_{h+1}^{k^n} \right) \right) \\
&+ \sum_{k=1}^{K} \sum_{n=1}^{N_h^{k-1}(s_h^k,a_h^k,b_h^k)} \eta_n^{N_h^{k-1}(s_h^k,a_h^k,b_h^k)} \left( \frac{1}{n} \sum_{i=1}^{n} \overline{V}_{h+1}^{\mathrm{R},k^i} \left( s_{h+1}^{k^i} \right) - \frac{1}{n} \sum_{i=1}^{n} \underline{V}_{h+1}^{\mathrm{R},k^i} \left( s_{h+1}^{k^i} \right) \right) \\
={}&\sum_{k=1}^{K} \sum_{n=1}^{N_h^{k-1}(s_h^k,a_h^k,b_h^k)} \eta_n^{N_h^{k-1}(s_h^k,a_h^k,b_h^k)} \left( V_{h+1}^\star \left( s_{h+1}^{k^n} \right) - \overline{V}_{h+1}^{\mathrm{R},k^n} \left( s_{h+1}^{k^n} \right) \right) \\
&+ \sum_{k=1}^{K} \sum_{n=1}^{N_h^{k-1}(s_h^k,a_h^k,b_h^k)} \eta_n^{N_h^{k-1}(s_h^k,a_h^k,b_h^k)} \left( \frac{1}{n} \sum_{i=1}^{n} \overline{V}_{h+1}^{\mathrm{R},k^i} \left( s_{h+1}^{k^i} \right) - P_{h,s_h^k,a_h^k,b_h^k} V_{h+1}^\star \right) \\
&- \sum_{k=1}^{K} \sum_{n=1}^{N_h^{k-1}(s_h^k,a_h^k,b_h^k)} \eta_n^{N_h^{k-1}(s_h^k,a_h^k,b_h^k)} \left( V_{h+1}^\star \left( s_{h+1}^{k^n} \right) - \underline{V}_{h+1}^{\mathrm{R},k^n} \left( s_{h+1}^{k^n} \right) \right) \\
&- \sum_{k=1}^{K} \sum_{n=1}^{N_h^{k-1}(s_h^k,a_h^k,b_h^k)} \eta_n^{N_h^{k-1}(s_h^k,a_h^k,b_h^k)} \left( \frac{1}{n} \sum_{i=1}^{n} \underline{V}_{h+1}^{\mathrm{R},k^i} \left( s_{h+1}^{k^i} \right) - P_{h,s_h^k,a_h^k,b_h^k} V_{h+1}^\star \right).
\end{aligned}
\tag{183}
$$

With $\Lambda_h^k = \sum_{k=1}^{K} \sum_{N=N_h^k(s_h^k,a_h^k,b_h^k)}^{N_h^{K-1}(s_h^k,a_h^k,b_h^k)} \eta_{N_h^k(s_h^k,a_h^k,b_h^k)}^N$, there is

$$
\begin{aligned}
(183) ={}& \Lambda_h^k \left( \left( P_h^k - P_{h,s_h^k,a_h^k,b_h^k} \right) \left( V_{h+1}^\star - \overline{V}_{h+1}^{\mathrm{R},k} \right) - \left( P_h^k - P_{h,s_h^k,a_h^k,b_h^k} \right) \left( V_{h+1}^\star - \underline{V}_{h+1}^{\mathrm{R},k} \right) \right) \\
&+ \Lambda_h^k \frac{\sum_{i=1}^{N_h^k(s_h^k,a_h^k,b_h^k)} \left( \overline{V}_{h+1}^{\mathrm{R},k^i} \left( s_{h+1}^{k^i} \right) - P_{h,s_h^k,a_h^k,b_h^k} \overline{V}_{h+1}^{\mathrm{R},k} \right)}{N_h^k \left( s_h^k, a_h^k, b_h^k \right)} \\
&- \Lambda_h^k \frac{\sum_{i=1}^{N_h^k(s_h^k,a_h^k,b_h^k)} \left( \underline{V}_{h+1}^{\mathrm{R},k^i} \left( s_{h+1}^{k^i} \right) - P_{h,s_h^k,a_h^k,b_h^k} \underline{V}_{h+1}^{\mathrm{R},k} \right)}{N_h^k \left( s_h^k, a_h^k, b_h^k \right)}.
\end{aligned}
$$

Here, the first equality holds since $V_{h+1}^\star(s_{h+1}^{k^n}) - V_{h+1}^{\mathrm{R},k^n}(s_{h+1}^{k^n}) = P_h^{k^n} \left( V_{h+1}^\star - V_{h+1}^{\mathrm{R},k^n} \right)$ (in view of the definition of $P_h^k$ in (15)), the final equality can be seen via simple rearrangement of the terms, while in the last line we replace $k^n$ (resp. $n$) with $k$ (resp. $N_h^k(s_h^k, a_h^k, b_h^k)$).

Taking the above bounds together with (177) and (31), we can rearrange terms to reach

$$\sum_{k=1}^{K} \left( \overline{V}_h^k(s_h^k) - \underline{V}_h^k(s_h^k) \right)$$

$$\leq \sum_{k=1}^{K} \left( \overline{Q}_h^{\mathrm{R},k}(s_h^k, a_h^k, b_h^k) - \underline{Q}_h^{\mathrm{R},k}(s_h^k, a_h^k, b_h^k) + \zeta_h^k \right)$$

$$\leq \left( 1 + \frac{1}{H} \right) \sum_{k=1}^{K} \left( \overline{V}_{h+1}^k(s_{h+1}^k) - \underline{V}_{h+1}^k(s_{h+1}^k) \right) + \overline{B}_h^{\mathrm{R},k}\left( s_h^k, a_h^k, b_h^k \right) + \underline{B}_h^{\mathrm{R},k}\left( s_h^k, a_h^k, b_h^k \right)$$

$$+ 2HSAB + 16c_{\mathrm{b}}(SAB)^{3/4}K^{1/4}H^2 \log \frac{SABT}{\delta} + \sum_{k=1}^{K} \zeta_h^k$$

$$+ \Lambda_h^k \left( \left( \left( P_h^k - P_{h,s_h^k,a_h^k,b_h^k} \right) \left( V_{h+1}^\star - \overline{V}_{h+1}^{\mathrm{R},k} \right) - \left( P_h^k - P_{h,s_h^k,a_h^k,b_h^k} \right) \left( V_{h+1}^\star - \underline{V}_{h+1}^{\mathrm{R},k} \right) \right) \right) \quad (184)$$

$$+ \Lambda_h^k \left( \frac{\sum_{i=1}^{N_h^k\left( s_h^k, a_h^k, b_h^k \right)} \left( \overline{V}_{h+1}^{\mathrm{R},k^i}\left( s_{h+1}^{k^i} \right) - P_{h,s_h^k,a_h^k,b_h^k}\overline{V}_{h+1}^{\mathrm{R},k} \right)}{N_h^k\left( s_h^k, a_h^k, b_h^k \right)} \right)$$

$$- \Lambda_h^k \left( \frac{\sum_{i=1}^{N_h^k\left( s_h^k, a_h^k, b_h^k \right)} \left( \underline{V}_{h+1}^{\mathrm{R},k^i}\left( s_{h+1}^{k^i} \right) - P_{h,s_h^k,a_h^k,b_h^k}\underline{V}_{h+1}^{\mathrm{R},k} \right)}{N_h^k\left( s_h^k, a_h^k, b_h^k \right)} \right). \quad (185)$$

Thus far, we have established a crucial connection between $\sum_{k=1}^{K} \left( \overline{V}_h^k(s_h^k) - \underline{V}_h^k(s_h^k) \right)$ at step $h$ and $\sum_{k=1}^{K} \left( \overline{V}_{h+1}^k(s_{h+1}^k) - \underline{V}_{h+1}^k(s_{h+1}^k) \right)$ at step $h+1$. Clearly, the term $\sum_{k=1}^{K} \left( \overline{V}_{h+1}^k(s_{h+1}^k) - \underline{V}_{h+1}^k(s_{h+1}^k) \right)$ can be further bounded in the same manner. As a result, by recursively applying the above relation (185) over the time steps $h = 1, 2, \cdots, H$ and using the terminal condition $\overline{V}_{H+1}^k = \underline{V}_{H+1}^k = 0$, we can immediately arrive at the advertised bound in Lemma 5.

## E   PROOF OF LEMMA 6

### E.1   BOUNDING THE TERM $\mathcal{D}_1$

First of all, let us look at the first two terms of $\mathcal{D}_1$ in (35a). Recognizing the following elementary inequality

$$\left( 1 + \frac{1}{H} \right)^{h-1} \leq \left( 1 + \frac{1}{H} \right)^H \leq e \quad \text{for all } h = 1, 2, \cdots, H+1, \quad (186)$$

we are allowed to upper bound the first two terms in (35a) as follows:

$$\sum_{h=1}^{H} \left( 1 + \frac{1}{H} \right)^{h-1} \left\{ 2HSAB + 16c_{\mathrm{b}}H^2(SAB)^{3/4}K^{1/4} \log \frac{SABT}{\delta} \right\}$$

$$\lesssim H^2 SAB + H^3(SAB)^{3/4}K^{1/4} \log \frac{SABT}{\delta} \lesssim H^{4.5}SAB \log^2 \frac{SABT}{\delta} + \sqrt{H^3 SABK}$$

$$= H^{4.5}SAB \log^2 \frac{SABT}{\delta} + \sqrt{H^2 SABT}, \quad (187)$$

where the last inequality can be shown using the AM-GM inequality as follows:

$$H^3(SAB)^{3/4}K^{1/4} \log \frac{SABT}{\delta} = \left( H^{9/4}\sqrt{SAB} \log \frac{SABT}{\delta} \right) \cdot (H^3 SABK)^{1/4}$$

$$\leq H^{4.5}SAB \log^2 \frac{SABT}{\delta} + \sqrt{H^3 SABK}.$$

We are now left with the last term of $\mathcal{D}_1$ in (35a). Towards this, we know that $\zeta_h^k$ is a martingale difference sequence (w.r.t both $h$ and $k$), and $\left|\zeta_h^k\right| \leq 4H$. Hence by the Azuma-Hoeffding inequality and a union bound, we have with probability at least $1 - \delta/2$,

$$\sum_{h=1}^H \left(1 + \frac{1}{H}\right)^{h-1} \sum_{k=1}^K \zeta_h^k \leq \sum_{h=1}^H \left(1 + \frac{1}{H}\right)^{h-1} \sqrt{HK \log \frac{SABT}{\delta}} \lesssim \sqrt{H^2 T \log \frac{SABT}{\delta}} \quad (188)$$

with probability exceeding $1 - \delta$.

Combining (187) and (188) with the definition (35a) of $\mathcal{D}_1$ immediately leads to the claimed bound.

### E.2 BOUNDING THE TERM $\mathcal{D}_2$

In view of the definition of $\overline{B}_h^{\mathrm{R},k}(s_h^k, a_h^k, b_h^k)$ (resp. $\underline{B}_h^{\mathrm{R},k}(s_h^k, a_h^k, b_h^k)$) in line 15 Algorithm 2, we can decompose $\mathcal{D}_2$ (cf. (35b)) as follows:

$$\begin{aligned}
\mathcal{D}_2 &= \sum_{h=1}^H \left(1 + \frac{1}{H}\right)^{h-1} c_{\mathrm{b}} \sqrt{H \log \frac{SABT}{\delta}} \sum_{k=1}^K \sqrt{\frac{\overline{\psi}_h^{\mathrm{a},k}(s_h^k, a_h^k, b_h^k) - (\overline{\phi}_h^{\mathrm{a},k}(s_h^k, a_h^k, b_h^k))^2}{N_h^k(s_h^k, a_h^k, b_h^k)}} \\
&\quad + \sum_{h=1}^H \left(1 + \frac{1}{H}\right)^{h-1} c_{\mathrm{b}} \sqrt{\log \frac{SABT}{\delta}} \sum_{k=1}^K \sqrt{\frac{\overline{\psi}_h^{\mathrm{r},k}(s_h^k, a_h^k, b_h^k) - (\overline{\phi}_h^{\mathrm{r},k}(s_h^k, a_h^k, b_h^k))^2}{N_h^k(s_h^k, a_h^k, b_h^k)}} \\
&\quad + \sum_{h=1}^H \left(1 + \frac{1}{H}\right)^{h-1} c_{\mathrm{b}} \sqrt{H \log \frac{SABT}{\delta}} \sum_{k=1}^K \sqrt{\frac{\underline{\psi}_h^{\mathrm{a},k}(s_h^k, a_h^k, b_h^k) - (\underline{\phi}_h^{\mathrm{a},k}(s_h^k, a_h^k, b_h^k))^2}{N_h^k(s_h^k, a_h^k, b_h^k)}} \\
&\quad + \sum_{h=1}^H \left(1 + \frac{1}{H}\right)^{h-1} c_{\mathrm{b}} \sqrt{\log \frac{SABT}{\delta}} \sum_{k=1}^K \sqrt{\frac{\underline{\psi}_h^{\mathrm{r},k}(s_h^k, a_h^k, b_h^k) - (\underline{\phi}_h^{\mathrm{r},k}(s_h^k, a_h^k, b_h^k))^2}{N_h^k(s_h^k, a_h^k, b_h^k)}} \\
&\lesssim \sqrt{H \log \frac{SABT}{\delta}} \sum_{h=1}^H \sum_{k=1}^K \sqrt{\frac{\overline{\psi}_h^{\mathrm{a},k}(s_h^k, a_h^k, b_h^k) - (\overline{\phi}_h^{\mathrm{a},k}(s_h^k, a_h^k, b_h^k))^2}{N_h^k(s_h^k, a_h^k, b_h^k)}} \\
&\quad + \sqrt{\log \frac{SABT}{\delta}} \sum_{h=1}^H \sum_{k=1}^K \sqrt{\frac{\overline{\psi}_h^{\mathrm{r},k}(s_h^k, a_h^k, b_h^k) - (\overline{\phi}_h^{\mathrm{r},k}(s_h^k, a_h^k, b_h^k))^2}{N_h^k(s_h^k, a_h^k, b_h^k)}},
\end{aligned} \quad (189)$$

where the last relation holds due to (186) and the symmetry of $[\overline{\psi}^{\mathrm{a}}, \underline{\psi}^{\mathrm{a}}]$, $[\overline{\phi}^{\mathrm{a}}, \underline{\phi}^{\mathrm{a}}]$, $[\overline{\psi}^{\mathrm{r}}, \underline{\psi}^{\mathrm{r}}]$ and $[\overline{\phi}^{\mathrm{r}}, \underline{\phi}^{\mathrm{r}}]$. In what follows, we intend to bound these two terms separately.

**Step 1: upper bounding the first term in (189).** Towards this, we make the observation that

$$\begin{aligned}
\sum_{k=1}^K \sqrt{\frac{\overline{\psi}_h^{\mathrm{a},k}(s_h^k, a_h^k, b_h^k) - (\overline{\phi}_h^{\mathrm{a},k}(s_h^k, a_h^k, b_h^k))^2}{N_h^k(s_h^k, a_h^k, b_h^k)}} &\leq \sum_{k=1}^K \sqrt{\frac{\overline{\psi}_h^{\mathrm{a},k}(s_h^k, a_h^k, b_h^k)}{N_h^k(s_h^k, a_h^k, b_h^k)}} \\
&= \sum_{k=1}^K \sqrt{\frac{\sum_{n=1}^{N_h^k(s_h^k, a_h^k, b_h^k)} \eta_n^{N_h^k(s_h^k, a_h^k, b_h^k)} \left(\overline{V}_{h+1}^{k^n}(s_{h+1}^{k^n}) - \overline{V}_{h+1}^{\mathrm{R},k^n}(s_{h+1}^{k^n})\right)^2}{N_h^k(s_h^k, a_h^k, b_h^k)}},
\end{aligned} \quad (190)$$

where the second line follows from the update rule of $\overline{\psi}_h^{\mathrm{a},k}$ in (102). Combining the relation $|\overline{V}_{h+1}^k(s_h^k) - \overline{V}_{h+1}^{\mathrm{R},k}(s_h^k)| \leq 2$ (cf. (27)) and the property $\sum_{n=1}^{N_h^k(s_h^k, a_h^k, b_h^k)} \eta_n^{N_h^k(s_h^k, a_h^k, b_h^k)} \leq 1$ (cf. (18)) with (190) yields

$$\sum_{k=1}^K \sqrt{\frac{\overline{\psi}_h^{\mathrm{a},k}(s_h^k, a_h^k, b_h^k) - (\overline{\phi}_h^{\mathrm{a},k}(s_h^k, a_h^k, b_h^k))^2}{N_h^k(s_h^k, a_h^k, b_h^k)}} \leq \sum_{k=1}^K \sqrt{\frac{4}{N_h^k(s_h^k, a_h^k, b_h^k)}} \leq 4\sqrt{SABK}. \quad (191)$$

Here, the last inequality holds due to the following fact:

$$\sum_{k=1}^{K} \sqrt{\frac{1}{N_h^k(s_h^k, a_h^k, b_h^k)}} = \sum_{(s,a,b) \in \mathcal{S} \times \mathcal{A} \times \mathcal{B}} \sum_{n=1}^{N_h^K(s,a,b)} \sqrt{\frac{1}{n}} \le 2 \sum_{(s,a,b) \in \mathcal{S} \times \mathcal{A} \times \mathcal{B}} \sqrt{N_h^K(s,a,b)}$$
$$\le 2 \sqrt{\sum_{(s,a,b) \in \mathcal{S} \times \mathcal{A} \times \mathcal{B}} 1} \cdot \sqrt{\sum_{(s,a,b) \in \mathcal{S} \times \mathcal{A} \times \mathcal{B}} N_h^K(s,a,b)} = 2\sqrt{SABK},$$
(192)

where the last line arises from Cauchy-Schwarz and the basic fact that $\sum_{(s,a,b)} N_h^K(s,a,b) = K$.

**Step 2: upper bounding the second term in (189).**   Recalling the update rules of $\overline{\phi}_h^{\mathrm{r},k}$ and $\overline{\psi}_h^{\mathrm{r},k}$ in (121), we have

$$\sum_{k=1}^{K} \sqrt{\frac{\overline{\psi}_h^{\mathrm{r},k}(s_h^k, a_h^k, b_h^k) - \left(\overline{\phi}_h^{\mathrm{r},k}(s_h^k, a_h^k, b_h^k)\right)^2}{N_h^k(s_h^k, a_h^k, b_h^k)}} = \sum_{k=1}^{K} \sqrt{\frac{1}{N_h^k(s_h^k, a_h^k, b_h^k)}} R_h^k. \quad (193)$$

with

$$R_h^k = \sqrt{\frac{\sum_{n=1}^{N_h^k(s_h^k, a_h^k, b_h^k)} \left(\overline{V}_{h+1}^{\mathrm{R},k^n}(s_{h+1}^{k^n})\right)^2}{N_h^k(s_h^k, a_h^k, b_h^k)} - \left(\frac{\sum_{n=1}^{N_h^k(s_h^k, a_h^k, b_h^k)} \overline{V}_{h+1}^{\mathrm{R},k^n}(s_{h+1}^{k^n})}{N_h^k(s_h^k, a_h^k, b_h^k)}\right)^2}.$$

Additionally, the quantity $R_h^k$ defined in (193) obeys

$$(R_h^k)^2 \le \frac{\sum_{n=1}^{N_h^k(s_h^k, a_h^k, b_h^k)} \left(\overline{V}_{h+1}^{\mathrm{R},k^n}(s_{h+1}^{k^n})\right)^2 - \left(V_{h+1}^\star(s_{h+1}^{k^n})\right)^2}{N_h^k(s_h^k, a_h^k, b_h^k)}$$
$$+ \frac{\sum_{n=1}^{N_h^k(s_h^k, a_h^k, b_h^k)} \left(V_{h+1}^\star(s_{h+1}^{k^n})\right)^2}{N_h^k(s_h^k, a_h^k, b_h^k)} - \left(\frac{\sum_{n=1}^{N_h^k(s_h^k, a_h^k, b_h^k)} V_{h+1}^\star(s_{h+1}^{k^n})}{N_h^k(s_h^k, a_h^k, b_h^k)}\right)^2 \le L_1 + L_2,$$
(194)

which arises from the fact that $H \ge \overline{V}_{h+1}^{\mathrm{R},k^n} \ge V_{h+1}^\star \ge 0$ for all $k^n \le K$, and

$$L_1 = \frac{\sum_{n=1}^{N_h^k(s_h^k, a_h^k, b_h^k)} 2H\left(\overline{V}_{h+1}^{\mathrm{R},k^n}(s_{h+1}^{k^n}) - V_{h+1}^\star(s_{h+1}^{k^n})\right)}{N_h^k(s_h^k, a_h^k, b_h^k)}, \quad (195)$$

$$L_2 = \frac{\sum_{n=1}^{N_h^k(s_h^k, a_h^k, b_h^k)} \left(V_{h+1}^\star(s_{h+1}^{k^n})\right)^2}{N_h^k(s_h^k, a_h^k, b_h^k)} - \left(\frac{\sum_{n=1}^{N_h^k(s_h^k, a_h^k, b_h^k)} V_{h+1}^\star(s_{h+1}^{k^n})}{N_h^k(s_h^k, a_h^k, b_h^k)}\right)^2. \quad (196)$$

Therefore, we have

$$\left(\overline{V}_{h+1}^{\mathrm{R},k^n}(s_{h+1}^{k^n})\right)^2 - \left(V_{h+1}^\star(s_{h+1}^{k^n})\right)^2 = \left(\overline{V}_{h+1}^{\mathrm{R},k^n}(s_{h+1}^{k^n}) + V_{h+1}^\star(s_{h+1}^{k^n})\right)\left(\overline{V}_{h+1}^{\mathrm{R},k^n}(s_{h+1}^{k^n}) - V_{h+1}^\star(s_{h+1}^{k^n})\right)$$
$$\le 2H\left(\overline{V}_{h+1}^{\mathrm{R},k^n}(s_{h+1}^{k^n}) - V_{h+1}^\star(s_{h+1}^{k^n})\right).$$

With (194) in mind, we shall proceed to bound each term in (194) separately.

- The first term $L_1$ can be straightforwardly bounded as follows

$$L_1 = \frac{2H}{N_h^k(s_h^k, a_h^k, b_h^k)} \left(\sum_{n=1}^{N_h^k(s_h^k, a_h^k, b_h^k)} V' \mathbb{1}\left(V' \le 3\right) + \Phi_h^k(s_h^k, a_h^k, b_h^k)\right)$$
$$\le 6H + \frac{2H}{N_h^k(s_h^k, a_h^k, b_h^k)} \Phi_h^k(s_h^k, a_h^k, b_h^k), \quad (197)$$

where $\Phi_h^k(s_h^k, a_h^k, b_h^k)$ is defined as

$$\Phi_h^k(s_h^k, a_h^k, b_h^k) := \sum_{n=1}^{N_h^k(s_h^k, a_h^k, b_h^k)} V' \mathbb{1}\left(V' > 3\right) \quad (198)$$

with $V' = \overline{V}_{h+1}^{\mathrm{R},k^n}(s_{h+1}^{k^n}) - V_{h+1}^\star(s_{h+1}^{k^n})$.

- When it comes to the second term $L_2$, we claim that

$$L_2 \lesssim \mathsf{Var}_{h,s_h^k,a_h^k,b_h^k}(V_{h+1}^\star) + H^2\sqrt{\frac{\log^2 \frac{SABT}{\delta}}{N_h^k(s_h^k,a_h^k,b_h^k)}}, \tag{199}$$

which will be justified in Appendix E.2.1.

Plugging (197) and (199) into (194) and (193) allows one to demonstrate that

$$\sum_{k=1}^K \sqrt{\frac{\overline{\psi}_h^{\mathrm{r},k}(s_h^k,a_h^k,b_h^k) - (\overline{\phi}_h^{\mathrm{r},k}(s_h^k,a_h^k,b_h^k))^2}{N_h^k(s_h^k,a_h^k,b_h^k)}}$$

$$\lesssim \sum_{k=1}^K \sqrt{\frac{1}{N_h^k(s_h^k,a_h^k,b_h^k)}} \sqrt{H + \frac{H\Phi_h^k(s_h^k,a_h^k,b_h^k)}{N_h^k(s_h^k,a_h^k,b_h^k)} + \mathsf{Var}_{h,s_h^k,a_h^k,b_h^k}(V_{h+1}^\star) + H^2\sqrt{\frac{\log \frac{SAT}{\delta}}{N_h^k(s_h^k,a_h^k,b_h^k)}}}$$

$$\leq \sum_{k=1}^K \left( \sqrt{\frac{H}{N_h^k(s_h^k,a_h^k,b_h^k)}} + \frac{\sqrt{H\Phi_h^k(s_h^k,a_h^k,b_h^k)}}{N_h^k(s_h^k,a_h^k,b_h^k)} + \sqrt{\frac{\mathsf{Var}_{h,s_h^k,a_h^k,b_h^k}(V_{h+1}^\star)}{N_h^k(s_h^k,a_h^k,b_h^k)}} + \frac{H\log^{1/2}\frac{SABT}{\delta}}{(N_h^k(s_h^k,a_h^k,b_h^k))^{3/4}} \right). \tag{200}$$

Combined with (192) and (179), there is

$$(200) \lesssim \sqrt{HSABK} + \sum_{k=1}^K \frac{\sqrt{H\Phi_h^k(s_h^k,a_h^k,b_h^k)}}{N_h^k(s_h^k,a_h^k,b_h^k)}$$

$$+ \sum_{k=1}^K \sqrt{\frac{\mathsf{Var}_{h,s_h^k,a_h^k,b_h^k}(V_{h+1}^\star)}{N_h^k(s_h^k,a_h^k,b_h^k)}} + H(SAB)^{3/4}\left(K\log^2\frac{SABT}{\delta}\right)^{1/4}. \tag{201}$$

**Step 3: putting together the preceding results.** Finally, the above results in (191) and (201) taken collectively with (189) lead to

$$\mathcal{D}_2 \lesssim \sqrt{H^3SABK\log\frac{SABT}{\delta}} + \sum_{h=1}^H \sqrt{\log\frac{SABT}{\delta}} \sum_{k=1}^K \sqrt{\frac{\overline{\psi}_h^{\mathrm{r},k}(s_h^k,a_h^k,b_h^k) - (\overline{\phi}_h^{\mathrm{r},k}(s_h^k,a_h^k,b_h^k))^2}{N_h^k(s_h^k,a_h^k,b_h^k)}}$$

$$\lesssim \sqrt{H^3SABK\log\frac{SABT}{\delta}} + H^2(SAB)^{3/4}K^{1/4}\log\frac{SABT}{\delta}$$

$$+ \sqrt{\log\frac{SABT}{\delta}} \sum_{h=1}^H \sum_{k=1}^K \sqrt{\frac{\mathsf{Var}_{h,s_h^k,a_h^k,b_h^k}(V_{h+1}^\star)}{N_h^k(s_h^k,a_h^k,b_h^k)}} + \sqrt{H\log\frac{SABT}{\delta}} \sum_{h=1}^H \sum_{k=1}^K \frac{\sqrt{\Phi_h^k(s_h^k,a_h^k,b_h^k)}}{N_h^k(s_h^k,a_h^k,b_h^k)}.$$

Furthermore, we claimed two inequalities as follows:

$$\sum_{h=1}^H \sum_{k=1}^K \sqrt{\frac{\mathsf{Var}_{h,s_h^k,a_h^k,b_h^k}(V_{h+1}^\star)}{N_h^k(s_h^k,a_h^k,b_h^k)}} \lesssim \sqrt{H^2SABT\log\frac{SABT}{\delta}} + H^{3.5}SAB\log\frac{SABT}{\delta}, \tag{202}$$

$$\sum_{h=1}^H \sum_{k=1}^K \frac{\sqrt{\Phi_h^k(s_h^k,a_h^k,b_h^k)}}{N_h^k(s_h^k,a_h^k,b_h^k)} \lesssim H^{7/2}SAB\log^{3/2}\frac{SABT}{\delta}, \tag{203}$$

whose proofs are postponed to Appendix E.2.2 and Appendix E.2.3, respectively. Thus, there is

$$\mathcal{D}_2 \lesssim \sqrt{H^3SABK\log\frac{SABT}{\delta}} + H^2(SAB)^{3/4}K^{1/4}\log\frac{SABT}{\delta} + H^4SAB\log^2\frac{SABT}{\delta}$$

$$\lesssim \sqrt{H^3SABK\log\frac{SABT}{\delta}} + H^4SAB\log^2\frac{SABT}{\delta}$$

$$= \sqrt{H^2SABT\log\frac{SABT}{\delta}} + H^4SAB\log^2\frac{SABT}{\delta}.$$

where the second line above is valid since

$$H^2(SAB)^{3/4}K^{1/4}\log^{5/4}\frac{SABT}{\delta} = \left(H^{5/4}(SAB)^{1/2}\log\frac{SABT}{\delta}\right)\cdot\left(H^3SABK\log\frac{SABT}{\delta}\right)^{1/4}$$

$$\lesssim H^{2.5}SAB\log^2\frac{SABT}{\delta} + \sqrt{H^3SABK\log\frac{SABT}{\delta}}$$

$$= H^{2.5}SAB\log^2\frac{SABT}{\delta} + \sqrt{H^2SABT\log\frac{SABT}{\delta}}$$

due to the Cauchy-Schwarz inequality. This concludes the proof of the advertised upper bound on $\mathcal{D}_2$.

### E.2.1 PROOF OF THE INEQUALITY (199)

Akin to the proof of $W_{4,1}$ in (125), let

$$F_{h+1}^k := (V_{h+1}^\star)^2 \qquad \text{and} \qquad G_h^k(s,a,b,N) := \frac{1}{N}.$$

By observing and setting

$$\|F_{h+1}^k\|_\infty \leq H^2 =: C_{\mathsf{f}}, \qquad C_{\mathsf{g}} := \frac{1}{N},$$

we can apply Lemma 7 to yield

$$\left|\frac{1}{N_h^k}\sum_{n=1}^{N_h^k}\left(V_{h+1}^\star(s_{h+1}^{k^n})\right)^2 - P_{h,s_h^k,a_h^k,b_h^k}(V_{h+1}^\star)^2\right|$$

$$= \left|\frac{1}{N_h^k}\sum_{n=1}^{N_h^k}\left(P_h^{k^n} - P_{h,s_h^k,a_h^k,b_h^k}\right)(V_{h+1}^\star)^2\right| \lesssim H^2\sqrt{\frac{\log^2\frac{SABT}{\delta}}{N_h^k}}$$

with probability at least $1 - \delta$. Similarly, by applying the trivial bound $\|V_{h+1}^\star\|_\infty \leq H$ and Lemma 7, we can obtain

$$\left|\frac{1}{N_h^k}\sum_{n=1}^{N_h^k}V_{h+1}^\star(s_{h+1}^{k^n}) - P_{h,s_h^k,a_h^k,b_h^k}V_{h+1}^\star\right| = \left|\frac{1}{N_h^k}\sum_{n=1}^{N_h^k}\left(P_h^{k^n} - P_{h,s_h^k,a_h^k,b_h^k}\right)V_{h+1}^\star\right| \lesssim H\sqrt{\frac{\log\frac{SABT}{\delta}}{N_h^k}}$$

with probability at least $1 - \delta$.

Recalling from (16) the definition

$$\mathsf{Var}_{h,s_h^k,a_h^k,b_h^k}(V_{h+1}^\star) = P_{h,s_h^k,a_h^k,b_h^k}(V_{h+1}^\star)^2 - \left(P_{h,s_h^k,a_h^k,b_h^k}V_{h+1}^\star\right)^2,$$

we can use the preceding two bounds and the triangle inequality to show that: with probability at least $1 - \delta$, there is

$$\left|\frac{1}{N_h^k}\sum_{n=1}^{N_h^k}V_{h+1}^\star(s_{h+1}^{k^n})^2 - \left(\frac{1}{N_h^k}\sum_{n=1}^{N_h^k}V_{h+1}^\star(s_{h+1}^{k^n})\right)^2 - \mathsf{Var}_{h,s_h^k,a_h^k,b_h^k}(V_{h+1}^\star)\right|$$

$$\leq \left|\frac{1}{N_h^k}\sum_{n=1}^{N_h^k}V_{h+1}^\star(s_{h+1}^{k^n})^2 - P_{h,s_h^k,a_h^k,b_h^k}(V_{h+1}^\star)^2\right| - \left|\left(\frac{1}{N_h^k}\sum_{n=1}^{N_h^k}V_{h+1}^\star(s_{h+1}^{k^n})\right)^2 - (P_{h,s_h^k,a_h^k,b_h^k}V_{h+1}^\star)^2\right|$$

$$\lesssim \left|\frac{1}{N_h^k}\sum_{n=1}^{N_h^k}V_{h+1}^\star(s_{h+1}^{k^n}) - P_{h,s_h^k,a_h^k,b_h^k}V_{h+1}^\star\right|\left|\frac{1}{N_h^k}\sum_{n=1}^{N_h^k}V_{h+1}^\star(s_{h+1}^{k^n}) + P_{h,s_h^k,a_h^k,b_h^k}V_{h+1}^\star\right| + H^2\sqrt{\frac{\log^2\frac{SABT}{\delta}}{N_h^k}}.$$

And thus, with the fact that $\|V_{h+1}^\star\|_\infty \leq H$, we get

$$\left|\frac{1}{N_h^k}\sum_{n=1}^{N_h^k}V_{h+1}^\star(s_{h+1}^{k^n})^2 - \left(\frac{1}{N_h^k}\sum_{n=1}^{N_h^k}V_{h+1}^\star(s_{h+1}^{k^n})\right)^2 - \mathsf{Var}_{h,s_h^k,a_h^k,b_h^k}(V_{h+1}^\star)\right| \lesssim H^2\sqrt{\frac{\log^2\frac{SABT}{\delta}}{N_h^k}}.$$

### E.2.2 PROOF OF THE INEQUALITY (202)

To begin with, we make the observation that

$$\sum_{k=1}^{K}\sqrt{\frac{\mathsf{Var}_{h,s_h^k,a_h^k,b_h^k}(V_{h+1}^{\star})}{N_h^k(s_h^k,a_h^k,b_h^k)}} = \sum_{(s,a,b)\in\mathcal{S}\times\mathcal{A}\times\mathcal{B}}\sum_{n=1}^{N_h^K(s,a,b)}\sqrt{\frac{\mathsf{Var}_{h,s,a,b}(V_{h+1}^{\star})}{n}}$$

$$\leq 2\sum_{(s,a,b)\in\mathcal{S}\times\mathcal{A}\times\mathcal{B}}\sqrt{N_h^K(s,a,b)\mathsf{Var}_{h,s,a,b}(V_{h+1}^{\star})},$$

which relies on the fact that $\sum_{n=1}^{N}1/\sqrt{n}\leq 2\sqrt{N}$. It then follows that

$$\sum_{h=1}^{H}\sum_{k=1}^{K}\sqrt{\frac{\mathsf{Var}_{h,s_h^k,a_h^k,b_h^k}(V_{h+1}^{\star})}{N_h^k(s_h^k,a_h^k,b_h^k)}}$$

$$\leq 2\sum_{h=1}^{H}\sum_{(s,a,b)\in\mathcal{S}\times\mathcal{A}\times\mathcal{B}}\sqrt{N_h^K(s,a,b)\mathsf{Var}_{h,s,a,b}(V_{h+1}^{\star})}$$

$$\leq 2\sqrt{\sum_{h=1}^{H}\sum_{(s,a,b)\in\mathcal{S}\times\mathcal{A}\times\mathcal{B}}1}\cdot\sqrt{\sum_{h=1}^{H}\sum_{(s,a,b)\in\mathcal{S}\times\mathcal{A}\times\mathcal{B}}N_h^K(s,a,b)\mathsf{Var}_{h,s,a,b}(V_{h+1}^{\star})}$$

$$= 2\sqrt{HSAB}\sqrt{\sum_{h=1}^{H}\sum_{k=1}^{K}\mathsf{Var}_{h,s_h^k,a_h^k,b_h^k}(V_{h+1}^{\star})}, \tag{204}$$

where the second inequality invokes the Cauchy-Schwarz inequality.

The rest of the proof is then dedicated to bounding (204). Towards this end, we first decompose

$$\sum_{h=1}^{H}\sum_{k=1}^{K}\mathsf{Var}_{h,s_h^k,a_h^k,b_h^k}(V_{h+1}^{\star})$$

$$\leq \sum_{h=1}^{H}\sum_{k=1}^{K}\mathsf{Var}_{h,s_h^k,a_h^k,b_h^k}(V_{h+1}^{\pi^k}) + \sum_{h=1}^{H}\sum_{k=1}^{K}\left|\mathsf{Var}_{h,s_h^k,a_h^k,b_h^k}(V_{h+1}^{\star}) - \mathsf{Var}_{h,s_h^k,a_h^k,b_h^k}(V_{h+1}^{\pi^k})\right|$$

$$\lesssim HT + H^3\log\frac{SABT}{\delta} + \sum_{h=1}^{H}\sum_{k=1}^{K}\left|\mathsf{Var}_{h,s_h^k,a_h^k,b_h^k}(V_{h+1}^{\star}) - \mathsf{Var}_{h,s_h^k,a_h^k,b_h^k}(V_{h+1}^{\pi^k})\right|, \tag{205}$$

where the last inequality follows from (Jin et al., 2018b, Lemma C.5) for a formal proof. The second term on the right-hand side of (205) can be bounded as follows

$$\sum_{h=1}^{H}\sum_{k=1}^{K}\left|\mathsf{Var}_{h,s_h^k,a_h^k,b_h^k}(V_{h+1}^{\star}) - \mathsf{Var}_{h,s_h^k,a_h^k,b_h^k}(V_{h+1}^{\pi^k})\right|$$

$$= \sum_{h=1}^{H}\sum_{k=1}^{K}\left|P_{h,s_h^k,a_h^k,b_h^k}(V_{h+1}^{\star})^2 - \left(P_{h,s_h^k,a_h^k,b_h^k}V_{h+1}^{\star}\right)^2 - P_{h,s_h^k,a_h^k,b_h^k}(V_{h+1}^{\pi^k})^2 + \left(P_{h,s_h^k,a_h^k,b_h^k}V_{h+1}^{\pi^k}\right)^2\right|$$

$$\leq \sum_{h=1}^{H}\sum_{k=1}^{K}\left\{\left|P_{h,s_h^k,a_h^k,b_h^k}\left(\left(V_{h+1}^{\star}-V_{h+1}^{\pi^k}\right)\left(V_{h+1}^{\star}+V_{h+1}^{\pi^k}\right)\right)\right|\right.$$

$$\left. + \left|\left(P_{h,s_h^k,a_h^k,b_h^k}V_{h+1}^{\star}\right)^2 - \left(P_{h,s_h^k,a_h^k,b_h^k}V_{h+1}^{\pi^k}\right)^2\right|\right\}. \tag{206}$$

Before next proof, we first state that

$$
\left| P_{h,s_h^k,a_h^k,b_h^k}\left(\left(V_{h+1}^\star - V_{h+1}^{\pi^k}\right)\left(V_{h+1}^\star + V_{h+1}^{\pi^k}\right)\right) \right|
$$
$$
\leq \left| P_{h,s_h^k,a_h^k,b_h^k}\left(V_{h+1}^\star - V_{h+1}^{\pi^k}\right) \right| \left( \left\| V_{h+1}^\star \right\|_\infty + \left\| V_{h+1}^{\pi^k} \right\|_\infty \right) \leq 2H \left| P_{h,s_h^k,a_h^k,b_h^k}\left(V_{h+1}^\star - V_{h+1}^{\pi^k}\right) \right|,
$$
$$
\left| \left(P_{h,s_h^k,a_h^k,b_h^k} V_{h+1}^\star\right)^2 - \left(P_{h,s_h^k,a_h^k,b_h^k} V_{h+1}^{\pi^k}\right)^2 \right|
$$
$$
\leq \left| P_{h,s_h^k,a_h^k,b_h^k}\left(V_{h+1}^\star - V_{h+1}^{\pi^k}\right) \right| \cdot \left| P_{h,s_h^k,a_h^k,b_h^k}\left(V_{h+1}^\star + V_{h+1}^{\pi^k}\right) \right| \leq 2H \left| P_{h,s_h^k,a_h^k,b_h^k}\left(V_{h+1}^\star - V_{h+1}^{\pi^k}\right) \right|.
$$

And thus, there is

$$
\begin{aligned}
(206) &\leq 4H \sum_{h=1}^H \sum_{k=1}^K \left| P_{h,s_h^k,a_h^k,b_h^k}\left(V_{h+1}^\star - V_{h+1}^{\pi^k}\right) \right| \\
&\stackrel{(i)}{\leq} 4H \sum_{h=1}^H \sum_{k=1}^K P_{h,s_h^k,a_h^k,b_h^k}\left(\overline{V}_{h+1}^k - \underline{V}_{h+1}^k\right) \\
&= 4H \sum_{h=1}^H \sum_{k=1}^K \left\{ \overline{V}_{h+1}^k(s_{h+1}^k) - \underline{V}_{h+1}^k(s_{h+1}^k) + \left(P_{h,s_h^k,a_h^k,b_h^k} - P_h^k\right)\left(\overline{V}_{h+1}^k - \underline{V}_{h+1}^k\right) \right\} \\
&\leq 4H \sum_{h=1}^H \sum_{k=1}^K \left(\delta_{h+1}^k + \chi_{h+1}^k\right) \\
&\stackrel{(ii)}{\lesssim} H^2 \sqrt{T \log \frac{SABT}{\delta}} + \sqrt{H^7 SABT \log \frac{SABT}{\delta}} + H^4 SAB \\
&\asymp \sqrt{H^7 SABT \log \frac{SAT}{\delta}} + H^4 SAB, \quad\quad\quad\quad\quad\quad (207)
\end{aligned}
$$

where we define

$$
\delta_{h+1}^k := \overline{V}_{h+1}^k(s_{h+1}^k) - \underline{V}_{h+1}^k(s_{h+1}^k), \quad\quad \chi_{h+1}^k := \left(P_{h,s_h^k,a_h^k,b_h^k} - P_h^k\right)\left(\overline{V}_{h+1}^k(s_{h+1}^k) - \underline{V}_{h+1}^k\right). \tag{208}
$$

We shall take a moment to explain how we derive (207). The inequality (ii) is valid since $\underline{V}_{h+1} \leq V^\star \leq \overline{V}_{h+1}$ and $\underline{V}_{h+1} \leq V^\pi \leq \overline{V}_{h+1}$; and (iii) results from the following two bounds:

$$
\begin{aligned}
\sum_{h=1}^H \sum_{k=1}^K \delta_{h+1}^k &\leq \sum_{h=1}^H \left( eH^2 SAB + \sqrt{H^3 T \log \frac{SABT}{\delta}} + 10 c_{\mathrm{b}} \sqrt{eH^3 SABT \log \frac{SABT}{\delta}} \right) \\
&\lesssim \sqrt{H^5 SABT \log \frac{SABT}{\delta}} + H^3 SAB, \quad\quad\quad\quad\quad\quad (209a)
\end{aligned}
$$
$$
\sum_{h=1}^H \sum_{k=1}^K \chi_{h+1}^k \lesssim H \sqrt{T \log \frac{SABT}{\delta}}, \quad\quad\quad\quad\quad\quad\quad\quad\quad\quad (209b)
$$

where the first inequality in (209a) comes from (144) with $w_k = 1$ and $C \leq K$, and (209b) comes from the Azuma-Hoeffding inequality and a union bound.

As a consequence, substituting (205) and (207) into (204), we reach

$$
\sum_{h=1}^{H} \sum_{k=1}^{K} \sqrt{\frac{\mathsf{Var}_{h,s_h^k,a_h^k,b_h^k}(V_{h+1}^\star)}{N_h^k(s_h^k,a_h^k,b_h^k)}}
$$

$$
\lesssim \sqrt{HSAB} \sqrt{HT + \sqrt{H^7 SABT \log \frac{SABT}{\delta}} + H^4 SAB}
$$

$$
\lesssim \sqrt{H^2 SABT} + H^{9/4}(SAB)^{3/4} \Big( T \log \frac{SABT}{\delta} \Big)^{1/4} + H^{2.5} SAB
$$

$$
= \sqrt{H^2 SABT} + \Big( H^2 SABT \log \frac{SABT}{\delta} \Big)^{1/4} \big( H^{3.5} SAB \big)^{1/2} + H^{2.5} SAB
$$

$$
\lesssim \sqrt{H^2 SABT \log \frac{SABT}{\delta}} + H^{3.5} SAB \log \frac{SABT}{\delta},
$$

where we have applied the basic inequality $2ab \le a^2 + b^2$ for any $a, b \ge 0$.

### E.2.3 PROOF OF THE INEQUALITY (203)

First, it is observed that

$$
\sum_{k=1}^{K} \frac{\sqrt{\Phi_h^k(s_h^k, a_h^k, b_h^k)}}{N_h^k(s_h^k, a_h^k, b_h^k)} = \sum_{(s,a,b) \in \mathcal{S} \times \mathcal{A} \times \mathcal{B}} \sum_{n=1}^{N_h^K(s,a,b)} \frac{\sqrt{\Phi_h^{k^n(s,a,b)}(s,a,b)}}{n}
$$

$$
\le \sum_{(s,a,b) \in \mathcal{S} \times \mathcal{A} \times \mathcal{B}} \sqrt{\Phi_h^{N_h^K(s,a,b)}(s,a,b)} \log T \le \sqrt{SAB \sum_{(s,a,b) \in \mathcal{S} \times \mathcal{A} \times \mathcal{B}} \Phi_h^{N_h^K(s,a,b)}(s,a,b)} \log T.
$$
(210)

Here, the first inequality holds by the monotonicity property of $\Phi_h^k(s_h, a_h)$ with respect to $k$ (see its definition in (198)) due to the same property of $V_{h+1}^{\mathrm{R},k}$, while the second inequality comes from Cauchy-Schwarz.

To continue, with $V' = \overline{V}_{h+1}^{\mathrm{R},k^n}(s_{h+1}^{k^n}) - V_{h+1}^\star(s_{h+1}^{k^n})$, note that

$$
\sum_{h=1}^{H} \sqrt{\sum_{(s,a,b) \in \mathcal{S} \times \mathcal{A} \times \mathcal{B}} \Phi_h^{N_h^K(s,a,b)}(s,a,b)} = \sum_{h=1}^{H} \sqrt{\sum_{k=1}^{K} (V') \mathbb{1}(V' > 3)}
$$

$$
\le \sum_{h=1}^{H} \sqrt{\sum_{k=1}^{K} \Big( \overline{V}_{h+1}^k(s_{h+1}^k) + 2 - \underline{V}_{h+1}^k(s_{h+1}^k) \Big) \mathbb{1}\Big( \overline{V}_{h+1}^k(s_{h+1}^k) + 2 - \underline{V}_{h+1}^k(s_{h+1}^k) > 3 \Big)}
$$

$$
= \sum_{h=1}^{H} \sqrt{\sum_{k=1}^{K} \Big( \overline{V}_{h+1}^k(s_{h+1}^k) + 2 - \underline{V}_{h+1}^k(s_{h+1}^k) \Big) \mathbb{1}\Big( \overline{V}_{h+1}^k(s_{h+1}^k) - \underline{V}_{h+1}^k(s_{h+1}^k) > 1 \Big)}
$$

$$
\le \sum_{h=1}^{H} \sqrt{\sum_{k=1}^{K} 3\Big( \overline{V}_{h+1}^k(s_{h+1}^k) - \underline{V}_{h+1}^k(s_{h+1}^k) \Big) \mathbb{1}\Big( \overline{V}_{h+1}^k(s_{h+1}^k) - \underline{V}_{h+1}^k(s_{h+1}^k) > 1 \Big)}
$$

$$
\le \sqrt{H} \sqrt{\sum_{h=1}^{H} \sum_{k=1}^{K} 3\Big( \overline{V}_{h+1}^k(s_{h+1}^k) - \underline{V}_{h+1}^k(s_{h+1}^k) \Big) \mathbb{1}\Big( \overline{V}_{h+1}^k(s_{h+1}^k) - \underline{V}_{h+1}^k(s_{h+1}^k) > 1 \Big)}, \quad (211)
$$

where the first inequality follows from Lemma 4 (cf. (27)) and Lemma 2 (so that $\overline{V}_{h+1}^{\mathrm{R},k}(s_{h+1}^k) - V_{h+1}^\star(s_{h+1}^k) \le \overline{V}_{h+1}^k(s_{h+1}^k) + 2 - \underline{V}_{h+1}^k(s_{h+1}^k)$), the penultimate inequality holds since $1 \le \overline{V}_{h+1}^k(s_{h+1}^k) - \underline{V}_{h+1}^k(s_{h+1}^k)$ when $\mathbb{1}\Big( \overline{V}_{h+1}^k(s_{h+1}^k) - \underline{V}_{h+1}^k(s_{h+1}^k) > 1 \Big) \ne 0$, and the last inequality is a consequence of the Cauchy-Schwarz inequality.

Combining the above relation with (171) and applying the triangle inequality, we can demonstrate that

$$\sum_{h=1}^{H} \sqrt{\sum_{(s,a,b)\in\mathcal{S}\times\mathcal{A}\times\mathcal{B}} \Phi_h^{N_h^K(s,a,b)}(s,a,b)} \lesssim \sqrt{H^7 SAB \log \frac{SABT}{\delta}},$$

where the inequality follows directly from (28. Substitution into (210) gives

$$\sum_{h=1}^{H}\sum_{k=1}^{K} \frac{\sqrt{\Phi_h^k(s_h^k,a_h^k,b_h^k)}}{N_h^k(s_h^k,a_h^k,b_h^k)} \lesssim \left(\sqrt{SAB}\log T\right)\cdot\sqrt{H^7 SAB\log\frac{SABT}{\delta}} \asymp H^{7/2}SAB\log^{3/2}\frac{SABT}{\delta},$$

thus concluding the proof.

### E.3 Bounding the term $\mathcal{D}_3$

First of all, there is

$$\lambda_h^k := \left(1+\frac{1}{H}\right)^{h-1} \sum_{n=N_h^k(s_h^k,a_h^k,b_h^k)}^{N_h^{K-1}(s_h^k,a_h^k,b_h^k)} \eta_{N_h^k(s_h^k,a_h^k,b_h^k)}^n \le \left(1+\frac{1}{H}\right)^h \le \left(1+\frac{1}{H}\right)^H \le e, \quad (212)$$

where the first inequality in (212) follows from the property $\sum_{N=n}^{\infty}\eta_n^N \le 1+1/H$ in Lemma 1 and the last inequality in (212) results from (186). Then we could decompose the expression of $\mathcal{D}_3$ in (35c) as follows

$$\mathcal{D}_3 := \mathcal{D}_{3,1} + \mathcal{D}_{3,2} - \mathcal{D}_{3,3} - \mathcal{D}_{3,4}$$

with

$$\mathcal{D}_{3,1} = \sum_{h=1}^{H}\sum_{k=1}^{K} \lambda_h^k \left(P_h^k - P_{h,s_h^k,a_h^k,b_h^k}\right)\left(V_{h+1}^\star - \overline{V}_{h+1}^{R,k}\right) \qquad (213)$$

$$\mathcal{D}_{3,2} = \sum_{h=1}^{H}\sum_{k=1}^{K} \lambda_h^k \frac{\sum_{i\le N_h^k(s_h^k,a_h^k,b_h^k)}\left(\overline{V}_{h+1}^{R,k^i}(s_{h+1}^{k^i}) - P_{h,s_h^k,a_h^k,b_h^k}\overline{V}_{h+1}^{R,k}\right)}{N_h^k(s_h^k,a_h^k,b_h^k)} \qquad (214)$$

$$\mathcal{D}_{3,3} = \sum_{h=1}^{H}\sum_{k=1}^{K} \lambda_h^k \left(P_h^k - P_{h,s_h^k,a_h^k,b_h^k}\right)\left(V_{h+1}^\star - \underline{V}_{h+1}^{R,k}\right) \qquad (215)$$

$$\mathcal{D}_{3,4} = \sum_{h=1}^{H}\sum_{k=1}^{K} \lambda_h^k \frac{\sum_{i\le N_h^k(s_h^k,a_h^k,b_h^k)}\left(\underline{V}_{h+1}^{R,k^i}(s_{h+1}^{k^i}) - P_{h,s_h^k,a_h^k,b_h^k}\underline{V}_{h+1}^{R,k}\right)}{N_h^k(s_h^k,a_h^k,b_h^k)}. \qquad (216)$$

In the sequel, we shall control each of these two terms separately.

**Step 1: upper bounding $\mathcal{D}_{3,1}$ and $\mathcal{D}_{3,3}$.** We plan to control this term by means of Lemma 8. For notational simplicity, let us define

$$N(s,a,b,h) := N_h^{K-1}(s,a,b)$$

and set

$$F_{h+1}^k := \overline{V}_{h+1}^{R,k} - V_{h+1}^\star \quad \text{and} \quad G_h^k(s_h^k,a_h^k) := \lambda_h^k = \left(1+\frac{1}{H}\right)^{h-1} \sum_{n=N_h^k(s_h^k,a_h^k,b_h^k)}^{N(s_h^k,a_h^k,b_h^k,h)} \eta_{N_h^k(s_h^k,a_h^k,b_h^k)}^n.$$

Given the fact that $\overline{V}_{h+1}^{R,k}(s), V_{h+1}^\star(s) \in [0,H]$ and the condition (212), it is readily seen that

$$\left\|F_{h+1}^k\right\|_\infty \le H =: C_f \qquad \text{and} \qquad \left|G_h^k(s_h^k,a_h^k,b_h^k)\right| \le e =: C_g.$$

Apply Lemma 8 to yield, with probability at least $1 - \delta/2$, there is

$$
|\mathcal{D}_{3,1}| = \left| \sum_{h=1}^{H} \sum_{k=1}^{K} \lambda_h^k \big(P_h^k - P_{h,s_h^k,a_h^k,b_h^k}\big)\big(V_{h+1}^\star - \overline{V}_{h+1}^{\mathrm{R},k}\big) \right|
$$

$$
\lesssim \sqrt{C_{\mathrm{g}}^2 C_{\mathrm{f}} H S A B \sum_{h=1}^{H} \sum_{k=1}^{K} \mathbb{E}_{k,h-1}\big[P_h^k F_{h+1}^k\big] \log\frac{K}{\delta} + C_{\mathrm{g}} C_{\mathrm{f}} H S A B \log\frac{K}{\delta}}
$$

$$
\lesssim \sqrt{H^2 S A B \sum_{h=1}^{H} \sum_{k=1}^{K} \mathbb{E}_{k,h-1}\big[P_h^k\big(\overline{V}_{h+1}^{\mathrm{R},k} - V_{h+1}^\star\big)\big] \log\frac{T}{\delta} + H^2 S A B \log\frac{T}{\delta}}
$$

$$
\asymp \sqrt{H^2 S A B \left\{ \sum_{h=1}^{H} \sum_{k=1}^{K} P_{h,s_h^k,a_h^k,b_h^k}\big(\overline{V}_{h+1}^{\mathrm{R},k} - V_{h+1}^\star\big) \right\} \log\frac{T}{\delta} + H^2 S A B \log\frac{T}{\delta}} \tag{217}
$$

with $X_{k,h} = \lambda_h^k \big(P_h^k - P_{h,s_h^k,a_h^k,b_h^k}\big)\big(V_{h+1}^\star - \overline{V}_{h+1}^{\mathrm{R},k}\big)$.

It then comes down to controlling the sum $\sum_{h=1}^{H} \sum_{k=1}^{K} P_{h,s_h^k,a_h^k,b_h^k}\big(\overline{V}_{h+1}^{\mathrm{R},k} - V_{h+1}^\star\big)$. Towards this end, we first single out the following useful fact: with probability at least $1 - \delta/4$,

$$
\sum_{h=1}^{H} \sum_{k=1}^{K} P_h^k\big(\overline{V}_{h+1}^{\mathrm{R},k} - V_{h+1}^\star\big) \overset{(\mathrm{i})}{\leq} \sum_{h=1}^{H} \sum_{k=1}^{K} P_h^k\big(\overline{V}_{h+1}^k + 2 - V_{h+1}^\star\big)
$$

$$
\leq 2HK + \sum_{h=1}^{H} \sum_{k=1}^{K} \Big(\overline{V}_{h+1}^k(s_{h+1}^k) - V_{h+1}^\star(s_{h+1}^k)\Big) \overset{(\mathrm{ii})}{\lesssim} H^3\sqrt{SABK \log\frac{SABT}{\delta}} + H^3 SAB + HK, \tag{218}
$$

where (i) holds according to (27), and (ii) is valid since

$$
\sum_{h=1}^{H} \sum_{k=1}^{K} \Big(\overline{V}_{h+1}^k(s_{h+1}^k) - V_{h+1}^\star(s_{h+1}^k)\Big) \leq \sum_{h=1}^{H} \sum_{k=1}^{K} \Big(\overline{V}_{h+1}^k(s_{h+1}^k) - \underline{V}_{h+1}^k(s_{h+1}^k)\Big)
$$

$$
\lesssim H^3\sqrt{SABK \log\frac{SABT}{\delta}} + H^3 SAB,
$$

where the first inequality follows since $V_{h+1}^\star \geq \underline{V}_{h+1}^k$, and the second inequality comes from (209a). Additionally, invoking Freedman's inequality (see Lemma 8) with $c_h = 1$ and $\widetilde{W}_h^i = \overline{V}_{h+1}^{\mathrm{R},k} - V_{h+1}^\star$ (so that $0 \leq \widetilde{W}_h^i(s) \leq H$) directly leads to

$$
\left| \sum_{h=1}^{H} \sum_{k=1}^{K} \big(P_h^k - P_{h,s_h^k,a_h^k,b_h^k}\big)\big(\overline{V}_{h+1}^{\mathrm{R},k} - V_{h+1}^\star\big) \right| \lesssim \sqrt{T H^2 \log\frac{1}{\delta}} + H \log\frac{1}{\delta} \asymp \sqrt{H^3 K \log\frac{1}{\delta}}
$$

with probability at least $1 - \delta/4$, which taken collectively with (218) reveals that

$$
\sum_{h=1}^{H} \sum_{k=1}^{K} P_{s_h^k,a_h^k,b_h^k,h}\big(\overline{V}_{h+1}^{\mathrm{R},k} - V_{h+1}^\star\big)
$$

$$
\leq \sum_{h=1}^{H} \sum_{k=1}^{K} P_h^k\big(\overline{V}_{h+1}^{\mathrm{R},k} - V_{h+1}^\star\big) + \left| \sum_{h=1}^{H} \sum_{k=1}^{K} \big(P_h^k - P_{s_h^k,a_h^k,h}\big)\big(\overline{V}_{h+1}^{\mathrm{R},k} - V_{h+1}^\star\big) \right|
$$

$$
\lesssim H^3\sqrt{SABK \log\frac{SABT}{\delta}} + H^3 SAB + HK \tag{219}
$$

with probability at least $1 - \delta/2$. Substitution into (217) then gives

$$
\begin{aligned}
|\mathcal{D}_{3,1}| &= \left| \sum_{h=1}^{H} \sum_{k=1}^{K} \lambda_h^k \left( P_h^k - P_{h,s_h^k,a_h^k,b_h^k} \right) \left( V_{h+1}^{\star} - \overline{V}_{h+1}^{\mathrm{R},k} \right) \right| \\
&\lesssim \sqrt{ H^2 SAB \sum_{h=1}^{H} \sum_{k=1}^{K} P_{h,s_h^k,a_h^k,b_h^k} \left( \overline{V}_{h+1}^{\mathrm{R},k} - V_{h+1}^{\star} \right) \log \frac{T}{\delta} + H^2 SAB \log \frac{T}{\delta} } \\
&\lesssim \sqrt{ H^2 SAB \left( H^3 \sqrt{ SABK \log \frac{SABT}{\delta} } + H^3 SAB + HK \right) \log \frac{T}{\delta} + H^2 SAB \log \frac{T}{\delta} }.
\end{aligned}
$$

According to the Cauchy-Schwarz inequality, there is

$$
\sqrt{ H^6 SABK \log \frac{SABT}{\delta} } = \sqrt{ H^5 SAB \log \frac{SABT}{\delta} } \sqrt{HK} \lesssim H^5 SAB \log \frac{SABT}{\delta} + HK,
$$

and then we can further obtain

$$
\begin{aligned}
|\mathcal{D}_{3,1}| &\asymp \sqrt{ H^2 SAB \left( H^5 SAB \log \frac{SABT}{\delta} + H^3 SAB + HK \right) \log \frac{T}{\delta} + H^2 SAB \log \frac{T}{\delta} } \\
&\lesssim \sqrt{ H^3 SABK \log \frac{SABT}{\delta} + H^{3.5} SAB \log \frac{SABT}{\delta} } \\
&= \sqrt{ H^2 SABT \log \frac{SABT}{\delta} + H^{3.5} SAB \log \frac{SABT}{\delta} }. \qquad (220)
\end{aligned}
$$

Similarly, $|\mathcal{D}_{3,3}| \lesssim \sqrt{ H^2 SABT \log \frac{SABT}{\delta} + H^{3.5} SAB \log \frac{SABT}{\delta} }$.

**Step 2: upper bounding $\mathcal{D}_{3,2}$ and $\mathcal{D}_{3,4}$.** According to the monotonicity property $\overline{V}_{h+1}^{\mathrm{R},k} \geq \overline{V}_{h+1}^{\mathrm{R},k+1} \geq \cdots \geq \overline{V}_{h+1}^{\mathrm{R},K}$, we make the following observation:

$$
\begin{aligned}
\mathcal{D}_{3,2} &\leq \sum_{h=1}^{H} \sum_{k=1}^{K} \frac{\lambda_h^k}{N_h^k(s_h^k,a_h^k,b_h^k)} \sum_{i \leq N_h^k(s_h^k,a_h^k,b_h^k)} \left( \overline{V}_{h+1}^{\mathrm{R},k^i}(s_{h+1}^{k^i}) - P_{h,s_h^k,a_h^k,b_h^k} \overline{V}_{h+1}^{\mathrm{R},K} \right) \\
&= \sum_{h=1}^{H} \sum_{k=1}^{K} \sum_{n=N_h^k(s_h^k,a_h^k,b_h^k)}^{N_h^{K-1}(s_h^k,a_h^k,b_h^k)} \frac{\lambda_h^k}{n} \left( \overline{V}_{h+1}^{\mathrm{R},k}(s_{h+1}^k) - \overline{V}_{h+1}^{\mathrm{R},K}(s_{h+1}^k) + \left( P_h^k - P_{h,s_h^k,a_h^k,b_h^k} \right) \overline{V}_{h+1}^{\mathrm{R},K} \right).
\end{aligned}
$$

And with the facts that $\sum_{n=N_h^k(s_h^k,a_h^k,b_h^k)}^{N_h^{K-1}(s_h^k,a_h^k,b_h^k)} \frac{1}{n} \leq \log T$ and $\lambda_h^k \leq e$ (cf. (212)), there is

$$
\begin{aligned}
\mathcal{D}_{3,2} &\leq (e \log T) \sum_{h=1}^{H} \sum_{k=1}^{K} \left( \overline{V}_{h+1}^{\mathrm{R},k}(s_{h+1}^k) - \overline{V}_{h+1}^{\mathrm{R},K}(s_{h+1}^k) \right) \\
&\quad + \sum_{h=1}^{H} \sum_{k=1}^{K} \sum_{n=N_h^k(s_h^k,a_h^k,b_h^k)}^{N_h^{K-1}(s_h^k,a_h^k,b_h^k)} \frac{\lambda_h^k}{n} \left( P_h^k - P_{h,s_h^k,a_h^k,b_h^k} \right) V_{h+1}^{\star} \\
&\quad + \sum_{h=1}^{H} \sum_{k=1}^{K} \sum_{n=N_h^k(s_h^k,a_h^k,b_h^k)}^{N_h^{K-1}(s_h^k,a_h^k,b_h^k)} \frac{\lambda_h^k}{n} \left( P_h^k - P_{h,s_h^k,a_h^k,b_h^k} \right) \left( \overline{V}_{h+1}^{\mathrm{R},K} - V_{h+1}^{\star} \right). \qquad (221)
\end{aligned}
$$

In what follows, we shall control the three terms in (221) separately.

- The first term in (221) can be controlled by Lemma 4 (cf. (28)) as follows:

$$
\sum_{h=1}^{H} \sum_{k=1}^{K} \left( \overline{V}_{h+1}^{\mathrm{R},k}(s_{h+1}^k) - \overline{V}_{h+1}^{\mathrm{R},K}(s_{h+1}^k) \right) \lesssim H^6 SAB \log \frac{SABT}{\delta} \qquad (222)
$$

with probability at least $1 - \delta/3$.

- To control the second term in (221), we abuse the notation by setting

$$N(s, a, b, h) := N_h^{K-1}(s, a, b)$$

and

$$F_{h+1}^k := V_{h+1}^\star, \qquad \text{and} \qquad G_h^k(s_h^k, a_h^k, b_h^k) := \sum_{n=N_h^k(s_h^k, a_h^k, b_h^k)}^{N(s_h^k, a_h^k, b_h^k, h)} \frac{\lambda_h^k}{n},$$

which clearly satisfy

$$\|F_{h+1}^k\|_\infty \le H =: C_{\mathsf{f}} \quad \text{and} \quad \left| G_h^k(s_h^k, a_h^k, b_h^k) \right| \le e \sum_{n=N_h^k(s_h^i, a_h^i, b_h^i)}^{N(s_h^k, a_h^k, b_h^k, h)} \frac{1}{n} \le e \log T =: C_{\mathsf{g}}.$$

Here, we have made use of the properties $\sum_{n=N_h^k(s_h^k, a_h^k, b_h^k)}^{N_h^{K-1}(s_h^k, a_h^k, b_h^k)} \frac{1}{n} \le \log T$ and $\lambda_h^k \le e$ (cf. (212)). With these in place, applying Lemma 8 reveals that, with probability at least $1 - \delta/3$,

$$\left| \sum_{h=1}^H \sum_{k=1}^K \sum_{n=N_h^k(s_h^k, a_h^k, b_h^k)}^{N_h^{K-1}(s_h^k, a_h^k, b_h^k)} \frac{\lambda_h^k}{n} \left( P_h^k - P_{h, s_h^k, a_h^k, b_h^k} \right) V_{h+1}^\star \right| = \left| \sum_{h=1}^H \sum_{k=1}^K X_{k,h} \right|$$

$$\lesssim \sqrt{ C_{\mathsf{g}}^2 HSAB \sum_{h=1}^H \sum_{i=1}^K \mathbb{E}_{i, h-1} \left[ \left| (P_h^i - P_{h, s_h^i, a_h^i}) W_{h+1}^k \right|^2 \right] \log \frac{T}{\delta} } + C_{\mathsf{g}} C_{\mathsf{f}} HSAB \log \frac{T}{\delta}$$

$$\asymp \sqrt{ \sum_{h=1}^H \sum_{k=1}^K \mathsf{Var}_{h, s_h^k, a_h^k, b_h^k}(V_{h+1}^\star) \cdot HSAB \log^3 \frac{T}{\delta} + H^2 SAB \log^2 \frac{T}{\delta} },$$

where $X_{k,h} = \sum_{n=N_h^k(s_h^k, a_h^k, b_h^k)}^{N_h^{K-1}(s_h^k, a_h^k, b_h^k)} \frac{\lambda_h^k}{n} \left( P_h^k - P_{h, s_h^k, a_h^k, b_h^k} \right) V_{h+1}^\star$ and the last line comes from the definition in (16). Therefore, we have

$$\left| \sum_{h=1}^H \sum_{k=1}^K \sum_{n=N_h^k(s_h^k, a_h^k, b_h^k)}^{N_h^{K-1}(s_h^k, a_h^k, b_h^k)} \frac{\lambda_h^k}{n} \left( P_h^k - P_{h, s_h^k, a_h^k, b_h^k} \right) V_{h+1}^\star \right|$$

$$\lesssim \sqrt{ HSAB \left( HT + \sqrt{H^7 SABT} + H^4 SAB \right) \log^4 \frac{SABT}{\delta} + H^2 SAB \log^2 \frac{T}{\delta} }$$

$$\lesssim \sqrt{ HSAB \left( HT + H^6 SAB + H^4 SAB \right) \log^4 \frac{SABT}{\delta} + H^2 SAB \log^2 \frac{T}{\delta} }$$

$$\lesssim \sqrt{ H^2 SABT \log^4 \frac{SABT}{\delta} + H^{3.5} SAB \log^2 \frac{SABT}{\delta} }, \tag{223}$$

where the second line holds due to (205) and (207), and the last line is valid since

$$HT + \sqrt{H^7 SABT} = HT + \sqrt{H^6 SAB} \cdot \sqrt{HT} \lesssim HT + H^6 SAB$$

due to the Cauchy-Schwarz inequality.

- Turning attention the third term of (221), we need to properly cope with the dependency between $P_h^k$ and $V_{h+1}^{\mathrm{R}, K}$. Towards this, we shall resort to the standard epsilon-net argument (see, e.g., (Tao, 2012)), which will be presented in Appendix E.3.1. The final bound reads like

$$\left| \sum_{h=1}^H \sum_{k=1}^K \sum_{n=N_h^k(s_h^k, a_h^k, b_h^k)}^{N_h^{K-1}(s_h^k, a_h^k, b_h^k)} \frac{\lambda_h^k}{n} \left( P_h^k - P_{h, s_h^k, a_h^k, b_h^k} \right) \left( \overline{V}_{h+1}^{\mathrm{R}, K} - V_{h+1}^\star \right) \right|$$

$$\lesssim H^{3.5} SAB \log^2 \frac{SABT}{\delta} + \sqrt{ H^3 SABK \log^3 \frac{SABT}{\delta} }. \tag{224}$$

- Combining (222), (223), and (224) with (221), we can use the union bound to demonstrate that

$$\mathcal{D}_{3,2} \leq C_{3,2}\left\{H^6 SAB \log^3 \frac{SABT}{\delta} + \sqrt{H^2 SABT \log^4 \frac{SABT}{\delta}}\right\} \qquad (225)$$

with probability at least $1-\delta$, where $C_{3,2} > 0$ is some constant.

Similarly, $\mathcal{D}_{3,4} \leq C_{3,4}\left\{H^6 SAB \log^3 \frac{SABT}{\delta} + \sqrt{H^2 SABT \log^4 \frac{SABT}{\delta}}\right\}$ with probability at least $1-\delta$, where $C_{3,4} > 0$ is some constant.

**Step 3: final bound of $\mathcal{D}_3$.** Putting the above results (220) and (225) together, we immediately arrive at

$$\mathcal{D}_3 \leq |\mathcal{D}_{3,1}| + \mathcal{D}_{3,2} + |\mathcal{D}_{3,3}| + \mathcal{D}_{3,4} \leq C_{\mathrm{r},3}\left\{H^6 SAB \log^3 \frac{SABT}{\delta} + \sqrt{H^2 SABT \log^4 \frac{SABT}{\delta}}\right\}$$
$$(226)$$

with probability at least $1-2\delta$, where $C_{\mathrm{r},3} > 0$ is some constant. This immediately concludes the proof.

### E.3.1 PROOF OF (224)

**Step 1: concentration bounds for a fixed group of vectors.** Consider a fixed group of vectors $\{V^{\mathrm{d}}_{h+1} \in \mathbb{R}^S \mid 1 \leq h \leq H\}$ obeying the following properties:

$$V^\star_{h+1} \leq V^{\mathrm{d}}_{h+1} \leq H \qquad \text{for } 1 \leq h \leq H. \qquad (227)$$

We intend to control the following sum

$$\sum_{h=1}^{H}\sum_{k=1}^{K}\sum_{n=N_h^k(s_h^k,a_h^k,b_h^k)}^{N_h^{K-1}(s_h^k,a_h^k,b_h^k)} \frac{\lambda_h^k}{n}\big(P_h^k - P_{h,s_h^k,a_h^k,b_h^k}\big)\big(V^{\mathrm{d}}_{h+1} - V^\star_{h+1}\big).$$

To do so, we shall resort to Lemma 8. For the moment, let us take $N(s,a,h) := N_h^{K-1}(s,a,b)$ and

$$F^k_{h+1} := V^{\mathrm{d}}_{h+1} - V^\star_{h+1}, \qquad \text{and} \qquad G^k_h(s_h^k,a_h^k) := \sum_{n=N_h^k(s_h^i,a_h^i,b_h^i)}^{N(s_h^k,a_h^k,b_h^k,h)} \frac{\lambda_h^k}{n}.$$

It is easy to see that

$$\big|G^k_h(s_h^k,a_h^k)\big| \leq e \sum_{n=N_h^k(s_h^k,a_h^k,b_h^k)}^{N(s_h^k,a_h^k,b_h^k,h)} \frac{1}{n} \leq e \log T =: C_{\mathrm{g}} \qquad \text{and} \qquad \|F^k_{h+1}\|_\infty \leq H =: C_{\mathrm{f}},$$

which hold due to the facts $\sum_{n=N_h^k(s_h^k,a_h^k,b_h^k)}^{N_h^K(s_h^k,a_h^k,b_h^k)} \frac{1}{n} \leq \log T$ and $\lambda_h^k \leq e$ (cf. (212)) as well as the property that $V^{\mathrm{d}}_{h+1}(s), V^\star_{h+1}(s) \in [0,H]$. Thus, invoking Lemma 8 yields

$$\left|\sum_{h=1}^{H}\sum_{k=1}^{K}\sum_{n=N_h^k(s_h^k,a_h^k,b_h^k)}^{N_h^{K-1}(s_h^k,a_h^k,b_h^k)} \frac{\lambda_h^k}{n}\big(P_h^k - P_{h,s_h^k,a_h^k,b_h^k}\big)\big(V^{\mathrm{d}}_{h+1} - V^\star_{h+1}\big)\right| = \left|\sum_{h=1}^{H}\sum_{k=1}^{K} X_{k,h}\right|$$

$$\lesssim \sqrt{C_{\mathrm{g}}^2 C_{\mathrm{f}} \sum_{h=1}^{H}\sum_{i=1}^{K} \mathbb{E}_{i,h-1}\big[P_h^i F^k_{h+1}\big] \log \frac{K^{HSAB}}{\delta_0}} + C_{\mathrm{g}} C_{\mathrm{f}} \log \frac{K^{HSAB}}{\delta_0}$$

$$\lesssim \sqrt{H \sum_{h=1}^{H}\sum_{i=1}^{K} P_{h,s_h^k,a_h^k,b_h^k}\big(V^{\mathrm{d}}_{h+1} - V^\star_{h+1}\big)\big(\log^2 T\big) \log \frac{K^{HSAB}}{\delta_0}} + H\big(\log T\big) \log \frac{K^{HSAB}}{\delta_0}$$
$$(228)$$

with probability at least $1-\delta_0$, where the choice of $\delta_0$ will be revealed momentarily.

**Step 2: constructing and controlling an epsilon net.** Our argument in Step 1 is only applicable to a fixed group of vectors. The next step is then to construct an epsilon net that allows one to cover the set of interest. Specifically, let us construct an epsilon net $\mathcal{N}_{h+1,\alpha}$ (the value of $\alpha$ will be specified shortly) for each $h \in [H]$ such that:

a) For any $V_{h+1} \in [0, H]^S$, one can find a point $V_{h+1}^{\text{net}} \in \mathcal{N}_{h+1,\alpha}$ obeying

$$0 \leq V_{h+1}(s) - V_{h+1}^{\text{net}}(s) \leq \alpha \qquad \text{for all } s \in \mathcal{S};$$

b) Its cardinality obeys

$$\left| \mathcal{N}_{h+1,\alpha} \right| \leq \left( \frac{H}{\alpha} \right)^S. \tag{229}$$

Clearly, this also means that

$$\left| \mathcal{N}_{2,\alpha} \times \mathcal{N}_{3,\alpha} \times \cdots \times \mathcal{N}_{H+1,\alpha} \right| \leq \left( \frac{H}{\alpha} \right)^{SH}.$$

Set $\delta_0 = \frac{1}{6}\delta / \left( \frac{H}{\alpha} \right)^{SH}$. Taking (228) together the union bound implies that: with probability at least $1 - \delta_0 \left( \frac{H}{\alpha} \right)^{SH} = 1 - \delta/6$, one has

$$\left| \sum_{h=1}^{H} \sum_{k=1}^{K} \sum_{n=N_h^k(s_h^k, a_h^k, b_h^k)}^{N_h^{K-1}(s_h^k, a_h^k, b_h^k)} \frac{\lambda_h^k}{n} \left( P_h^k - P_{h, s_h^k, a_h^k, b_h^k} \right) \left( V_{h+1}^{\text{net}} - V_{h+1}^{\star} \right) \right|$$

$$\lesssim \sqrt{ H \sum_{h=1}^{H} \sum_{i=1}^{K} P_{h, s_h^k, a_h^k, b_h^k} \left( V_{h+1}^{\text{net}} - V_{h+1}^{\star} \right) \left( \log^2 T \right) \log \frac{K^{HSAB}}{\delta_0} + H(\log T) \log \frac{K^{HSAB}}{\delta_0} }$$

$$\lesssim \sqrt{ H^2 SAB \sum_{h=1}^{H} \sum_{i=1}^{K} P_{h, s_h^k, a_h^k, b_h^k} \left( V_{h+1}^{\text{net}} - V_{h+1}^{\star} \right) \left( \log^2 T \right) \log \frac{SAT}{\delta\alpha} + H^2 SAB \log^2 \frac{SABT}{\delta\alpha} } \tag{230}$$

simultaneously for all $\{ V_{h+1}^{\text{net}} \mid 1 \leq h \leq H \}$ obeying $V_{h+1}^{\text{d}} \in \mathcal{N}_{h+1,\alpha}$ ($h \in [H]$).

**Step 3: obtaining uniform bounds.** We are now positioned to establish a uniform bound over the entire set of interest. Consider an arbitrary group of vectors $\{ V_{h+1}^{\text{u}} \in \mathbb{R}^S \mid 1 \leq h \leq H \}$ obeying (227). By construction, one can find a group of points $\{ V_{h+1}^{\text{net}} \in \mathcal{N}_{h+1,\alpha} \mid h \in [H] \}$ such that

$$0 \leq V_{h+1}^{\text{u}}(s) - V_{h+1}^{\text{net}}(s) \leq \alpha \qquad \text{for all } (h, s) \in \mathcal{S} \times [H]. \tag{231}$$

It is readily seen that

$$\left| \sum_{k=1}^{K} \sum_{n=N_h^k(s_h^k, a_h^k, b_h^k)}^{N_h^{K-1}(s_h^k, a_h^k, b_h^k)} \frac{\lambda_h^k}{n} \left( P_h^k - P_{h, s_h^k, a_h^k, b_h^k} \right) \left( V_{h+1}^{\text{u}} - V_{h+1}^{\text{net}} \right) \right|$$

$$\leq \left| \sum_{k=1}^{K} \sum_{n=N_h^k(s_h^k, a_h^k, b_h^k)}^{N_h^{K-1}(s_h^k, a_h^k, b_h^k)} \frac{\lambda_h^k}{n} \left( \left\| P_h^k \right\|_1 + \left\| P_{h, s_h^k, a_h^k, b_h^k} \right\|_1 \right) \left\| V_{h+1}^{\text{u}} - V_{h+1}^{\text{net}} \right\|_\infty \right|$$

$$\leq 2eK\alpha \log T, \tag{232}$$

where the last inequality follows from $\sum_{n=N_h^i(s_h^i,a_h^i,b_h^i)}^{N_h^{K-1}(s_h^i,a_h^i,b_h^i)} \frac{1}{n} \le \log T$ and $\lambda_h^k \le e$ (cf. (212)). Consequently, by taking $\alpha = 1/(SABT)$, we can deduce that

$$\left| \sum_{h=1}^{H} \sum_{k=1}^{K} \sum_{n=N_h^k(s_h^k,a_h^k,b_h^k)}^{N_h^{K-1}(s_h^k,a_h^k,b_h^k)} \frac{\lambda_h^k}{n} \left( P_h^k - P_{h,s_h^k,a_h^k,b_h^k} \right) \left( V_{h+1}^{\mathrm{u}} - V_{h+1}^{\star} \right) \right|$$

$$\le \left| \sum_{h=1}^{H} \sum_{k=1}^{K} \sum_{n=N_h^k(s_h^k,a_h^k,b_h^k)}^{N_h^{K-1}(s_h^k,a_h^k,b_h^k)} \frac{\lambda_h^k}{n} \left( P_h^k - P_{h,s_h^k,a_h^k,b_h^k} \right) \left( V_{h+1}^{\mathrm{net}} - V_{h+1}^{\star} \right) \right|$$

$$+ \sum_{h=1}^{H} \left| \sum_{k=1}^{K} \sum_{n=N_h^k(s_h^k,a_h^k,b_h^k)}^{N_h^{K-1}(s_h^k,a_h^k,b_h^k)} \frac{\lambda_h^k}{n} \left( P_h^k - P_{h,s_h^k,a_h^k,b_h^k} \right) \left( V_{h+1}^{\mathrm{u}} - V_{h+1}^{\mathrm{net}} \right) \right|$$

$$\lesssim \left| \sum_{h=1}^{H} \sum_{k=1}^{K} \sum_{n=N_h^k(s_h^k,a_h^k,b_h^k)}^{N_h^{K-1}(s_h^k,a_h^k,b_h^k)} \frac{\lambda_h^k}{n} \left( P_h^k - P_{h,s_h^k,a_h^k,b_h^k} \right) \left( V_{h+1}^{\mathrm{net}} - V_{h+1}^{\star} \right) \right| + HK\alpha \log T$$

$$\lesssim \sqrt{H^2 SAB \sum_{h=1}^{H} \sum_{i=1}^{K} P_{h,s_h^k,a_h^k,b_h^k} \left( V_{h+1}^{\mathrm{net}} - V_{h+1}^{\star} \right) \left( \log^2 T \right) \log \frac{SABT}{\delta\alpha}}$$

$$\qquad + H^2 SAB \log^2 \frac{SABT}{\delta\alpha} + HK\alpha \log T$$

$$\asymp \sqrt{H^2 SAB \sum_{h=1}^{H} \sum_{i=1}^{K} P_{h,s_h^k,a_h^k,b_h^k} \left( V_{h+1}^{\mathrm{u}} - V_{h+1}^{\star} \right) \left( \log^2 T \right) \log \frac{SABT}{\delta} + H^2 SAB \log^2 \frac{SABT}{\delta}},$$
(233)

where the last line holds due to the condition (231 and our choice of $\alpha$. To summarize, with probability exceeding $1 - \delta/6$, the property (233) holds simultaneously for all $\{V_{h+1}^{\mathrm{u}} \in \mathbb{R}^S \mid 1 \le h \le H\}$ obeying (227).

**Step 4: controlling the original term of interest.** With the above union bound in hand, we are ready to control the original term of interest

$$\sum_{h=1}^{H} \sum_{k=1}^{K} \sum_{n=N_h^k(s_h^k,a_h^k,b_h^k)}^{N_h^{K-1}(s_h^k,a_h^k,b_h^k)} \frac{\lambda_h^k}{n} \left( P_h^k - P_{h,s_h^k,a_h^k,b_h^k} \right) \left( \overline{V}_{h+1}^{\mathrm{R},K} - V_{h+1}^{\star} \right).$$
(234)

To begin with, it can be easily verified using (26) that

$$V_{h+1}^{\star} \le \overline{V}_{h+1}^{\mathrm{R},K} \le H \qquad \text{for all } 1 \le h \le H.$$
(235)

Moreover, we make the observation that

$$\sum_{h=1}^{H} \sum_{k=1}^{K} P_{h,s_h^k,a_h^k,b_h^k} \left( \overline{V}_{h+1}^{\mathrm{R},K} - V_{h+1}^{\star} \right) \overset{(i)}{\le} \sum_{h=1}^{H} \sum_{k=1}^{K} P_{h,s_h^k,a_h^k,b_h^k} \left( \overline{V}_{h+1}^{\mathrm{R},k} - V_{h+1}^{\star} \right)$$

$$\overset{(ii)}{\le} \sqrt{H^6 SABK \log \frac{SABT}{\delta}} + H^3 SAB + HK$$
(236)

with probability exceeding $1 - \delta/6$, where (i) holds because $\overline{V}_{h+1}^{\mathrm{R}}$ is monotonically non-increasing (in view of the monotonicity of $\overline{V}_h(s)$ in (23) and the update rule in line 17 of Algorithm 3), and (ii)

follows from (219). Substitution into (233) yields

$$\left| \sum_{h=1}^{H} \sum_{k=1}^{K} \sum_{n=N_h^k(s_h^k,a_h^k,b_h^k)}^{N_h^{K-1}(s_h^k,a_h^k,b_h^k)} \frac{\lambda_h^k}{n} \left( P_h^k - P_{h,s_h^k,a_h^k,b_h^k} \right) \left( \overline{V}_{h+1}^{\mathrm{R},K} - V_{h+1}^\star \right) \right|$$

$$\lesssim \sqrt{ H^2 SAB \sum_{h=1}^{H} \sum_{i=1}^{K} P_{h,s_h^k,a_h^k,b_h^k} \left( \overline{V}_{h+1}^{\mathrm{R},K} - V_{h+1}^\star \right) \left( \log^2 T \right) \log \frac{SABT}{\delta} + H^2 SAB \log^2 \frac{SABT}{\delta} }$$

$$\lesssim \sqrt{ H^2 SAB \left\{ \sqrt{ H^6 SABK \log \frac{SABT}{\delta} } + H^3 SAB + HK \right\} \left( \log^2 T \right) \log \frac{SABT}{\delta} }$$

$$+ H^2 SAB \log^2 \frac{SABT}{\delta}.$$

According to the fact that

$$\sqrt{ H^6 SABK \log \frac{SABT}{\delta} } = \sqrt{ H^5 SAB \log \frac{SABT}{\delta} } \sqrt{HK} \lesssim H^5 SAB \log \frac{SABT}{\delta} + HK,$$

we have

$$\left| \sum_{h=1}^{H} \sum_{k=1}^{K} \sum_{n=N_h^k(s_h^k,a_h^k,b_h^k)}^{N_h^{K-1}(s_h^k,a_h^k,b_h^k)} \frac{\lambda_h^k}{n} \left( P_h^k - P_{h,s_h^k,a_h^k,b_h^k} \right) \left( \overline{V}_{h+1}^{\mathrm{R},K} - V_{h+1}^\star \right) \right|$$

$$\lesssim \sqrt{ H^2 SAB \left\{ H^5 SAB \log \frac{SABT}{\delta} + H^3 SAB + HK \right\} \log^3 \frac{SABT}{\delta} } + H^2 SAB \log^2 \frac{SABT}{\delta}$$

$$\lesssim H^{3.5} SAB \log^2 \frac{SABT}{\delta} + \sqrt{ H^3 SABK \log^3 \frac{SABT}{\delta} }. \tag{237}$$

## F MULTIPLAYER GENERAL-SUM MARKOV GAMES

In this section, we extend ME-Nash-QL to the setting of multiplayer general-sum Markov games and present the corresponding theoretical guarantees.

### F.1 PROBLEM FORMULATION

A general-sum Markov game (general-sum MG) is a tuple $\mathcal{M}(\mathcal{S}, \{\mathcal{A}_i\}_{i=1}^m, H, \{P_h\}_{h=1}^H, \{r_i\}_{i=1}^m)$ with $m$ players, where $\mathcal{S}$ denotes the state space and $H$ is the horizon length. We have $m$ different action spaces, where $\mathcal{A}_i$ is the action space for the $i^{\text{th}}$ player and $|\mathcal{A}_i| = A_i$. We let $\mathcal{A} = \mathcal{A}_1 \times \cdots \times \mathcal{A}_m$ denote the joint action space, and let $\boldsymbol{a} := (a_1, \cdots, a_m) \in \mathcal{A}$ denote the (tuple of) joint actions by all $m$ players. $\{P_h\}_{h \in [H]}$ is a collection of transition matrices, so that $P_h(\cdot|s, \boldsymbol{a})$ gives the distribution of the next state if actions $\boldsymbol{a}$ are taken at state $s$ at step $h$, and $r_i = \{r_{h,i}\}_{h \in [H]}$ is a collection of reward functions for the $i^{\text{th}}$ player, so that $r_{h,i}(s, \boldsymbol{a})$ gives the reward received by the $i^{\text{th}}$ player if actions $\boldsymbol{a}$ are taken at state $s$ at step $h$.

The policy of the $i^{\text{th}}$ player is denoted as $\pi_i := \left\{ \pi_{h,i} : \mathcal{S} \to \Delta_{\mathcal{A}_i} \right\}_{h \in [H]}$. We denote the product policy of all players as $\pi := \pi_1 \times \cdots \times \pi_M$, and denote the policy of all players except the $i^{\text{th}}$ player as $\pi_{-i}$. We define $V_{h,i}^\pi(s)$ as the expected cumulative reward that will be received by the $i^{\text{th}}$ player if starting at state $s$ at step $h$ and all players follow policy $\pi$. For any strategy $\pi_{-i}$, there also exists a *best response* of the $i^{\text{th}}$ player, which is a policy $\mu^\dagger(\pi_{-i})$ satisfying $V_{h,i}^{\mu^\dagger(\pi_{-i}),\pi_{-i}}(s) = \sup_{\pi_i} V_{h,i}^{\pi_i,\pi_{-i}}(s)$ for any $(s, h) \in \mathcal{S} \times [H]$. For convenience, we denote $V_{h,i}^{\dagger,\pi_{-i}} := V_{h,i}^{\mu^\dagger(\pi_{-i}),\pi_{-i}}$. The $Q$-functions of the best response can be defined similarly.

In general, there are three versions of equilibrium for general-sum MGs: Nash equilibrium (NE), correlated equilibrium (CE), and coarse correlated equilibrium (CCE), all being standard solution

notions in games (Nisan et al., 2007). These three notions coincide in two-player zero-sum games, but are not equivalent to each other in multi-player general-sum games; any one of them could be desired depending on the application at hand. In this section, we first consider CCE and then extend to NE and CE in the final of Section F.2.

The CCE is a relaxed version of Nash equilibrium in which we consider general correlated policies instead of product policies.

**Definition 4 (CCE in general-sum MGs)** *A (correlated) policy* $\pi := \{\pi_h(s) \in \Delta_{\mathcal{A}} : (h, s) \in [H] \times \mathcal{S}\}$ *is a **CCE** if* $\max_{i \in [m]} V_{h,i}^{\dagger, \pi_{-i}}(s) \leq V_{h,i}^{\pi}(s)$ *for all* $(s, h) \in \mathcal{S} \times [H]$.

Compared with a Nash equilibrium, a CCE is not necessarily a product policy, that is, we may not have $\pi_h(s) \in \Delta_{\mathcal{A}_1} \times \cdots \times \Delta_{\mathcal{A}_m}$. Similarly, we also define $\epsilon$-approximate CCE and CCE-regret as below.

**Definition 5 ($\epsilon$-approximate CCE in general-sum MGs)** *A policy* $\pi := \{\pi_h(s) \in \Delta_{\mathcal{A}} : (h, s) \in [H] \times \mathcal{S}\}$ *is an $\epsilon$-approximate CCE if* $\frac{1}{K} \sum_{k=1}^{K} \max_{i \in [m]} (V_{1,i}^{\dagger, \pi_{-i}} - V_{1,i}^{\pi})(s_1^k) \leq \epsilon$.

---

**Algorithm 4:** Multi-player Memory-Efficient Nash Q-Learning (Multi-ME-Nash-QL)

---

1 **Parameter:** some universal constant $c_{\mathrm{b}} > 0$ and probability of failure $\delta \in (0, 1)$

2 **Initialize:** $\overline{Q}_{h,i}(s, \boldsymbol{a}), Q_{h,i}^{\mathrm{UCB}}(s, \boldsymbol{a}), \overline{Q}_{h,i}^{\mathrm{R}}(s, \boldsymbol{a}), \leftarrow H; \underline{Q}_{h,i}(s, \boldsymbol{a}), \underline{Q}_{h,i}^{\mathrm{R}}(s, \boldsymbol{a}), Q_{h,i}^{\mathrm{LCB}}(s, \boldsymbol{a}) \leftarrow 0;$

   $\overline{V}_{h,i}(s), \overline{V}_{h,i}^{\mathrm{R}}(s) \leftarrow H; \underline{V}_{h,i}(s), \underline{V}_{h,i}^{\mathrm{R}}(s) \leftarrow 0; N_h(s, \boldsymbol{a}) \leftarrow 0; \overline{\phi}_{h,i}^{\mathrm{r}}(s, \boldsymbol{a}), \underline{\phi}_{h,i}^{\mathrm{r}}(s, \boldsymbol{a}), \overline{\psi}_{h,i}^{\mathrm{r}}(s, \boldsymbol{a}),$

   $\underline{\psi}_{h,i}^{\mathrm{r}}(s, \boldsymbol{a}), \overline{\phi}_{h,i}^{\mathrm{a}}(s, \boldsymbol{a}), \underline{\phi}_{h,i}^{\mathrm{a}}(s, \boldsymbol{a}), \overline{\psi}_{h,i}^{\mathrm{a}}(s, \boldsymbol{a}), \underline{\psi}_{h,i}^{\mathrm{a}}(s, \boldsymbol{a}), \overline{\varphi}_{h,i}^{\mathrm{R}}(s, \boldsymbol{a}), \underline{\varphi}_{h,i}^{\mathrm{R}}(s, \boldsymbol{a}), \overline{B}_{h,i}^{\mathrm{R}}(s, \boldsymbol{a}),$

   $\underline{B}_{h,i}^{\mathrm{R}}(s, \boldsymbol{a}) \leftarrow 0;$ and $u_{\mathrm{r},i}(s) = \mathsf{True}$ for all $(s, \boldsymbol{a}, h) \in \mathcal{S} \times \mathcal{A} \times [H]$.

3 **for** *Episode* $k = 1, \ldots, K$ **do**

4 $\quad$ Set initial state $s_1 \leftarrow s_1^k$.

5 $\quad$ **for** *Step* $h = 1, \ldots, H$ **do**

6 $\quad\quad$ Take action $\boldsymbol{a}_h \sim \pi_h(\cdot | s_h)$, and draw $s_{h+1} \sim P_h(\cdot \mid s_h, \boldsymbol{a}_h)$.

7 $\quad\quad$ $N_h(s_h, \boldsymbol{a}_h) \leftarrow N_h(s_h, \boldsymbol{a}_h) + 1; \quad n \leftarrow N_h(s_h, \boldsymbol{a}_h); \quad \eta_n \leftarrow \frac{H+1}{H+n}.$

8 $\quad\quad$ **for** *player* $i = 1, 2, \cdots, m$ **do**

9 $\quad\quad\quad$ $\left[Q_{h,i}^{\mathrm{UCB}}, Q_{h,i}^{\mathrm{LCB}}\right](s_h, \boldsymbol{a}_h) \leftarrow \text{update-q}\,().$

10 $\quad\quad\quad$ $\overline{Q}_{h,i}^{\mathrm{R}}(s_h, \boldsymbol{a}_h) \leftarrow \text{update-ur}\,().$

11 $\quad\quad\quad$ $\underline{Q}_{h,i}^{\mathrm{R}}(s_h, \boldsymbol{a}_h) \leftarrow \text{update-lr}\,().$

12 $\quad\quad\quad$ $\overline{Q}_{h,i}(s_h, \boldsymbol{a}_h) \leftarrow \min\{\overline{Q}_{h,i}^{\mathrm{R}}(s_h, \boldsymbol{a}_h), Q_{h,i}^{\mathrm{UCB}}(s_h, \boldsymbol{a}_h), \overline{Q}_{h,i}(s_h, \boldsymbol{a}_h)\}.$

13 $\quad\quad\quad$ $\underline{Q}_{h,i}(s_h, \boldsymbol{a}_h) \leftarrow \max\{\underline{Q}_{h,i}^{\mathrm{R}}(s_h, \boldsymbol{a}_h), Q_{h,i}^{\mathrm{LCB}}(s_h, \boldsymbol{a}_h), \underline{Q}_{h,i}(s_h, \boldsymbol{a}_h)\}.$

14 $\quad\quad\quad$ **if** $\overline{Q}_{h,i}(s_h, \boldsymbol{a}_h) = \min\{\overline{Q}_{h,i}^{\mathrm{R}}(s_h, \boldsymbol{a}_h), Q_{h,i}^{\mathrm{UCB}}(s_h, \boldsymbol{a}_h)\}$ *and*

   $\quad\quad\quad\quad \underline{Q}_{h,i}(s_h, \boldsymbol{a}_h) = \max\{\underline{Q}_{h,i}^{\mathrm{R}}(s_h, \boldsymbol{a}_h), Q_{h,i}^{\mathrm{LCB}}(s_h, \boldsymbol{a}_h)\}$ **then**

15 $\quad\quad\quad\quad$ $\pi_h(\cdot | s_h) \leftarrow \text{CCE}(\overline{Q}_{h,1}(s_h, \cdot), \cdots, \underline{Q}_{h,m}(s_h, \cdot)).$

16 $\quad\quad\quad$ $\overline{V}_{h,i}(s_h) \leftarrow \min\{(\mathbb{D}_{\pi_h}\overline{Q}_{h,i})(s_h), \overline{V}_{h,i}(s_h)\};$

17 $\quad\quad\quad$ $\underline{V}_{h,i}(s_h) \leftarrow \max\{(\mathbb{D}_{\pi_h}\underline{Q}_{h,i})(s_h), \underline{V}_{h,i}(s_h)\}.$

18 $\quad\quad\quad$ **if** $\overline{V}_{h,i}(s_h) - \underline{V}_{h,i}(s_h) > 1$ **then**

19 $\quad\quad\quad\quad$ $\overline{V}_{h,i}^{\mathrm{R}}(s_h) \leftarrow \overline{V}_{h,i}(s_h); \quad \underline{V}_{h,i}^{\mathrm{R}}(s_h) \leftarrow \underline{V}_{h,i}(s_h).$

20 $\quad\quad\quad$ **else if** $u_{\mathrm{r},i}(s_h) = \mathsf{True}$ **then**

21 $\quad\quad\quad\quad$ $\overline{V}_{h,i}^{\mathrm{R}}(s_h) \leftarrow \overline{V}_{h,i}(s_h); \quad \underline{V}_h^{\mathrm{R}}(s_h) \leftarrow \underline{V}_{h,i}(s_h); \quad u_{\mathrm{r},i}(s_h) = \mathsf{False}.$

22 **Output:** $\{\pi_h\}_{h=1}^{H}$.

---

**Definition 6 (CCE-regret in general-sum MGs)** *Let policy $\pi^k$ denote the (correlated) policy deployed by the algorithm in the $k^{th}$ episode. After a total of $K$ episodes, the regret is defined as*

$$\text{Regret}(K) = \sum_{k=1}^{K} \max_{i \in [m]} (V_{1,i}^{\dagger,\pi^k_{-i}} - V_{1,i}^{\pi^k})(s_1^k). \tag{238}$$

In addition, for general-sum MGs, we have $\{\text{NE}\} \subseteq \{\text{CE}\} \subseteq \{\text{CCE}\}$, so that they form a nested set of notions of equilibria (Nisan et al., 2007). Finally, since a Nash equilibrium always exists, so a CE and CCE equilibrium always exist.

---

**Algorithm 5:** Auxiliary functions of Multi-ME-Nash-QL

---

1 **Function** $\texttt{update-q}\Big(\Big[Q_{h,i}^{\text{UCB}}, Q_{h,i}^{\text{LCB}}\Big](s_h, \boldsymbol{a}_h), \Big[\overline{V}_{h+1,i}, \underline{V}_{h+1,i}\Big](s_{h+1})\Big)\textbf{:}$

2 $\quad Q_{h,i}^{\text{UCB}}(s_h, \boldsymbol{a}_h) \leftarrow$
$\quad\quad (1-\eta_n)\, Q_{h,i}^{\text{UCB}}(s_h, \boldsymbol{a}_h) + \eta_n \left( r_{h,i}(s_h, \boldsymbol{a}_h) + \overline{V}_{h+1,i}(s_{h+1}) + c_b\sqrt{\frac{1}{n}H^3 \log \frac{S\prod_{i=1}^{m}A_iT}{\delta}} \right);$

3 $\quad Q_{h,i}^{\text{LCB}}(s_h, \boldsymbol{a}_h) \leftarrow$
$\quad\quad (1-\eta_n)\, Q_{h,i}^{\text{LCB}}(s_h, \boldsymbol{a}_h) + \eta_n \left( r_{h,i}(s_h, \boldsymbol{a}_h) + \underline{V}_{h+1,i}(s_{h+1}) - c_b\sqrt{\frac{1}{n}H^3 \log \frac{S\prod_{i=1}^{m}A_iT}{\delta}} \right).$

4 **Function** $\texttt{update-ur}\Big(\Big[\overline{\phi}_{h,i}^{\text{r}}, \overline{\psi}_{h,i}^{\text{r}}, \overline{\phi}_{h,i}^{\text{a}}, \overline{\psi}_{h,i}^{\text{a}}, \overline{B}_{h,i}^{\text{R}}, \overline{Q}_{h,i}^{\text{R}}\Big](s_h, \boldsymbol{a}_h), \Big[\overline{V}_{h+1,i}^{\text{R}}, \overline{V}_{h+1,i}\Big](s_{h+1})\Big)\textbf{:}$

5 $\quad \Big[\overline{\phi}_{h,i}^{\text{r}}(s_h, \boldsymbol{a}_h), \overline{b}_{h,i}^{\text{R}}\Big] \leftarrow$
$\quad\quad \texttt{update-q-bonus}\Big(\Big[\overline{\phi}_{h,i}^{\text{r}}, \overline{\psi}_{h,i}^{\text{r}}, \overline{\phi}_{h,i}^{\text{a}}, \overline{\psi}_{h,i}^{\text{a}}, \overline{B}_{h,i}^{\text{R}}\Big](s_h, \boldsymbol{a}_h), \Big[\overline{V}_{h+1,i}^{\text{R}}, \overline{V}_{h+1,i}\Big](s_{h+1})\Big);$

6 $\quad \overline{Q}_{h,i}^{\text{R}}(s_h, \boldsymbol{a}_h) \leftarrow (1 - \eta_n)\, \overline{Q}_{h,i}^{\text{R}}(s_h, \boldsymbol{a}_h) +$
$\quad\quad \eta_n \left( r_{h,i}(s_h, \boldsymbol{a}_h) + \overline{V}_{h+1,i}(s_{h+1}) - \overline{V}_{h+1,i}^{\text{R}}(s_{h+1}) + \overline{\phi}_{h,i}^{\text{r}}(s_h, \boldsymbol{a}_h) + \overline{b}_{h,i}^{\text{R}} \right).$

7 **Function**
$\texttt{update-lr}\Big(\Big[\underline{\phi}_{h,i}^{\text{r}}, \underline{\psi}_{h,i}^{\text{r}}, \underline{\phi}_{h,i}^{\text{a}}, \underline{\psi}_{h,i}^{\text{a}}, \underline{B}_{h,i}^{\text{R}}, \underline{Q}_{h,i}^{\text{R}}\Big](s_h, \boldsymbol{a}_h), \Big[\underline{V}_{h+1,i}^{\text{R}}, \underline{V}_{h+1,i}\Big](s_{h+1})\Big)\textbf{:}$

8 $\quad \Big[\underline{\phi}_{h,i}^{\text{r}}(s_h, \boldsymbol{a}_h), \underline{b}_{h,i}^{\text{R}}\Big] \leftarrow$
$\quad\quad \texttt{update-q-bonus}\Big(\Big[\underline{\phi}_{h,i}^{\text{r}}, \underline{\psi}_{h,i}^{\text{r}}, \underline{\phi}_{h,i}^{\text{a}}, \underline{\psi}_{h,i}^{\text{a}}, \underline{B}_{h,i}^{\text{R}}\Big](s_h, \boldsymbol{a}_h), \Big[\underline{V}_{h+1,i}^{\text{R}}, \underline{V}_{h+1,i}\Big](s_{h+1})\Big);$

9 $\quad \underline{Q}_{h,i}^{\text{R}}(s_h, \boldsymbol{a}_h) \leftarrow (1 - \eta_n)\, \underline{Q}_{h,i}^{\text{R}}(s_h, \boldsymbol{a}_h) +$
$\quad\quad \eta_n \left( r_{h,i}(s_h, \boldsymbol{a}_h) + \underline{V}_{h+1,i}(s_{h+1}) - \underline{V}_{h+1,i}^{\text{R}}(s_{h+1}) + \underline{\phi}_{h,i}^{\text{r}}(s_h, \boldsymbol{a}_h) - \underline{b}_{h,i}^{\text{R}} \right).$

10 **Function**
$\texttt{update-q-bonus}\Big(\Big[\phi_{h,i}^{\text{r}}, \psi_{h,i}^{\text{r}}, \phi_{h,i}^{\text{a}}, \psi_{h,i}^{\text{a}}, B_{h,i}^{\text{R}}\Big](s_h, \boldsymbol{a}_h), \Big[V_{h+1,i}^{\text{R}}, V_{h+1,i}\Big](s_{h+1})\Big)\textbf{:}$

11 $\quad \phi_{h,i}^{\text{r}}(s_h, \boldsymbol{a}_h) \leftarrow \left(1 - \frac{1}{n}\right)\phi_{h,i}^{\text{r}}(s_h, \boldsymbol{a}_h) + \frac{1}{n}V_{h+1,i}^{\text{R}}(s_{h+1});$

12 $\quad \psi_{h,i}^{\text{r}}(s_h, \boldsymbol{a}_h) \leftarrow \left(1 - \frac{1}{n}\right)\psi_{h,i}^{\text{r}}(s_h, \boldsymbol{a}_h) + \frac{1}{n}\left(V_{h+1,i}^{\text{R}}(s_{h+1})\right)^2;$

13 $\quad \phi_{h,i}^{\text{a}}(s_h, \boldsymbol{a}_h) \leftarrow (1 - \eta_n)\phi_{h,i}^{\text{a}}(s_h, \boldsymbol{a}_h) + \eta_n \left(V_{h+1,i}(s_{h+1}) - V_{h+1,i}^{\text{R}}(s_{h+1})\right);$

14 $\quad \psi_{h,i}^{\text{a}}(s_h, \boldsymbol{a}_h) \leftarrow (1 - \eta_n)\psi_{h,i}^{\text{a}}(s_h, \boldsymbol{a}_h) + \eta_n \left(V_{h+1,i}(s_{h+1}) - V_{h+1,i}^{\text{R}}(s_{h+1})\right)^2;$

15 $\quad B_{h,i}^{\text{temp}}(s_h, \boldsymbol{a}_h) \leftarrow c_{\text{b}}\sqrt{\frac{\log^2 \frac{S\prod_{i=1}^{m}A_iT}{\delta}}{n}}\sqrt{\psi_{h,i}^{\text{r}}(s_h, \boldsymbol{a}_h) - \left(\phi_{h,i}^{\text{r}}(s_h, \boldsymbol{a}_h)\right)^2} +$
$\quad\quad c_{\text{b}}\sqrt{\frac{\log^2 \frac{S\prod_{i=1}^{m}A_iT}{\delta}}{n}}\sqrt{H}\sqrt{\psi_{h,i}^{\text{a}}(s_h, \boldsymbol{a}_h) - \left(\phi_{h,i}^{\text{a}}(s_h, \boldsymbol{a}_h)\right)^2};$

16 $\quad \varphi_{h,i}^{\text{R}}(s_h, \boldsymbol{a}_h) \leftarrow B_{h,i}^{\text{temp}}(s_h, \boldsymbol{a}_h) - B_{h,i}^{\text{R}}(s_h, \boldsymbol{a}_h);$

17 $\quad B_{h,i}^{\text{R}}(s_h, \boldsymbol{a}_h) \leftarrow B_{h,i}^{\text{temp}}(s_h, \boldsymbol{a}_h);$

18 $\quad b_{h,i}^{\text{R}} \leftarrow B_{h,i}^{\text{R}}(s_h, \boldsymbol{a}_h) + (1 - \eta_n)\frac{\varphi_{h,i}^{\text{R}}(s_h, \boldsymbol{a}_h)}{\eta_n} + c_{\text{b}}\frac{H^2 \log^2 \frac{S\prod_{i=1}^{m}A_iT}{\delta}}{n^{3/4}}.$

---

## F.2 MULTI-ME-NASH-QL

Here we present the Multi-ME-Nash-QL algorithm in Algorithm 4, which is an extension of Algorithm 1 for multi-player general-sum Markov games.

Remarkably, the CCE operation in line 15 in Algorithm 4 could be replaced by NE or CE operation which finds the NE or CE for *one-step* games. When using NE operation, the worst-case computational complexity will be PPAD-hard. When using CE or CCE, it can be solved in polynomial time using linear programming. However, the policies found are not guaranteed to be a product policy. We remark that in Algorithm 1 we used the CCE subroutine for finding Nash in two-player zero-sum games, which seemingly contrasts the principle of using the NE subroutine for finding the Nash equilibrium, but nevertheless works as the Nash equilibrium and CCE are equivalent in zero-sum games.

## F.3 ANALYSIS OF MULTI-ME-NASH-QL

In this section, we prove Theorem 3. Similar to Lemma 2 in Appendix A.3.1, we can obtain the properties of the Q-estimate and V-estimate, as asserted by the following lemma.

**Lemma 10** *Consider any $\delta \in (0, 1)$. Suppose that $c_{\rm b} > 0$ is some sufficiently large constant. Then with probability at least $1 - \delta$,*

$$\overline{Q}_{h,i}^{{\rm R},k}(s, \boldsymbol{a}) \geq \overline{Q}_{h,i}^k(s, \boldsymbol{a}) \geq Q_{h,i}^{\dagger, \pi_{-i}^k}(s, \boldsymbol{a}), \quad \overline{V}_{h,i}^k(s) \geq V_{h,i}^{\dagger, \pi_{-i}^k}(s), \tag{239}$$

$$\underline{Q}_{h,i}^{{\rm R},k}(s, \boldsymbol{a}) \leq \underline{Q}_{h,i}^k(s, \boldsymbol{a}) \leq Q_{h,i}^{\pi^k}(s, \boldsymbol{a}), \quad \underline{V}_{h,i}^k(s) \leq V_{h,i}^{\pi^k}(s) \tag{240}$$

*hold simultaneously for all $(s, \boldsymbol{a}, h) \in \mathcal{S} \times \mathcal{A} \times [H]$.*

**Step 1: regret decomposition.** Firstly, we can apply Lemma 10 to reformulate (238) as

$$\text{Regret}(K) = \sum_{k=1}^K \max_i \left( V_{1,i}^{\dagger, \pi_{-i}^k} - V_{1,i}^{\pi^k} \right) (s_1^k) \leq \sum_{k=1}^K \max_i \left( \overline{V}_{1,i}^k - \underline{V}_{1,i}^k \right) (s_1^k). \tag{241}$$

According to lines 13-14 in Algorithm 4, we obtain

$$\overline{Q}_{h,i}^{{\rm R},k}(s, \boldsymbol{a}) \geq \overline{Q}_{h,i}^k(s, \boldsymbol{a}), \quad \underline{Q}_{h,i}^{{\rm R},k}(s, \boldsymbol{a}) \leq \underline{Q}_{h,i}^k(s, \boldsymbol{a}), \tag{242}$$

$$Q_{h,i}^{{\rm UCB},k}(s, \boldsymbol{a}) \geq \overline{Q}_{h,i}^k(s, \boldsymbol{a}), \quad Q_{h,i}^{{\rm LCB},k}(s, \boldsymbol{a}) \leq \underline{Q}_{h,i}^k(s, \boldsymbol{a}). \tag{243}$$

Based on this relation, we notice the following propagation by lines 18-21 in Algorithm 4 and update rules in Algorithm 5 with $c_b^{{\rm ULCB}} = c_b \sqrt{\frac{1}{n} H^3 \log \frac{S \prod_{i=1}^m A_i T}{\delta}}$:

$$\begin{cases} \left( \overline{Q}_{h,i}^k - \underline{Q}_{h,i}^k \right) (s, \boldsymbol{a}) \leq \left( Q_{h,i}^{{\rm UCB},k} - Q_{h,i}^{{\rm LCB},k} \right) (s, \boldsymbol{a}), \\ \left( Q_{h,i}^{{\rm UCB},k} - Q_{h,i}^{{\rm LCB},k} \right) (s, \boldsymbol{a}) = (1 - \eta_n) \left( Q_{h,i}^{{\rm UCB},k} - Q_{h,i}^{{\rm LCB},k} \right) (s, \boldsymbol{a}) \\ \qquad + \eta_n \left( \left( \overline{V}_{h+1,i}^k - \underline{V}_{h+1,i}^k \right) (s_{h+1}^k) + 2c_b^{{\rm ULCB}} \right), \\ \left( \overline{V}_{h,i}^k - \underline{V}_{h,i}^k \right) (s) = \left[ \mathbb{D}_{\pi_h} \left( \overline{Q}_{h,i}^k - \underline{Q}_{h,i}^k \right) \right] (s). \end{cases} \tag{244}$$

We can define $\hat{V}_h^k(s)$, $\check{V}_h^k(s)$, $\hat{Q}_h^k(s, \boldsymbol{a})$, $\check{Q}_h^k(s, \boldsymbol{a})$, $\hat{Q}_h^{{\rm UCB},k}(s, \boldsymbol{a})$ and $\check{Q}_h^{{\rm LCB},k}$ recursively by $\hat{V}_{H+1}^k(s) = 0$ and $\check{V}_{H+1}^k(s) = 0$, and there is

$$\begin{cases} \left( \hat{Q}_h^k - \check{Q}_h^k \right) (s, \boldsymbol{a}) = \left( \hat{Q}_h^{{\rm UCB},k} - \check{Q}_h^{{\rm LCB},k} \right) (s, \boldsymbol{a}), \\ \left( \hat{Q}_h^{{\rm UCB},k+1} - \check{Q}_h^{{\rm LCB},k+1} \right) (s, \boldsymbol{a}) = (1 - \eta_n) \left( \hat{Q}_h^{{\rm UCB},k} - \check{Q}_h^{{\rm LCB},k} \right) (s, \boldsymbol{a}) \\ \qquad + \eta_n \left( \left( \hat{V}_{h+1}^k - \check{V}_{h+1}^k \right) (s_{h+1}^k) + 2c_b^{{\rm ULCB}} \right), \\ \left( \hat{V}_h^k - \check{V}_h^k \right) (s) = \left[ \mathbb{D}_{\pi_h} \left( \hat{Q}_h^k - \check{Q}_h^k \right) \right] (s). \end{cases} \tag{245}$$

Similarly, we also have the following definitions.

$$
\begin{cases}
\hat{\phi}_h^{\mathrm{r},k+1}(s,\boldsymbol{a}) = \left(1 - \frac{1}{n}\right)\hat{\phi}_h^{\mathrm{r},k}(s,\boldsymbol{a}) + \frac{1}{n}\hat{V}_{h+1}^{\mathrm{R},k}(s), \\[8pt]
\check{\phi}_h^{\mathrm{r},k+1}(s,\boldsymbol{a}) = \left(1 - \frac{1}{n}\right)\check{\phi}_h^{\mathrm{r},k}(s,\boldsymbol{a}) + \frac{1}{n}\check{V}_{h+1}^{\mathrm{R},k}(s), \\[8pt]
\hat{\psi}_h^{\mathrm{r},k+1}(s,\boldsymbol{a}) = \left(1 - \frac{1}{n}\right)\hat{\psi}_h^{\mathrm{r},k}(s,\boldsymbol{a}) + \frac{1}{n}\left(\hat{V}_{h+1}^{\mathrm{R},k}(s)\right)^2, \\[8pt]
\check{\psi}_h^{\mathrm{r},k+1}(s,\boldsymbol{a}) = \left(1 - \frac{1}{n}\right)\check{\psi}_h^{\mathrm{r},k}(s,\boldsymbol{a}) + \frac{1}{n}\left(\check{V}_{h+1}^{\mathrm{R},k}(s)\right)^2, \\[8pt]
\hat{\phi}_h^{\mathrm{a},k+1}(s,\boldsymbol{a}) = (1 - \eta_n)\hat{\phi}_h^{\mathrm{a},k}(s,\boldsymbol{a}) + \eta_n\left(\hat{V}_{h+1}^k(s) - \hat{V}_{h+1}^{\mathrm{R},k}(s)\right), \\[8pt]
\check{\phi}_h^{\mathrm{a},k+1}(s,\boldsymbol{a}) = (1 - \eta_n)\check{\phi}_h^{\mathrm{a},k}(s,\boldsymbol{a}) + \eta_n\left(\check{V}_{h+1}^k(s) - \check{V}_{h+1}^{\mathrm{R},k}(s)\right), \\[8pt]
\hat{\psi}_h^{\mathrm{a},k+1}(s,\boldsymbol{a}) = (1 - \eta_n)\hat{\psi}_h^{\mathrm{a},k}(s,\boldsymbol{a}) + \eta_n\left(\hat{V}_{h+1}^k(s) - \hat{V}_{h+1}^{\mathrm{R},k}(s)\right)^2, \\[8pt]
\check{\psi}_h^{\mathrm{a},k+1}(s,\boldsymbol{a}) = (1 - \eta_n)\check{\psi}_h^{\mathrm{a},k}(s,\boldsymbol{a}) + \eta_n\left(\check{V}_{h+1}^k(s) - \check{V}_{h+1}^{\mathrm{R},k}(s)\right)^2, \\[8pt]
\widetilde{B}_h^{\mathrm{temp},k+1}(s,\boldsymbol{a}) = c_{\mathrm{b}}\sqrt{\frac{\log^2\frac{S\prod_{i=1}^m A_i T}{\delta}}{n}} \times \Bigg( \\[8pt]
\qquad \left(\sqrt{\hat{\psi}_h^{\mathrm{r},k+1}(s,\boldsymbol{a}) - \left(\hat{\phi}_h^{\mathrm{r},k+1}(s,\boldsymbol{a})\right)^2} + \sqrt{\check{\psi}_h^{\mathrm{r},k+1}(s,\boldsymbol{a}) - \left(\check{\phi}_h^{\mathrm{r},k+1}(s,\boldsymbol{a})\right)^2}\right) \\[8pt]
\qquad + \sqrt{H}\left(\sqrt{\hat{\psi}_h^{\mathrm{a},k+1}(s,\boldsymbol{a}) - \left(\hat{\phi}_h^{\mathrm{a},k+1}(s,\boldsymbol{a})\right)^2} + \sqrt{\check{\psi}_h^{\mathrm{a},k+1}(s,\boldsymbol{a}) - \left(\check{\phi}_h^{\mathrm{a},k+1}(s,\boldsymbol{a})\right)^2}\right)\Bigg); \\[8pt]
\widetilde{\varphi}_h^{\mathrm{R},k+1}(s,\boldsymbol{a}) = \widetilde{B}_h^{\mathrm{temp},k+1}(s,\boldsymbol{a}) - \widetilde{B}_h^{\mathrm{R},k}(s,\boldsymbol{a}); \\[8pt]
\widetilde{B}_h^{\mathrm{R},k+1}(s,\boldsymbol{a}) = \widetilde{B}_h^{\mathrm{temp},k+1}(s,\boldsymbol{a}); \\[8pt]
\widetilde{b}_h^{\mathrm{R},k+1} = \widetilde{B}_h^{\mathrm{R},k+1}(s,\boldsymbol{a}) + (1 - \eta_n)\frac{\widetilde{\varphi}_h^{\mathrm{R},k+1}(s,\boldsymbol{a})}{\eta_n} + 2c_{\mathrm{b}}\frac{H^2\log^2\frac{S\prod_{i=1}^m A_i T}{\delta}}{n^{3/4}}, \\[8pt]
\widetilde{Q}_h^{\mathrm{R},k+1}(s,\boldsymbol{a}) = (1 - \eta_n)\widetilde{Q}_h^{\mathrm{R},k}(s,\boldsymbol{a}) \\[8pt]
\qquad + \eta_n\left(\widetilde{b}_h^{\mathrm{R},k+1} + \left(\hat{V}_{h+1}^k - \check{V}_{h+1}^k\right)(s) - \left(\hat{V}_{h+1}^{\mathrm{R},k} - \check{V}_{h+1}^{\mathrm{R},k}\right)(s) + \left(\hat{\phi}_h^{\mathrm{r},k+1} - \check{\phi}_h^{\mathrm{r},k+1}\right)(s,\boldsymbol{a})\right),
\end{cases}
\tag{246}
$$

where $\hat{V}_h^{\mathrm{R},k}$ and $\check{V}_h^{\mathrm{R},k}$ are associated with $\hat{V}_h^k(s)$ and $\check{V}_h^k(s)$ and updated similar to lines 18-21 in Algorithm 4. Then we can prove inductively that for any $k$, $h$, $s$ and $\boldsymbol{a}$,

$$
\max_i(\overline{V}_{h,i}^k - \underline{V}_{h,i}^k)(s) \le \hat{V}_h^k(s) - \check{V}_h^k(s).
\tag{247}
$$

Therefore, we only need to bound $\sum_{k=1}^K \left(\hat{V}_1^k(s) - \check{V}_1^k(s)\right)$, that is,

$$
\mathrm{Regret}(K) \le \sum_{k=1}^K \max_i \left(\overline{V}_{1,i}^k - \underline{V}_{1,i}^k\right)(s_1^k) \le \sum_{k=1}^K \left(\hat{V}_1^k(s_1^k) - \check{V}_1^k(s_1^k)\right).
\tag{248}
$$

To continue, we intend to examine $\left(\hat{V}_h^k - \check{V}_h^k\right)(s_h^k)$ across all time steps $1 \le h \le H$, which admits the following decomposition:

$$
\begin{aligned}
\hat{V}_h^k(s_h^k) - \check{V}_h^k(s_h^k) &\le \mathbb{E}_{\boldsymbol{a} \sim \pi_h^k}(\hat{Q}_h^k - \check{Q}_h^k)(s_h^k, \boldsymbol{a}) = \hat{Q}_h^k(s_h^k, \boldsymbol{a}_h^k) - \check{Q}_h^k(s_h^k, \boldsymbol{a}_h^k) + \zeta_h^k \\
&\le \widetilde{Q}_h^{\mathrm{R},k}(s_h^k, \boldsymbol{a}_h^k) + \zeta_h^k,
\end{aligned}
\tag{249}
$$

where

$$
\zeta_h^k := \mathbb{E}_{\boldsymbol{a} \sim \pi_h^k}(\hat{Q}_h^k - \check{Q}_h^k)(s_h^k, \boldsymbol{a}) - (\hat{Q}_h^k - \check{Q}_h^k)(s_h^k, \boldsymbol{a}_h^k).
\tag{250}
$$

Summing (248) and (249) over $1 \leq k \leq K$, we reach at

$$\text{Regret}(K) \leq \sum_{k=1}^{K} \widetilde{Q}_1^{\text{R},k}(s_1^k, \boldsymbol{a}_1^k) + \sum_{k=1}^{K} \zeta_1^k. \tag{251}$$

**Step 2: managing regret by recursion.** The regret can be further manipulated by leveraging the update rule of $\hat{Q}_h^{\text{R},k}$ and $\check{Q}_h^{\text{R},k}$, which is similar to that of $\overline{Q}_{h,i}^{\text{R},k}$ and $\underline{Q}_{h,i}^{\text{R},k}$. This leads to a key decomposition as summarized as follows. The proof of Lemma 11 is similar to that of Lemma 5, and is omitted here.

**Lemma 11** *Fix $\delta \in (0, 1)$. Suppose that $c_{\text{b}} > 0$ is a sufficiently large constant. Then with probability at least $1 - \delta$, one has*

$$\sum_{k=1}^{K} \hat{V}_1^k(s_1^k) - \check{V}_1^k(s_1^k) \leq \mathcal{H}_1 + \mathcal{H}_2 + \mathcal{H}_3, \tag{252}$$

*where*

$$\mathcal{H}_1 = \sum_{h=1}^{H} \left(1 + \frac{1}{H}\right)^{h-1} \left(2HS \prod_{i=1}^{m} A_i + 16c_{\text{b}}(S \prod_{i=1}^{m} A_i)^{3/4} K^{1/4} H^2 \log \frac{S \prod_{i=1}^{m} A_i T}{\delta} + \sum_{k=1}^{K} \zeta_h^k\right), \tag{253a}$$

$$\mathcal{H}_2 = \sum_{h=1}^{H} \left(1 + \frac{1}{H}\right)^{h-1} \left(\sum_{k=1}^{K} \widetilde{B}_h^{\text{R},k}\left(s_h^k, \boldsymbol{a}_h^k\right)\right), \tag{253b}$$

$$\mathcal{H}_3 = \sum_{h=1}^{H} \sum_{k=1}^{K} \lambda_h^k \left(P_h^k - P_{h,s_h^k,\boldsymbol{a}_h^k}\right)\left(\check{V}_{h+1}^{\text{R},k} - \hat{V}_{h+1}^{\text{R},k}\right)$$

$$+ \sum_{h=1}^{H} \sum_{k=1}^{K} \lambda_h^k \frac{\sum_{i=1}^{N_h^k(s_h^k,\boldsymbol{a}_h^k)} \left(\hat{V}_{h+1}^{\text{R},k^i}\left(s_{h+1}^{k^i}\right) - P_{h,s_h^k,\boldsymbol{a}_h^k}\hat{V}_{h+1}^{\text{R},k}\right)}{N_h^k\left(s_h^k, \boldsymbol{a}_h^k\right)}$$

$$- \sum_{h=1}^{H} \sum_{k=1}^{K} \lambda_h^k \left(\frac{\sum_{i=1}^{N_h^k(s_h^k,\boldsymbol{a}_h^k)} \left(\check{V}_{h+1}^{\text{R},k^i}\left(s_{h+1}^{k^i}\right) - P_{h,s_h^k,\boldsymbol{a}_h^k}\check{V}_{h+1}^{\text{R},k}\right)}{N_h^k\left(s_h^k, \boldsymbol{a}_h^k\right)}\right) \tag{253c}$$

*with*

$$\lambda_h^k = \left(1 + \frac{1}{H}\right)^{h-1} \sum_{N=N_h^k\left(s_h^k,a_h^k,b_h^k\right)}^{N_h^{K-1}\left(s_h^k,a_h^k,b_h^k\right)} \eta_{N_h^k\left(s_h^k,a_h^k,b_h^k\right)}^{N}. \tag{254}$$

**Step 3: controlling the terms in Step 2 separately.** Each of the terms in Step 2 can be well controlled. To derive the above bounds, the main strategy is to apply the Bernstein-type concentration inequalities carefully, and to upper bound the sum of variance. We provide the bounds for these terms as Lemma 12. The proof is similar to that of Lemma 6 and is omitted here.

**Lemma 12** *With any $\delta \in (0, 1)$, the following upper bounds hold with probability at least $1 - \delta$:*

$$\mathcal{H}_1 \lesssim \sqrt{H^2 S \prod_{i=1}^{m} A_i T \log \frac{S \prod_{i=1}^{m} A_i T}{\delta}} + H^{4.5} S \prod_{i=1}^{m} A_i \log^2 \frac{S \prod_{i=1}^{m} A_i T}{\delta}, \tag{255}$$

$$\mathcal{H}_2 \lesssim \sqrt{H^2 S \prod_{i=1}^{m} A_i T \log \frac{S \prod_{i=1}^{m} A_i T}{\delta}} + H^4 S \prod_{i=1}^{m} A_i \log^2 \frac{S \prod_{i=1}^{m} A_i T}{\delta}, \tag{256}$$

$$\mathcal{H}_3 \lesssim \sqrt{H^2 S \prod_{i=1}^{m} A_i T \log^4 \frac{S \prod_{i=1}^{m} A_i T}{\delta}} + H^6 S \prod_{i=1}^{m} A_i \log^3 \frac{S \prod_{i=1}^{m} A_i T}{\delta}. \tag{257}$$

**Step 4: putting all this together.** We now establish our main result. Taking the bounds in Step 3 together with Step 2, we see that with probability at least $1 - \delta$ and a constant $C_0 > 0$, one has

$$\text{Regret}(K) \leq \mathcal{H}_1 + \mathcal{H}_2 + \mathcal{H}_3$$

$$\leq C_0 \left( \sqrt{H^2 S \prod_{i=1}^{m} A_i T \log^4 S \prod_{i=1}^{m} A_i T/\delta} + H^6 S \prod_{i=1}^{m} A_i \log^3 S \prod_{i=1}^{m} A_i T/\delta \right). \quad (258)$$

Theorem 3 is proved under sample complexity with $\varepsilon$ average regret (i.e., $\frac{1}{K} \text{Regret}(K) \leq \varepsilon$). Notably, the sample complexity is proportional to $\prod_{i=1}^{m} A_i$, which increases exponentially as the number of players increases.

