# OpenReview forum: "Provable Memory Efficient Self-Play Algorithm for Model-free Reinforcement Learning"
_ICLR.cc/2024/Conference — ICLR 2024 poster_

### Official Review · Reviewer_nN5J · 2023-10-21

**Soundness:** 3 good
**Presentation:** 3 good
**Contribution:** 3 good
**Rating:** 8
**Confidence:** 4

**Summary:**

This paper studies provably efficient reinforcement learning in two-player zero-sum Markov games, an important special case of multi-agent RL. This paper improves existing results in the following directions: sample complexity, memory efficiency, Markov output policy, and burn-in cost.

**Strengths:**

This paper studies an important topic in MARL theory. The proposed algorithm simultaneously achieved state-of-the-art results in all the aspects it considers: It matches the best sample complexity bounds, reduces the burn-in cost, and improves the space complexity while still outputting a Markov policy. The theoretical analysis looks solid.

**Weaknesses:**

I reviewed this paper at NeurIPS 2023. My concern was about the technical novelty of the paper because the proposed algorithm follows the mature framework of Nash Q-learning. The improved sample complexity is achieved by also following an existing reference-advantage decomposition technique.

In terms of the bounds, the biggest improvements that this paper makes over existing works are regarding the space complexity and burn-in cost. In my opinion, these are less important metrics compared to sample complexity or time complexity, yet this work has to optimize these metrics at the cost of a much more complicated algorithm and proof procedure.
While I still hold most of my previous opinions, I appreciate the authors’ effort in improving their work and would like to increase my score compared to my NeurIPS evaluation.

Compared to the NeurIPS submission, the new major results are Theorems 2 and 3. I found that the extension to multi-player general-sum games (Theorem 3) particularly interesting, but I was not able to find any algorithm or proof for this theorem. What is the learning target for general-sum games, Nash or correlated equilibria?

**Questions:**

1.	Could you please point me to the proofs of Theorem 3? Also what is the algorithm for this theorem (as I assume that Algorithm 1 only applies to two-player zero-sum games)? I do not think the extension from zero-sum to multi-player general-sum is straightforward and would hope to see a more detailed discussion.

2.	Since you now also consider multi-player general-sum games, it is probably helpful to include related works for learning in general-sum games, especially those using Nash V-learning (to name a few):

a. Song, Ziang, Song Mei, and Yu Bai. "When can we learn general-sum Markov games with a large number of players sample-efficiently?." arXiv preprint arXiv:2110.04184 (2021).

b. Mao, Weichao, and Tamer Başar. "Provably Efficient Reinforcement Learning in Decentralized General-Sum Markov Games." arXiv preprint arXiv:2110.05682 (2021).

c. Daskalakis, Constantinos, Noah Golowich, and Kaiqing Zhang. "The complexity of markov equilibrium in stochastic games." The Thirty Sixth Annual Conference on Learning Theory. PMLR, 2023.

3.	From what I understsand, the new major results compared to the NeurIPS submission are Theorems 2 and 3. Could you please let me know if there are any other new results that I am missing?

4.	In your future work, you mentioned the possibility of achieving A+B sample complexity instead of AB. Does the Nash V-Learning algorithm help with this?

---

> ### Author Response · Authors · 2023-11-17
>
> **Weakness:** Thank you for providing your insightful feedback once again. We greatly respect your opinions and appreciate your openness to various research directions. We would like to take this opportunity to provide a more detailed explanation of our ideas.
>
> We first make use of the reference-advantage decomposition (also known as a variant of variance reduction) for Markov games to achieve sample complexity $\widetilde{O}(H^4SAB/\varepsilon^2)$, space complexity $O(SABH)$, computational complexity $O(T \mathrm{poly}(SAB))$, and burn-in cost $O(SAB\mathrm{poly}(H))$, which is never shown in previous works. Specifically, we design a pair of optimistic and pessimistic value functions and an early-settlement method to get such results. This is the first time to show the effectiveness of these ideas in Markov games, which is of significant importance in our opinion. We have stated these at the beginning of Section 3.1 in the revised version.
>
> In terms of performance metrics, we have several noteworthy points to highlight. On the one hand, as per your previous suggestions, we have provided a comprehensive comparison of time complexity as shown in Table 1. It demonstrates that our algorithm achieves the best time complexity. On the other hand, the complex algorithm and proof process may not necessarily pose practical challenges for the algorithm implementation. On the contrary, we address a current research gap, which we think is more important.
>
> Finally, we apologize for not presenting the implications of Theorem 3 in our initial submission. In the revised version, we have included additional explanations in Appendix F to address this concern.
>
> **Q1:** Thank you for your comments and sorry for the confusion. We have included the algorithm and proof for multi-player general-sum games in Appendix F to address your comment in the revised version.
>
> **Q2:** Thank you for your comments. In the revised version, we have included a paragraph in Section 1.2 to introduce the related works on multi-player general-sum Markov games.
>
> **Q3:** Thank you for your question. The new major results compared with the NeurIPS version are indeed Theorems 2 and 3. Additionally, we have made several improvements. On the one hand, we have improved the presentation of Table 1 which compares various algorithms in terms of several performance metrics. Specifically, We have added a comparison of whether the output policy is a Markov/Nash-VI policy and a comparison of computational complexity. On the other hand, according to the feedback from the NeurIPS reviewers, we have provided a more detailed explanation of our reference-advantage decomposition and early-settlement methods in Section 3.1 on page 7.
>
> **Q4:** Thank you for your question. Currently, we do not have a clear plan for achieving a sample complexity of $O(A+B)$. Algorithms such as Nash V-learning, or those based on V-learning, achieve a sample complexity of $O(A+B)$ because they do not need to store the Q-function and only require the Value function. It seems that they may help in future work. However, the policies learned by V-learning are non-Markov and history-dependent in general. It remains to study how to achieve a sample complexity of $O(A+B)$ without compromising the performance of existing metrics. We have discussed this in the conclusion of the revised version.

---

> > ### Comment · Reviewer_nN5J · 2023-11-17
> >
> > I thank the authors for the response and for including the new results & proofs in Appendix F. They look good to me and I am happy to increase my rating to "8: accept, good paper".

---

> > > ### Author Response · Authors · 2023-11-18
> > >
> > > I would like to express my sincere gratitude for your thoughtful consideration and efforts in refining our work. Your prompt and positive feedback stands as a meaningful source of encouragement for us.

---

### Official Review · Reviewer_M7D7 · 2023-10-31

**Soundness:** 4 excellent
**Presentation:** 3 good
**Contribution:** 3 good
**Rating:** 8
**Confidence:** 3

**Summary:**

This paper studies two-player zero-sum Markov games (TZMG). It proposes the model-free algorithm Memory-Efficient Nash Q-Learning (ME-Nash-QL), which achieves state-of-the-art space and computational complexity, nearly optimal sample complexity, and the best burn-in cost compared to previous results with the same sample complexity. Moreover, the proposed algorithm generates a single Markov and Nash policy rather than a nested mixture of Markov policies, by computing a relaxation of the Nash equilibrium instead, i.e. Coarse Correlated Equilibrium (CCE).

**Strengths:**

# Originality
- The related works are covered in detail.
# Quality
- The theoretical proofs seem to be rigorous.
# Clarity
- This paper is in general well-written and easy to follow. The design idea of the algorithm is clearly explained.
# Significance
- The theoretical results of this work are strong. It achieves state-of-the-art space and computational complexity, nearly optimal sample complexity, and the best burn-in cost compared to previous results with the same sample complexity.
- TZMG is foundational and critically significant for MARL. This research has the potential to establish a new benchmark, providing a foundation for further studies in the related literature.

**Weaknesses:**

- Although the proposed algorithm is compared to Nash-VI (Liu et al., July 2021) and V-learning (Jin et al., 2022) in detail, the design idea of the proposed algorithm seems to share certain similarities with those from the two works. For example, they all compute a CCE policy and take the marginal policies; the choice of learning rate $\frac{H+1}{H+N}$, the form of bonus terms, and the update of lower and upper bounds for Q-functions are similar. The originality of this paper could be significantly enhanced if the authors could discuss thoroughly the fundamental distinctions between the ideas of the proposed algorithm and the aforementioned Nash-VI and V-learning.
- The theoretical findings are limited to the TZMG and CCE setting, which somewhat diminishes the overall contribution of this paper.
- The auxiliary functions in Algorithm 2 are too nested, making it hard to read.
### Minor:
- There seems to be a blank section A.3.1 on page 14.

**Questions:**

- How is $\operatorname{CCE}(\bar{Q}, Q)$ compuated? I was anticipating a detailed introduction to its calculation to ensure the paper's comprehensiveness. An explicit explanation would greatly contribute to the paper's self-containment.
- Is the achievement of the space complexity independent of $T$ attributed to the fact that the output policy is a single Markov policy? In this context, do the authors consider the CCE as an essential relaxation for realizing such space complexity?

---

> ### Author Response · Authors · 2023-11-17
>
> **Weakness 1:** We would like to express our sincere gratitude for your valuable comments. The fundamental distinctions between the proposed algorithm and reference algorithms (Nash-VI and V-learning) are the application of the ideas of variance reduction and the early-settlement method. Moreover, we first show the effectiveness of these ideas in TZMG by proving sample complexity $\widetilde{O}(H^4SAB/\varepsilon^2)$, space complexity $O(SABH)$, computational complexity $O(T \mathrm{poly}(SAB))$, and burn-in cost $O(SAB\mathrm{poly}(H))$.
>
> Nash-VI uses the model-based idea and designs a different style of bonus term, which achieves the sample complexity $\widetilde{O}(H^4SAB/\varepsilon^2)$ with burn-in cost $O(S^3ABH^4)$. However, Nash-VI needs space complexity $O(S^2ABH)$ to store the empirical transition matrix. Moreover, it computes the output policy for all states $s\in \mathcal{S}$ in each step, which leads to a computational complexity of $S$ times higher than that of our algorithm.
>
> V-learning uses the FTRL idea (from adversarial bandit, i.e., $H = 1$), which overcomes the challenges posed by the curse of multi-agent, and achieves the sample complexity $\widetilde{O}(H^6S(A+B)/\varepsilon^2)$. However, this sample complexity depends greatly on $H$. Besides, the output policy of V-learning is neither **Nash** nor Markov policy due to the incremental updates of V-learning and causes the space complexity $O(S(A+B)T)$ to increase with the number of samples $T$.
>
> To address the aforementioned issues, we have designed the ME-Nash-QL. First, we propose a model-free algorithm to shave the $S$ factor in space complexity compared with Nash-VI. Secondly, we have successfully obtained a lower sample complexity and a lower burn-in cost by designing a pair of optimistic and pessimistic value functions along with an early-settlement method based on the reference-advantage decomposition technique, which is a variant of variance reduction. Specifically, as detailed in Section 3.1, the standard deviation of $\widehat{P} _{h,s,a,b}\big(\overline{V} _{h+1}-\overline{V}^\mathrm{R} _{h+1}\big)$ might be $O(H)$ times smaller than the stochastic term used in Nash Q-learning. This is a key observation to weaken the dependency of the regret bound on $H$, and obtain the burn-in cost $\widetilde{O}(SABH^{10})$. Thirdly, our algorithm computes marginal policies using a CCE subroutine for a specific state in each step, which leads to a lower space complexity $(SABH)$ and a lower computational complexity $T{\rm poly}(AB)$ compared with Nash VI, and meanwhile a Markov and Nash output policy. We have clarified these in Section 1 in the revised version.
>
> **Weakness 2:** Thanks for your insightful feedback. Regarding the TZMG setting, we have extended it to multi-player general-sum Markov games in Theorem 3. We have also included a detailed explanation of this in Appendix F in the revised version.
>
> For the CCE setting, we would like to discuss it in two cases, i.e., TZMG and multi-player general-sum Markov games. In the former case, CCE, CE, and NE are equivalent, meaning that CCE in zero-sum games is guaranteed to be Nash. Thus our algorithm achieves NE in this case. For the latter case, we can achieve CCE, CE, and NE by replacing the subroutine in line 15 in Algorithm 3. However, it's important to note that NE leads to a much higher computational complexity, such as PPAD-complete (a complexity class widely believed to be computationally challenging). We have clarified these in Appendix F in the revised version.
>
> **Weakness 3:** Thanks for your comments. We have reorganized Algorithm 2 in the revised version to enhance its readability.
>
> **Minor:** Thanks for your comments, and we apologize for this typo. We have corrected this in the revised version.
>
> **Q1:** Thank you for your question. There are many well-established computational methods to calculate CCE, as demonstrated by Xie et al. in June 2022. We have supplemented relevant explanations on page 8 in the revised version of our paper to make it self-contained.
>
> Qiaomin Xie, et al., Learning zero-sum simultaneous-move Markov games using function approximation and correlated equilibrium.
>
> **Q2:** Thanks for your questions. For the first question, the fact that space complexity remains independent of $T$ is attributed to the fact that the output policy is a single Markov policy.
>
> For the second question, our decision to opt for CCE over NE is primarily motivated by the goal of reducing computational complexity, rather than space complexity. This choice is driven by the understanding that NE methods often come with high computational complexity, involving PPAD-complete problems (widely believed to be computationally challenging). In other words, we could achieve NE with the same space complexity by replacing CCE with NE in the algorithm, but the computational complexity is prohibitive.
>
> We have clarified these at the end of Section 3.1 and Appendix F.2 in the revised version.

---

> > ### Comment · Reviewer_M7D7 · 2023-11-22
> >
> > I thank the authors for providing detailed explanations for my concerns, and for the extended results in Appendix F. They all look good to me so I increased my rating to 8.

---

> > > ### Author Response · Authors · 2023-11-22
> > >
> > > Thank you for your feedback and your increased score. We extend our gratitude for your insights and the considerable time you've dedicated to this review process.

---

### Official Review · Reviewer_3yJL · 2023-10-31

**Soundness:** 3 good
**Presentation:** 3 good
**Contribution:** 3 good
**Rating:** 6
**Confidence:** 3

**Summary:**

This paper introduces a model-free algorithm for two-player zero-sum Markov game, which enjoys low sample complexity and computational/space complexity. The resulting algorithm has optimal dependency on S and H but sub-optimal dependence on the number of actions. The algorithm design features the early-settlement method and the reference-advantage decomposition technique.

**Strengths:**

+ The paper is well written and easy to follow.
+ The proposed algorithm outperforms existing algorithms in terms of space complexity and computational complexity.

**Weaknesses:**

- My main concern is the technical novelty. The reference-advantage decomposition technique has already been incorporated in two-player zero-sum Markov game by Feng el al (2023) (not cited by this work), which achieves a regret in \tilde{O}(\sqrt{H^2SABT}) and matches with the regret bound in this work. The main novelty of the algorithm design thus lies in the early-settlement design in order to reduce the burn-in cost, which is not new in the literature.

Feng, S., Yin, M., Wang, Y. X., Yang, J., & Liang, Y. (2023). Model-Free Algorithm with Improved Sample Efficiency for Zero-Sum Markov Games. arXiv preprint arXiv:2308.08858.

**Questions:**

+ Regarding my point in weakness section, is there any other technical contributions besides reference-advantage decomposition and early-settlement design?

+ Is it possible to obtain similar result for learning CCE in multi-agent general-sum Markov games?

---

> ### Author Response · Authors · 2023-11-13
> **A gentle reminder for the reviewer guideline**
>
> We would like to express our cordial thanks to the reviewer for your precious time and feedback on our paper. However, it appears that there might be some misunderstanding regarding the reviewer guideline (https://iclr.cc/Conferences/2024/ReviewerGuide).
>
> According to the guideline, it states: "*We consider papers contemporaneous if they are published (available in online proceedings) within the last four months. That means, since our full paper deadline is September 28, if a paper was **published** (i.e., at a **peer-reviewed venue**) on or after **May 28, 2023**, authors are not required to compare their own work to that paper.*"
>
> The work by Feng et al. was available on **arXiv** on **August 17, 2023**, which is contemporaneous with our work.  We appreciate the reviewer for bringing this reference to our attention, and we will ensure to cite it in the final revised version of our paper.

---

> > ### Comment · Reviewer_3yJL · 2023-11-20
> >
> > I understand the reviewer guideline and I acknowledge that those works are concurrent. I was wondering whether you can provide a more detailed comparison between those two works in terms of algorithm design and analysis, and whether such comparison will be reflected in the revised paper. Usually researchers compare concurrent works in detail, especially the potential overlapping parts.

---

> > > ### Author Response · Authors · 2023-11-21
> > >
> > > Thank you for your valuable feedback and suggestions. We agree with you that we should compare with any related work. We **do not mean that there is no need for us to compare with the concurrent work** by citing the reviewer guideline. Actually, in our previous response, we have said that **we would cite and compare this work properly**.
> > >
> > > We are sorry that we did not provide the detailed comparison earlier because we treat this as a minor point and that the time is limited. In response to your suggestion, we have now incorporated a comprehensive comparison in Section 1.2 of the revised manuscript: "During the preparation of our work, we observed that (Feng et al., 2023) also employs CCE and the reference-advantage decomposition technique, achieving comparable sample and computational complexity with ours. The differences in algorithm design are as follows: algorithm in (Feng et al., 2023) uses the stage-based update approach and the certified policy, while our algorithm leverages the early-settlement technique and outputs the marginal policy. Consequently, our algorithm outperforms by achieving a lower burn-in cost of $SABH^{10}$ than $S^6A^4B^4H^{28}$ in their work and outputting Nash and Markov policies."
> > >
> > > We hope this detailed comparison adequately addresses your concerns. Please feel free to provide any further feedback. We are here to assist and provide any additional details you might require.

---

> > > > ### Comment · Reviewer_3yJL · 2023-11-22
> > > >
> > > > Thank you for the response. It would better if Feng et al (2023) is included in Table 1 as well.

---

> > > > > ### Author Response · Authors · 2023-11-22
> > > > >
> > > > > Thank you for your feedback and suggestions. We have added Feng et al (2023) in Table 1 to offer a more in-depth comparison of our approach, covering specific aspects you highlighted during the review.
> > > > >
> > > > > To make it clear, we shall show how we calculate the computational complexity, space complexity, and burn-in cost. Akin to (Zhang et al., 2020b), we can calculate that the number of policy updates in the algorithm of (Feng et al., 2023) is $H^2SAB$, meaning that lines 9-20 in Algorithm 1 of (Feng et al., 2023) are implemented $H^2SAB$ times. Based on this result, the computational complexity consists of two parts: lines 6-7 in Algorithm 1 require a total computation of $O(T)$, and CCE has computational complexity $\mathrm{poly}(AB)$, so the overall computational complexity is $T+H^2\mathrm{poly}(SAB)$, which is lower than ours. Notably, we have stated this in Section 1.2 in the revised manuscript. For the space complexity, since the algorithm in (Feng et al., 2023) employs the certified policy for computing the output policy, each updated policy needs to be stored, resulting in a space complexity of $H^2SAB\times AB=SA^2B^2H^2$. Additionally, based on Lemma 4.6 in (Feng et al., 2023), we can derive that the burn-in cost is $S^6A^4B^4H^{28}$.
> > > > >
> > > > > If you have any additional comments, please don't hesitate to inform us.
> > > > >
> > > > > Zihan Zhang, Yuan Zhou, and Xiangyang Ji. Almost optimal model-free reinforcement learning via reference-advantage decomposition. In Proceedings of the 34th International Conference on Neural Information Processing Systems, Red Hook, NY, USA, 2020b. Curran Associates Inc. ISBN 9781713829546.

---

> > > > > > ### Comment · Reviewer_3yJL · 2023-11-22
> > > > > >
> > > > > > Thank you for your response. I have increased my rating based on the revision.

---

> > > > > > > ### Author Response · Authors · 2023-11-23
> > > > > > >
> > > > > > > Thank you immensely for your review and the insightful feedback. We appreciate your consideration and understanding for the increased score. Your support and suggestions are valuable to us.

---

> ### Author Response · Authors · 2023-11-17
>
> **Q1:** Thank you for your question. Besides applying the reference-advantage decomposition and early-settlement techniques, our main technical contribution is to show their effectiveness by proving sample complexity $\widetilde{O}(H^4SAB/\varepsilon^2)$, space complexity $O(SABH)$, computational complexity $O(T \mathrm{poly}(SAB))$, and burn-in cost of $O(SAB\mathrm{poly}(H))$. In addition, we extend our algorithm from the TZMG setting to multi-player general-sum Markov games in Theorem 3, which is also one of our contributions. We have stated these in Section 1.1 in the revised version.
>
> **Q2:** Thank you for your question. Similar conclusions can be obtained in multi-player general-sum Markov games, as stated in Theorem 3 in the main text. We have added Appendix F in the revised version to provide detailed supplements for Theorem 3.

---

### Official Review · Reviewer_Pgbr · 2023-11-01

**Soundness:** 3 good
**Presentation:** 3 good
**Contribution:** 3 good
**Rating:** 6
**Confidence:** 3

**Summary:**

This paper proposes a model-free algorithm for learning Nash policy in Two-player Zero-sum Markov Game. The authors prove that this algorithm enjoy many benign properties, including outputting Markov policy, low computational/sample/space complexity in certain regime and low burn-in cost.

**Strengths:**

The proposed algorithm enjoys several benign properties, as mentioned in the summary. In particular, the algorithm perform well when the horizon is very long while retaining other nice properties such as Markov output policy and low burn-in cost.

**Weaknesses:**

1. The proposed algorithm does not break the curse of multi-agent. Although the authors argue that there are many scenarios where horizon length is very long, I still feel that this is not general enough. I personally would still be more interested in algorithms that have $O(A+B)$ dependence in complexity.
2. The algorithmic novelty is a bit unclear to me.

**Questions:**

1. There are many elements mentioned in the paper, such as complexity, burn-in cost, Nash policy, Markov policy etc. While I understand that no prior algorithm surpassing this algorithm in every aspect, I wonder what do the authors think is the most important aspect/what is the main focus?
2. Can the authors explain what is the most salient algorithmic novelty to the newly proposed algorithm?

---

> ### Author Response · Authors · 2023-11-17
>
> **Weakness 1:** Thank you for your feedback and concerns regarding the issue of the curse of multi-agent. We do agree that some research works focus on the curse of multi-agent, but there indeed exists a lot of attention on the dependency of long-horizon and size of state space, such as Nash Q-learning (Bai et al., 2020), OMNI-VI (Xie et al., Jun. 2022), Nash-UCRL (Chen et al., 2022), Optimistic PO (Qiu et al., 2021), VI-Explore/VI-UCLB (Bai \& Jin, 2020), and Nash-VI (Liu et al., July 2021). Compared with them, we design a model-free self-play algorithm for TZMG that improves performance from three aspects simultaneously, including sample complexity ($\widetilde{O}(H^4SAB/\varepsilon^2)$), space complexity ($O(SABH)$), and computational complexity ($O(T \mathrm{poly}(SAB))$). Furthermore, our algorithm achieves a $\sqrt{T}$-regret bound with the best burn-in cost of $O(SAB\mathrm{poly}(H))$ as long as $\min\{A, B\}\ll H^2$. In addition, our algorithm produces a Markov and Nash policy as output. To the best of our knowledge, these algorithms with $O(A+B)$ sample complexity do not output Nash policies. We consider our work to be of significant research value and hope that ME-Nash-QL can inspire other researchers to achieve a $O(A+B)$ dependency in their work without compromising the performance of existing metrics. To address the reviewer's concern, we have discussed the curse of multi-agent in the conclusion of the revised version.
>
> Yu Bai, Chi Jin, and Tiancheng Yu. Near-optimal reinforcement learning with self-play.
>
> Qiaomin Xie, Yudong Chen, Zhaoran Wang, and Zhuoran Yang. Learning zero-sum simultaneous-move Markov games using function approximation and correlated equilibrium.
>
> Zixiang Chen, Dongruo Zhou, and Quanquan Gu. Almost optimal algorithms for two-player zero-sum linear mixture Markov games.
>
> Shuang Qiu, Xiaohan Wei, Jieping Ye, Zhaoran Wang, and Zhuoran Yang. Provably efficient fictitious play policy optimization for zero-sum Markov games with structured transitions.
>
> Yu Bai and Chi Jin. Provable self-play algorithms for competitive reinforcement learning.
>
> Qinghua Liu, Tiancheng Yu, Yu Bai, and Chi Jin. A sharp analysis of model-based reinforcement learning with self-play.
>
> **Weakness 2:** We express our heartfelt gratitude for your feedback. The algorithmic novelty lies in two aspects. First, we apply the reference-advantage decomposition technique to the design of Markov game algorithms and achieve the $\tilde{O}(H^4SAB/\varepsilon^2)$ sample complexity. Second, we use an innovative early-settlement approach, substantially reducing burn-in costs. Notably, we are the first to integrate the reference-advantage decomposition and early-settlement technique into the domain of two-player zero-sum Markov games, and successfully prove the improvement of the performance. To make it clear, we have stated this at the beginning of Section 3.1.
>
> **Q1:** Thank you for your question. While we believe that each performance metric plays a crucial role in assessing algorithm quality, if we were to choose one as the most important, we would consider **sample complexity** and **burn-in cost** to be paramount. These two metrics are closely tied to the amount of data required and have a significant impact on data-driven learning algorithms, especially in tasks that involve expensive or hazardous data collection, such as autonomous driving. Our work focuses on the context of online learning, which means that algorithms need to excel in terms of real-time adaptability to the environment or task. This underscores the significance of sample complexity and burn-in cost. In addition, we think the most important role in multi-player general-sum Markov games may output the **Nash policy**, which is deficient in the existing research.
>
> **Q2:** Thank you for your question. As we outlined in our response to Weakness 2, our main contribution lies in the innovative adaptation and application of the reference-advantage decomposition technique and the early-settlement method to two-player zero-sum Markov games. We first show the effectiveness of these ideas by proving the $\tilde{O}(H^4SAB/\varepsilon^2)$ sample complexity and the $O(SAB\mathrm{poly}(H))$ burn-in cost for the proposed model-free TZMG algorithms. We have stated this at the beginning of Section 3.1.

---

> > ### Comment · Reviewer_Pgbr · 2023-11-20
> >
> > Thank you for your explanations! After confirming that sample complexity/burn-in cost are the most important elements and reference-advantage/early settlement are designed to improve the result, this paper is now clearer to me. Thus I choose to increase my score to 6.

---

> > > ### Author Response · Authors · 2023-11-20
> > >
> > > Thank you for your time, expertise, and positive acknowledgment. We are particularly grateful for the additional score you allocated, and truly appreciate your efforts in advancing the quality and impact of our work.

---

### Meta-Review · Area_Chair_tiYi · 2023-12-07

**Metareview:**

This paper studies provably efficient reinforcement learning in two-player zero-sum Markov games, an important special case of multi-agent RL. This paper improves existing results in the following directions: sample complexity, memory efficiency, Markov output policy, and burn-in cost.
The problem studied in this paper is a technically challenging problem. All reviewers appreciate the contributions to this paper. The AC agrees and thus recommends acceptance.

**Justification For Why Not Higher Score:**

While this paper improves upon existing work on some aspects. It is still far from optimal.

**Justification For Why Not Lower Score:**

This paper studies a very challenging problem and the improvements are non-trivial.

---

### Decision · Program_Chairs · 2024-01-16

Accept (poster)